# Hyperparameter Optimization Is Deceiving Us, and How to Stop It

**A. Feder Cooper**[*]
Cornell University
afc78@cornell.edu

**Yucheng Lu**
Cornell University
yl2967@cornell.edu

**Jessica Zosa Forde**
Brown University
jforde2@cs.brown.edu

**Christopher De Sa**
Cornell University
cdesa@cs.cornell.edu

## Abstract

Recent empirical work shows that inconsistent results based on choice of hyper-parameter optimization (HPO) configuration are a widespread problem in ML research. When comparing two algorithms $\mathcal{J}$ and $\mathcal{K}$, searching one subspace can yield the conclusion that $\mathcal{J}$ outperforms $\mathcal{K}$, whereas searching another can entail the opposite. In short, the way we choose hyperparameters can deceive us. We provide a theoretical complement to this prior work, arguing that, to avoid such deception, the process of drawing conclusions from HPO should be made more rigorous. We call this process *epistemic hyperparameter optimization* (EHPO), and put forth a logical framework to capture its semantics and how it can lead to inconsistent conclusions about performance. Our framework enables us to prove EHPO methods that are guaranteed to be defended against deception, given bounded compute time budget $t$. We demonstrate our framework's utility by proving and empirically validating a defended variant of random search.

## 1 Introduction

Machine learning can be informally thought of as a double-loop optimization problem. The *inner loop* is what is typically called *training*: It learns the parameters of some model by running a training algorithm on a training set. This is usually done to minimize some training loss function via an algorithm such as stochastic gradient descent (SGD). Both the inner-loop training algorithm and the model are parameterized by a vector of *hyperparameters* (HPs). Unlike the learned output parameters of a ML model, HPs are inputs provided to the learning algorithm that guide the learning process, such as learning rate and network size. The *outer-loop* optimization problem is to find HPs (from a set of allowable HPs) that result in a trained model that performs the best in expectation on "fresh" examples drawn from the same source as the training set, as measured by some loss or loss approximation. An algorithm that attempts this task is called a *hyperparameter optimization* (HPO) procedure [12, 20].

From this setup comes the natural question: How do we pick the subspace for the HPO procedure to search over? The HPO search space is enormous, suffering from the curse of dimensionality; training, which is also expensive, has to be run for each HP configuration tested. Thus, we have to make hard choices. With limited compute resources, we typically pick a small subspace of possible HPs and perform grid search or random search over that subspace. This involves comparing the empirical performance of the resulting trained models, and then reporting on the model that performs best in

---

[*]Corresponding author: https://cacioepe.pe

35th Conference on Neural Information Processing Systems (NeurIPS 2021).

terms of a chosen validation metric [20, 37, 41]. For grid search, the grid points are often manually set to values put forth in now-classic papers as good rules-of-thumb concerning, for example, how to set the learning rate [36, 45, 46, 56]. In other words, how we choose which HPs to test can seem rather ad-hoc. We may have a good rationale in mind, but we often elide the details of that rationale on paper; we choose an HPO configuration without explicitly justifying our choice.

Much recent empirical work has critiqued this practice [7, 11, 16, 48, 50, 53, 62, 65]. The authors examine HPO configuration choices in prior work, and find that those choices can have an outsize impact on convergence, correctness, and generalization. They therefore argue that more attention should be paid to the origins of empirical gains in ML, as it is often difficult to tell whether measured improvements are attributable to training or to well-chosen (or lucky) HPs. Yet, this empirical work does not suggest a path forward for formalizing this problem or addressing it theoretically.

To this end, **we argue that the process of drawing conclusions using HPO should itself be an object of study**. Our contribution is to put forward, to the best of our knowledge, the first theoretically-backed characterization for making trustworthy conclusions about algorithm performance using HPO. We model theoretically the following empirically-observed problem: When comparing two algorithms, $\mathcal{J}$ and $\mathcal{K}$, searching one subspace can pick HPs that yield the conclusion that $\mathcal{J}$ outperforms $\mathcal{K}$, whereas searching another can select HPs that entail the opposite result. In short, the way we choose hyperparameters can deceive us—a problem that we call *hyperparameter deception*. We formalize this problem, and prove and empirically validate a defense against it. Importantly, our proven defense does not make any promises about ground-truth algorithm performance; rather, it is guaranteed to avoid the possibility of drawing inconsistent conclusions about algorithm performance within some bounded HPO time budget $t$. In summary, we:

- Formalize the process of drawing conclusions from HPO (epistemic HPO, Section 3).

- Leverage the flexible semantics of modal logic to construct a framework for reasoning rigorously about 1) uncertainty in epistemic HPO, and 2) how this uncertainty can mislead the conclusions drawn by even the most well-intentioned researchers (Section 4).

- Exercise our logical framework to demonstrate that it naturally suggests defenses with guarantees against being deceived by EHPO, and offer a specific, defended-random-search EHPO (Section 5).

## 2 Preliminaries: Problem Intuition and Prevalence in ML Research

Principled HPO methods include *grid search* [41] and *random search* [3]. For the former, we perform HPO on a grid of HP-values, constructed by picking a set for each HP and taking the Cartesian product. For the latter, the HP-values are randomly sampled from chosen distributions. Both of these HPO algorithms are parameterized themselves: Grid search requires inputting the spacing between different configuration points in the grid, and random search requires distributions from which to sample. We call these HPO-procedure-input values *hyper-hyperparameters* (hyper-HPs).[2] To make HPO outputs comparable, we also introduce the notion of a *log*:

**Definition 1.** *A log $\ell$ records all the choices and measurements made during an HPO run, including the total time $T$ it took to run. It has all necessary information to make the HPO run reproducible.*

A log can be thought of as everything needed to produce a table in a research paper: code, random seed, choice of hyper-HPs, information about the learning task, properties of the learning algorithm, all of the observable results. We formalize all of the randomness in HPO in terms of a random seed $r$ and a pseudo-random number generator (PRNG) $G$. Given a seed, $G$ deterministically produces a sequence of pseudo-random numbers: all numbers lie in some set $\mathcal{I}$ (typically 64-bit integers), i.e. $r \in \mathcal{I}$ and PRNG $G : \mathcal{I} \to \mathcal{I}^\infty$. With this, we can now define HPO formally:

**Definition 2.** *An HPO procedure $H$ is a tuple $(H_*, \mathcal{C}, \Lambda, \mathcal{A}, \mathcal{M}, G, X)$ where $H_*$ is a randomized algorithm, $\mathcal{C}$ is a set of allowable hyper-HPs (i.e., allowable configurations for $H_*$), $\Lambda$ is a set of allowable HPs (i.e., of HP sets $\lambda$), $\mathcal{A}$ is a training algorithm (e.g. SGD), $\mathcal{M}$ is a model (e.g. VGG16), $G$ is a PRNG, and $X$ is some dataset (usually split into train and validation sets). When run, $H_*$ takes as input a hyper-HP configuration $c \in \mathcal{C}$ and a random seed $r \in \mathcal{I}$, then proceeds to run $\mathcal{A}_\lambda$ (on $\mathcal{M}_\lambda$*

---

[2]We provide a glossary of all definitions and symbols for reference at the beginning of the Appendix.

*using $G(r)$ and data[3] from $X$) some number of times for different HPs $\lambda \in \Lambda$. Finally, $H_*$ outputs a tuple $(\lambda^*, \ell)$, where $\lambda^*$ is the HP configuration chosen by HPO and $\ell$ is the log documenting the run.*

Running $H$ is a crucial part of model development. As part of an empirical, scientific procedure, we specify different training algorithms and a learning task, run potentially many HPO passes, and try to make general conclusions about overall algorithm performance. That is, we aim to develop knowledge regarding whether one of the algorithms outperforms the others. However, recent empirical findings indicate that it is actually really challenging to pick hyper-HPs that yield reliable knowledge about general algorithm performance. In fact, it is a surprisingly common occurrence to be able to draw inconsistent conclusions based on our choice of hyper-HPs [11, 16, 48, 65].

**An example illustrating the possibility of drawing inconsistent conclusions from HPO.** As a first step to studying HPO as a procedure for developing reliable knowledge, we provide an example of how being inadvertently deceived by HPO is a real problem, even in excellent research (we give an additional example in the Appendix).[4] We first reproduce Wilson et al. [72], in which the authors trained VGG16 with different optimizers on CIFAR-10 (Figure 1a). This experiment uses grid search, with a powers-of-2 grid for the learning rate $\alpha$ crossed with the default HPs for Adam. Based on the best-performing HPO per algorithm ($\alpha = 1$), it is reasonable to conclude that non-adaptive methods (e.g., SGD) perform better than adaptive ones (e.g., Adam [42]), as the non-adaptive optimizers demonstrate higher test accuracy.

However, this setting of grid search's hyper-HPs directly informs this particular conclusion; using different hyper-HPs makes it possible to conclude the opposite. Inspired by Choi et al. [11], we perform grid search over a different subspace, tuning both learning rate and Adam's $\epsilon$ parameter. Our results entail the logically opposite conclusion: Non-adaptive methods *do not* outperform adaptive ones. Rather, when choosing the HPs that maximize test accuracy, all of the optimizers essentially have equivalent performance (Figure 1b, Appendix). Notably, as we can see from the confidence intervals in Figure 1, **satisfying statistical significance is not sufficient to avoid being deceived about comparative algorithm performance** [73]. Thus, we will require additional tools aside from statistical tests to reason about this, which we discuss in Sections 4 & 5.

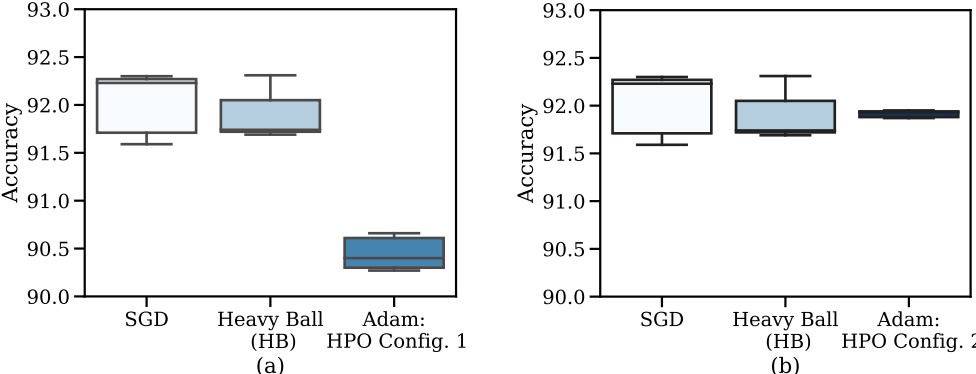

Figure 1: Demonstrating the possibility of drawing inconsistent conclusions from HPO (what we shorthand *hyperparameter deception*) when training VGG16 on CIFAR-10. Each box plot represents a log. In (a), we replicate Wilson et al. [72] and show the best-performing results: One can reasonably conclude that Adam under-performs non-adaptive methods. In (b), we change the HPO search space for Adam, and similarly show the best-performing results: In contradiction, one can reasonably conclude that Adam performs just as well as non-adaptive methods in terms of test accuracy.

This example is not exceptional, or even particularly remarkable, in terms of illustrating the hyperparameter deception problem. We simply chose it for convenience: The experiment does not require highly-specialized ML sub-domain expertise to understand it, and it is arguably broadly familiar, as it very well-cited [72]. However, we emphasize that hyperparameter deception is rather common. Additional examples can be found in numerous empirical studies across ML sub-fields [7, 11, 16, 49, 50, 53, 60, 65] (Appendix). This work shows that reported results tend to

---

[3]Definition 2 does not preclude cross-validation, as this can be part of $H_*$. The input dataset $X$ can be split in various ways, as a function of the random seed $r$.

[4]All code can be found at `https://github.com/pasta41/deception`.

be impressive for the tested hyper-HP configurations, but that modifying HPO can lead to vastly different performance outcomes that entail contradictory conclusions. More generally, it is possible to develop results that are wrong about performance, or else correct about performance but for the wrong reasons (e.g., by picking "lucky" hyperparameters). Neither of these outcomes constitutes reliable knowledge [27, 47]. As scientists, this is disheartening. We want to have confidence in the conclusions we draw from our experiments. We want to trust that we are deriving reliable knowledge about algorithm performance. **In the sections that follow, our aim is to study HPO in this reliable-knowledge sense: We want to develop ways to reason rigorously and confidently about how we derive knowledge from empirical investigations involving HPO.**

## 3 Epistemic Hyperparameter Optimization

Our discussion in Section 2 shows that applying standard HPO methodologies can be deceptive: Our beliefs about algorithm performance can be controlled by happenstance, wishful thinking, or, even worse, potentially by an adversary trying to trick us with a tampered set of HPO logs. This leaves us in a position where the "knowledge" we derived may not be knowledge at all—since we could have easily (had circumstances been different) concluded the opposite. To address this, we propose that the process of drawing conclusions using HPO should itself be an object of study. We formalize this reasoning process, which we call *epistemic hyperparameter optimization* (EHPO), and we provide an intuition for how EHPO can help us think about the hyperparameter deception problem.

**Definition 3.** *An **epistemic hyperparameter optimization procedure (EHPO)** is a tuple $(\mathcal{H}, \mathcal{F})$ where $\mathcal{H}$ is a set of HPO procedures $H$ (Definition 2) and $\mathcal{F}$ is a function that maps a set of HPO logs $\mathcal{L}$ (Definition 1) to a set of logical formulas $\mathcal{P}$, i.e. $\mathcal{F}(\mathcal{L}) = \mathcal{P}$. An execution of EHPO involves running each $H \in \mathcal{H}$ some number of times (each run produces a log $\ell$), and then evaluating $\mathcal{F}$ on the logs $\mathcal{L}$ produced in order to output the conclusions $\mathcal{F}(\mathcal{L})$ we draw from all of the HPO runs.*

In practice, it is common to run EHPO for two training algorithms, $\mathcal{J}$ and $\mathcal{K}$, and to compare their performance to conclude which is better-suited for the task at hand. $\mathcal{H}$ contains at least one HPO that runs $\mathcal{J}$ and at least one HPO that runs $\mathcal{K}$. The possible conclusions in output $\mathcal{P}$ include $p =$ "$\mathcal{J}$ performs better than $\mathcal{K}$", and $\neg p =$ "$\mathcal{J}$ does not perform better than $\mathcal{K}$". Intuitively, EHPO is deceptive whenever it could produce $p$ and also could (if configured differently or due to randomness) produce $\neg p$. That is, we can be deceived if the EHPO procedure we use to derive knowledge about algorithm performance could entail logically inconsistent results.

Our example in Section 2 is deceptive because using different hyper-HP-configured grid searches for $\mathcal{H}$ could produce contradictory conclusions. We ran two variants of EHPO $(\mathcal{H}, \mathcal{F})$: The first replicated Wilson et al. [72]'s original $\mathcal{H}$ of 3 grid-searches on SGD, HB, and Adam (Figure 1a), and the second used 3 grid-searches with a modified grid search for Adam that also tuned $\epsilon$ (Figure 1b). Each EHPO produced a $\mathcal{L}$ with 3 logs. For both, to draw conclusions $\mathcal{F}$ picks the best-performing HP-config per $\mathcal{A}$ and maps them to formulas including "SGD outperforms Adam." From the 3 logs in Figure 1a, we conclude $p$: "Non-adaptive optimizers outperform adaptive ones"; from the 3 logs in Figure 1b, we conclude $\neg p$: "Non-adaptive methods do not outperform adaptive ones." How can we formally reason about EHPO to avoid this possibility of drawing inconsistent conclusions—to guard against deceiving ourselves about algorithm performance when running EHPO?

**Framing an adversary who can deceive us.** To begin answering this question, we take inspiration from Descartes' deceptive demon thought experiment (Appendix). We frame the problem in terms of a powerful adversary trying to deceive us—one that can cause us to doubt ourselves and our conclusions. Notably, the demon is not a real adversary; rather, it models a worst-case setting of configurations and randomness that are usually set arbitrarily or by happenstance in EHPO.

Imagine an evil demon who is trying to deceive us about the relative performance of different algorithms via running EHPO. At any time, the demon maintains a set $\mathcal{L}$ of HPO logs, which it can modify either by running an HPO $H \in \mathcal{H}$ with whatever hyper-HPs $c \in \mathcal{C}$ and seed $r \in \mathcal{I}$ it wants (producing a new log $\ell$, which it adds to $\mathcal{L}$) or by erasing some of the logs in its set. Eventually, it stops and presents us with $\mathcal{L}$, from which we will draw some conclusions using $\mathcal{F}$, i.e. $\mathcal{F}(\mathcal{L})$. The demon's EHPO could deceive us via the conclusions we draw from the set of logs it produces. For example, $\mathcal{L}$ may lead us to conclude that one algorithm performs better than another, when in fact picking a different set of hyper-HPs could have generated logs that would lead us to conclude differently. We want to be sure that we will not be deceived by any logs the demon ***could*** produce. Of

course, this intuitive definition is lacking: It is not clear what is meant by ***could***. Our contribution in the sections that follow is to pin down a formal, reasonable definition of ***could*** in this context, so that we can suggest an EHPO procedure that can defend against such a maximally powerful adversary. We intentionally imagine such a powerful adversary because, if we can defend against it, then we will also be defended against weaker or accidental deception.

# 4    A Logic for Reasoning about EHPO

The informal notion of ***could*** established above encompasses numerous sources of uncertainty. There is the time to run EHPO and the choices of random seed, algorithms to compare, HPO procedures, hyper-HPs, and learning task. Then, once we have completed EHPO and have a set of logs, we have to digest those logs into logical formulas from which we base our conclusions. This introduces more uncertainty, as we need to reason about whether we believe those conclusions or not. Our formalization needs to capture all of these sources of uncertainty, and needs to be sufficiently expressive to capture how they could combine to cause us to believe deceptive conclusions. It needs to be expansive enough to handle the common case—of a well-intentioned researcher with limited resources making potentially incorrect conclusions—and the rarer, worst case—of gaming results.

**Why not statistics?**  As the common toolkit in ML, statistics might seem like the right choice for modeling all this uncertainty. However, statistics is great for reasoning about uncertainty that is *quantifiable*. For this problem, not all of the sources of uncertainty are easily quantifiable. In particular, it is very difficult to quantify the different hyper-HP possibilities. It is not reasonable to model hyper-HP selection as a random process; we do not sample from a distribution and, even if we wanted to, it is not clear how we would pick the distribution from which to sample. Moreover, as we saw in our example in Section 2, testing for statistical significance is not sufficient to prevent deception. While the results under consideration may be statistically significant, they can still fail to prevent the possibility of yielding inconsistent conclusions. For this reason, when it comes to deception, statistical significance can even give us false confidence in the conclusions we draw.

**Why modal logic?** Modal logic is the standard mathematical tool for formalizing reasoning about uncertainty [10, 18]—for formalizing the thus far informal notion of what the demon ***could*** bring about running EHPO. It is meant precisely for dealing with different types of uncertainty, particularly uncertainty that is difficult to quantify, and has been successfully employed for decades in AI [6, 30, 31], programming languages [13, 44, 58], and distributed systems [21, 32, 55]. In each of these computer science fields, modal logic's flexible semantics has been indispensable for writing proofs about higher-level specifications with multiple sources of not-precisely-quantifiable, lower-level uncertainty. For example, in distributed computing, it lets us write proofs about overall system correctness, abstracting away from the specific non-determinism introduced by each lower-level computing process [21]. Analogously, modal logic can capture the uncertainty in EHPO without being prescriptive about particular hyper-HP choices. Our notion of correctness, which we want to reason about and guarantee, is not being deceived. Therefore, while modal logic may be an atypical choice for ML, it comes with a huge payoff. By constructing the right semantics, we can capture all the sources of uncertainty described above and we can write simple proofs about whether we can be deceived by the EHPO we run. In Section 5, it is this formalization that ultimately enables us to naturally suggest a defense against being deceived.

## 4.1    Introducing our logic: syntax and semantics overview

Modal logic inherits the tools of more-familiar propositional logic and adds two operators: $\Diamond$ to represent *possibility* and $\Box$ to represent *necessity*. These operators enable reasoning about *possible worlds*—a semantics for representing how the world *is* or *could be*, making modal logic the natural choice to express the "could" intuition from Section 3. The well-formed formulas $\phi$ of modal logic are given recursively in Backus-Naur form, where $P$ is any atomic proposition:

$$\phi := P \mid \neg\phi \mid \phi \wedge \phi \mid \Diamond\phi$$

$\Diamond p$ reads, "It is possible that $p$."; $p$ is true at *some* possible world, which we *could* reach (Appendix). Note that $\Box$ is syntactic sugar, with $\Box p \equiv \neg\Diamond\neg p$. Similarly, "or" has $p \vee q \equiv \neg(\neg p \wedge \neg q)$ and "implies" has $p \rightarrow q \equiv \neg p \vee q$. The axioms of modal logic are as follows:

$$\vdash Q \rightarrow \Box Q \qquad \textit{(necessitation)}. \qquad \Box(Q \rightarrow R) \rightarrow (\Box Q \rightarrow \Box R) \qquad \textit{(distribution)}.$$

where $Q$ and $R$ are any formula, and $\vdash Q$ means $Q$ is a theorem of propositional logic. We can now provide the syntax and an intuitive notion of the semantics of our logic for reasoning about deception.

**Syntax.** Our logic requires an extension of standard modal logic. We need *two* modal operators to reckon with two overarching modalities: the possible results of the demon running EHPO ($\Diamond_t$) and our beliefs about conclusions from those results ($\mathcal{B}$). Combining these modalities yields well-formed formulas $\psi$ where, for any atomic proposition $P$ and any positive real $t$,

$$\psi := P \mid \neg\psi \mid \psi \wedge \psi \mid \Diamond_t\psi \mid \mathcal{B}\psi$$

Note the EHPO modal operator here is *indexed*: $\Diamond_t$ captures "how possible" ($\Diamond$) something is, quantified by the compute capabilities of the demon ($t$) [6, 18, 33].

**Semantics intuition.** We suppose that an EHPO user has in mind some atomic propositions (propositions of the background logic unrelated to possibility or belief, such as "the best-performing log for $\mathcal{J}$ has lower loss than the best-performing log for $\mathcal{K}$") with semantics that are already defined. $\wedge$ and $\neg$ inherit their semantics from ordinary propositional logic, which can combine propositions to form formulas. A set of EHPO logs $\mathcal{L}$ (Definition 1) can be digested into such logical formulas. That is, we define our semantics using logs $\mathcal{L}$ as models over formulas $p$: $\mathcal{L} \models p$, which reads "$\mathcal{L}$ models $p$", means that $p$ is true for the set of logs $\mathcal{L}$. We will extend this intuition to give semantics for possibility $\Diamond_t$ (Section 4.2) and belief $\mathcal{B}$ (Section 4.3), culminating in a tool that lets us reason about whether or not EHPO can deceive us by possibly yielding inconsistent conclusions (Section 4.4).

**Using our concrete example to ground us.** To clarify our presentation below, we will map our semantics to the example from Section 2, providing an informal intuition before formal definitions.

## 4.2 Expressing the possible outcomes of EHPO using $\Diamond_t$

Our formalization for possible EHPO is based on the demon of Section 3. Recall, the demon models a worst-case scenario. In practice, we deal with the easier case of well-intentioned ML researchers. The notion of possibility we define here gives limits on what possible world a demon *with bounded EHPO time* could reliably bring about. We first define a *strategy* the demon can execute for EHPO:

**Definition 4.** *A randomized **strategy** $\sigma$ is a function that specifies which action the demon will take. Given $\mathcal{L}$, its current set of logs, $\sigma(\mathcal{L})$ gives a distribution over concrete actions, where each action is either 1) running a new $H$ with its choice of hyper-HPs $c$ and seed $r$ 2) erasing some logs, or 3) returning. We let $\Sigma$ denote the set of all such strategies.*

The demon we model controls the hyper-HPs $c$ and the random seed $r$, but importantly does *not* fully control the PRNG $G$. From the adversary's perspective, for a strategy $\sigma$ to be reliable it must succeed regardless of the specific $G$. Informally, the demon cannot hack the PRNG.[5]

**Informally**, we now want to *execute a strategy* to bring about a particular outcome $p$. In Section 2, our good-faith strategy was simple: We ran each $H$ with its own hyper-HPs and random seed, then returned. The demon is trickier: It is adopting a strategy to try to bring about a deceptive outcome. **Formally**, we model the demon executing strategy $\sigma$ on logs $\mathcal{L}$ with a PRNG unknown to the demon as follows. Let $\mathcal{G}$ denote the distribution over PRNGs $G : \mathcal{I} \to \mathcal{I}^\infty$, in which all number sequence elements are drawn independently and uniformly from $\mathcal{I}$ (recall, $\mathcal{I}$ is typically the 64-bit integers). First, draw $G$ from $\mathcal{G}$, conditioned on $G$ being consistent with all the runs in $\mathcal{L}$.[6] The demon then performs a random action drawn from $\sigma(\mathcal{L})$, using $G$ as the PRNG when running a new HPO $H$, and continues—updating the working set of logs $\mathcal{L}$ as it goes—until the "return" action is chosen.

Using this process, we define what outcomes $p$ the demon can reliably bring about (i.e., what is possible, $\Diamond$) in the EHPO output logs $\mathcal{L}$ by running this random strategy $\sigma$ in bounded time $t$. **Informally**, $\Diamond_t p$ means that an adversary could adopt a strategy $\sigma$ that is guaranteed to cause the desired outcome $p$ to be the case while taking time at most $t$ in expectation. In Section 2, where $p$ is "Non-adaptive methods outperform adaptive ones", Figure 1a shows $\Diamond_t p$. **Formally**,

**Definition 5.** *Let $\sigma[\mathcal{L}]$ denote the logs output from executing strategy $\sigma$ on logs $\mathcal{L}$, and let $\tau_\sigma(\mathcal{L})$ denote the total time spent during execution. $\tau_\sigma(\mathcal{L})$ is equivalent to the sum of the times $T$ it took*

---

[5]We do not consider adversaries that can directly control how data is ordered and submitted to the algorithms under evaluation. This distinction shows that our logical construction non-trivial: We are able to defend against strong adversaries that can game the output of EHPO, which is separate from cheating by hacking the PRNG.

[6]i.e., All random events recorded in $\mathcal{L}$ should agree with the corresponding random numbers produced by $G$.

*each HPO procedure $H \in \mathcal{H}$ executed in strategy $\sigma$ to run. Note that both $\sigma[\mathcal{L}]$ and $\tau_\sigma(\mathcal{L})$ are random variables, as a function of the randomness of selecting $G$ and the actions sampled from $\sigma(\mathcal{L})$. For any formula $p$ and any $t \in \mathbb{R}_{>0}$, we say $\mathcal{L} \models \Diamond_t p$, i.e. "$\mathcal{L}$ models that it is possible $p$ in time $t$," if*

$$there\ exists\ a\ strategy\ \sigma \in \Sigma,\ such\ that\quad \mathbb{P}(\sigma[\mathcal{L}] \models p) = 1\ \ and\ \ \mathbb{E}[\tau_\sigma(\mathcal{L})] \leq t.$$

We will usually choose $t$ to be an upper bound on what is considered a reasonable amount of time to run EHPO. It does not make sense for $t$ to be unbounded, since this corresponds to the unrealistic setting of having infinite compute time to perform HPO runs. We model our budget in terms of time; however, we could use this setup to reason about other monotonically increasing resource costs, such as energy usage. Our indexed modal logic inherits many axioms of modal logic, with indexes added (Appendix), e.g.:

$$\vdash (p \rightarrow q) \rightarrow (\Diamond_t p \rightarrow \Diamond_t q) \quad \textit{(necess. + distribution)} \qquad p \rightarrow \Diamond_t p \quad \textit{(reflexivity)}$$
$$\Diamond_t \Diamond_s p \rightarrow \Diamond_{t+s} p \qquad\qquad \textit{(transitivity)} \quad \Diamond_s \Box_t p \rightarrow \Box_t p \quad \textit{(symmetry)}$$
$$\Diamond_t (p \wedge q) \rightarrow (\Diamond_t p \wedge \Diamond_t q) \qquad\qquad \textit{(dist. over $\wedge$)},$$

**To summarize**: The demon knows all possible hyper-HPs; it can pick whichever ones it wants to run EHPO within a bounded time budget $t$ to realize the outcome $p$ it wants. That is, if with some probability the demon can deceive us in some amount of time, then the demon can reliably deceive us with any larger time budget: If the demon fails to produce a deceptive result, it can use the strategy of just re-running until it yields the result it desires. Since $\Diamond_t$ models the worst-case all-powerful demon, it can also model any weaker EHPO user with time budget $t$.

### 4.3 Expressing how we draw conclusions using $\mathcal{B}$

We employ the modal operator $\mathcal{B}$ from the logic of belief[7] to model ourselves as an observer who believes in the truth of the conclusions drawn from running EHPO. $\mathcal{B}p$ reads "It is concluded that $p$." For example, when comparing the performance of two algorithms for a task, $p$ could be "$\mathcal{J}$ is better than $\mathcal{K}$" and thus $\mathcal{B}p$ would be understood as, "It is concluded that $\mathcal{J}$ is better than $\mathcal{K}$."

We model ourselves as a consistent *Type 1* reasoner [67]. **Informally**, this means we believe all propositional tautologies (necessitation), our belief distributes over implication (distribution), and we do not derive contradictions (consistency). We do not require completeness: We allow the possibility of not concluding anything about $p$ (i.e., neither $\mathcal{B}p$ nor $\mathcal{B}\neg p$). **Formally**, for any formulas $p$ and $q$,

$$\vdash p \rightarrow \mathcal{B}p \quad \textit{(necess.)}; \quad \mathcal{B}(p \rightarrow q) \rightarrow (\mathcal{B}p \rightarrow \mathcal{B}q) \quad \textit{(dist.)}; \quad \neg(\mathcal{B}p \wedge \mathcal{B}\neg p) \quad \textit{(consistency)}.$$

To understand our belief semantics, recall that EHPO includes a function $\mathcal{F}$, which maps a set of output logs $\mathcal{L}$ to our conclusions (i.e., $\mathcal{F}(\mathcal{L}) = \mathcal{P}$ is our set of conclusions). **Informally**, when our conclusion set $\mathcal{F}(\mathcal{L})$ contains a formula $p$, we say the set of logs $\mathcal{L}$ models our belief $\mathcal{B}$ in that formula $p$. In Section 2, the logs of Figure 1a model $\mathcal{B}p$ and the logs of Figure 1b model $\mathcal{B}\neg p$. **Formally**,

**Definition 6.** *For any formula $p$, we say $\mathcal{L} \models \mathcal{B}p$, "$\mathcal{L}$ models our belief in $p$", if $p \in \mathcal{F}(\mathcal{L})$.*

Note we constrain what $\mathcal{F}$ can output. For a reasonable notion of belief, $\mathcal{F}$ must model the consistent *Type 1* reasoner axioms above. Otherwise, deception aside, $\mathcal{F}$ is an unreasonable way to draw conclusions, since it is not even compatible with our belief logic.

### 4.4 Expressing hyperparameter deception

So far we have defined the semantics of our two separate modal operators, $\Diamond_t$ and $\mathcal{B}$. We now begin to reveal the benefit of using modal logic for our formalization. These operators can interact to formally express what we informally illustrated in Section 2: a notion of hyperparameter deception. It is a well-known result that we can combine modal logics [61] (Appendix). We do so to define an axiom that, if satisfied, guarantees EHPO will not be able to deceive us. For any formula $p$,

$$\neg (\Diamond_t \mathcal{B}p \ \wedge \ \Diamond_t \mathcal{B}\neg p) \qquad\qquad \textit{(t-non-deceptive)}.$$

**Informally**, our running example can be considered a proof by exhibition: It violates this axiom because Figure 1a's logs model $\Diamond_t \mathcal{B}p$ and Figure 1b's logs model $\Diamond_t \mathcal{B}\neg p$. That is, $\Diamond_t \mathcal{B}p \wedge \Diamond_t \mathcal{B}\neg p$ using grid search for this task.

---

[7]$\mathcal{B}$ is syntactically analogous to the $\Box$ modal operator in standard modal logic [35, 63, 69] (Appendix).

For the worst-case, *t-non-deceptiveness* expresses the following: **If there exists a strategy $\sigma$ by which the demon could get us to conclude $p$ in $t$ expected time, then there can exist no $t$-time strategy by which the demon could have gotten us to believe $\neg p$.** To make this concrete, suppose our $t$-non-deceptive axiom holds for an EHPO method that results in $p$. Intuitively, given a maximum reasonable time budget $t$, if there is no adversary that can consistently control whether we believe $p$ or its negation when running that EHPO, then the EHPO is defended against deception. Conversely, if an adversary could consistently control our conclusions, then the EHPO is potentially gameable. That is, if our $t$-non-deceptive axiom does not hold (i.e., we can be deceived, $\Diamond_t \mathcal{B}p \wedge \Diamond_t \mathcal{B}\neg p$), then even if we conclude $p$ after running EHPO, we cannot claim to *know* $p$. Our belief as to the truth-value of $p$ could be under the complete control of an adversary—or just a result of happenstance.

**To summarize**: An EHPO is $t$-non-deceptive if it satisfies all of the axioms above. Our example in Section 2 is $t$-deceptive because the axioms do not hold. The semantics of these axioms capture all of the possible uncertainty from the process of drawing conclusions from EHPO–and how that uncertainty can combine to cause us to believe $t$-deceptive conclusions.

## 5 Constructing Defended EHPO

Now that we have a formal notion of what it means for EHPO to be (non)-deceptive, **we can write proofs about what it means for an EHPO method to be guaranteed to be deception-free.** Importantly, these proofs will increase our confidence that our conclusions from EHPO are not due to the happenstance of picking a particular set of hyper-HPs.

To talk about defenses, we need to understand what it means to construct a "defended reasoner." In other words, for an EHPO $(\mathcal{H}, \mathcal{F})$, we need $\mathcal{F}$ to yield conclusions that we can defend against deception. Recall from Definition 6 that logs $\mathcal{L}$ model our belief in a formula $p$, i.e. $\mathcal{L} \models \mathcal{B}p \equiv p \in \mathcal{F}(\mathcal{L})$. With this in mind, we begin by supposing we have a naive EHPO $(\mathcal{H}, \mathcal{F}_n)$ featuring a naive reasoner $\mathcal{B}_n$ with corresponding belief function $\mathcal{F}_n$. We want to construct a new "defended reasoner" $\mathcal{B}_*$ that has a "skeptical" belief function $\mathcal{F}_*$. $\mathcal{F}_*$ should weaken the conclusions of $\mathcal{F}_n$ (i.e., $\mathcal{F}_*(\mathcal{L}) \subseteq \mathcal{F}_n(\mathcal{L})$ for any $\mathcal{L}$) and result in an EHPO $(\mathcal{H}, \mathcal{F}_*)$ that is guaranteed to be $t$-non-deceptive. In other words, defended reasoner $\mathcal{B}_*$ never concludes more than the naive reasoner $\mathcal{B}_n$. **Informally,** a straightforward way to do this is to have $\mathcal{B}_*$ conclude $p$ only if both the naive $\mathcal{B}_n$ would have concluded $p$, and it is impossible for an adversary to get $\mathcal{B}_n$ to conclude $\neg p$ in time $t$. **Formally,** construct $\mathcal{B}_*$ such that for any $p$, $\mathcal{B}_*p \equiv \mathcal{B}_n p \wedge \neg \Diamond_t \mathcal{B}_n \neg p$  (1).

Directly from our axioms (Section 4), we can now prove $\mathcal{B}_*$ is defended. We will suppose it is possible for $\mathcal{B}_*$ to be deceived, demonstrate a contradiction, and thereby guarantee that $\mathcal{B}_*$ is $t$-non-deceptive. Suppose $\mathcal{B}_*$ can be deceived in time $t$, i.e. $\Diamond_t \mathcal{B}_* p \wedge \Diamond_t \mathcal{B}_* \neg p$ is True. Starting with the left, $\Diamond_t \mathcal{B}_* p$ :

|  |  | Rule |
|---|---|---|
| $\Diamond_t \mathcal{B}_* p$ | $\equiv \ \Diamond_t (\mathcal{B}_n p \wedge \neg \Diamond_t \mathcal{B}_n \neg p)$ | Applying $\Diamond_t$ to the definition of $\mathcal{B}_* p$   (1) |
|  | $\rightarrow \ \Diamond_t (\neg \Diamond_t \mathcal{B}_n \neg p)$ | Reducing a conjunction to either of its terms: $(a \wedge b) \rightarrow b$ |
|  | $\rightarrow \ \neg \Diamond_t \mathcal{B}_n \neg p$ | Symmetry; dropping all but the right-most operator: $\Diamond_t(\Diamond_t a) \rightarrow \Diamond_t a$ |

We then pause to apply our axioms to the right side of the conjunction, $\Diamond_t \mathcal{B}_* \neg p$ :

|  |  | Rule |
|---|---|---|
| $\Diamond_t \mathcal{B}_* \neg p$ | $\equiv \ \Diamond_t (\mathcal{B}_n \neg p \wedge \neg \Diamond_t \mathcal{B}_n p)$ | Applying $\Diamond_t$ to the definition of $\mathcal{B}_* \neg p$   (1) |
|  | $\rightarrow \ \Diamond_t \mathcal{B}_n \neg p \wedge \Diamond_t \neg \Diamond_t \mathcal{B}_n p$ | Distributing $\Diamond_t$ over $\wedge$: $\Diamond_t(a \wedge b) \rightarrow (\Diamond_t a \wedge \Diamond_t b)$ |
|  | $\rightarrow \ \Diamond_t \mathcal{B}_n \neg p$ | Reducing a conjunction to either of its terms: $(a \wedge b) \rightarrow a$ |

We now bring both sides of the conjunction back together: $\Diamond_t \mathcal{B}_* p \wedge \Diamond_t \mathcal{B}_* \neg p \ \equiv \ \neg \Diamond_t \mathcal{B}_n \neg p \wedge \Diamond_t \mathcal{B}_n \neg p$. The right-hand side is of the form $\neg a \wedge a$, which must be False. This contradicts our initial assumption that $\mathcal{B}_*$ is $t$-deceptive (i.e., $\Diamond_t \mathcal{B}_* p \wedge \Diamond_t \mathcal{B}_* \neg p$ is True). Therefore, $\mathcal{B}_*$ is $t$-non-deceptive.

This example illustrates the power of our choice of formalization. In just a few lines of simple logic, we can validate defenses against deception. **This analysis shows that a $t$-defended reasoner $\mathcal{B}_*$ is always possible**, and it does so without needing to refer to the particular underlying semantics of an

EHPO. However, we intend this example to only be illustrative, as it may not be practical to compute $\mathcal{B}_*$ as defined in (1) if we cannot easily evaluate whether $\Diamond_t \mathcal{B}_n \neg p$. We next suggest a concrete EHPO with a defended $\mathcal{B}_*$, and show how deception can be avoided in our Section 2 example by using this EHPO instead of grid search.

**A defended random search EHPO.** Random search takes two hyper-HPs, a distribution $\mu$ over the HP space and a number of trials $K \in \mathbb{N}$ to run. HPO consists of $K$ independent trials of training algorithms $\mathcal{A}_{\lambda_1}, \mathcal{A}_{\lambda_2}, \ldots, \mathcal{A}_{\lambda_K}$, where the HPs $\lambda_k$ are independently drawn from $\mu$, taking expected time proportional to $K$. When drawing conclusions, we usually look at the "best" run for each algorithm. For simplicity, we suppose there is only one algorithm, $\mathcal{A}$. We bound how much the choice of hyper-HPs can affect the HPs, and define a defended EHPO based on a variant of random search.

**Definition 7.** *Suppose that we are given a naive EHPO procedure $(\{H\}, \mathcal{F}_n)$, in which $H$ is random search and is the only HPO in our EHPO, and $\mathcal{F}_n$ is a "naive" belief function associated with a naive reasoner $\mathcal{B}_n$. For any $K, R \in \mathbb{N}$, we define the "$(K, R)$-defended" belief function $\mathcal{F}_*$ for a skeptical reasoner $\mathcal{B}_*$ as the following conclusion-drawing procedure. First, $\mathcal{F}_*$ only makes conclusion set $\mathcal{P}_*$ from a single log $\hat{\ell}$ with $K * R$ trials; otherwise, it concludes nothing, outputting $\emptyset$. Second, $\mathcal{F}_*$ splits the single $\hat{\ell}$ into $R$ logs $\ell_1, \ell_2, \ldots, \ell_R$, each containing $K$ independent-random-search trials.[8] Finally, $\mathcal{F}_*$ outputs the intersection of what the naive reasoner would have output on each log $\ell_i$,*

$$\mathcal{F}_*(\{\hat{\ell}\}) = \mathcal{P}_* \equiv \mathcal{F}_n(\{\ell_1\}) \cap \mathcal{F}_n(\{\ell_2\}) \cap \cdots \cap \mathcal{F}_n(\{\ell_R\}).$$

*Equivalently, $\{\hat{\ell}\} \models \mathcal{B}_* p$ only if $\{\ell_i\} \models \mathcal{B}_n p$ for all $i$.*

**Informally**, to draw a conclusion using this EHPO, $\mathcal{B}_*$ splits a random-search-trial log of size $K * R$ into $R$ groups of $K$-trial logs, passing each $K$-trial log to one of an ensemble of $R$ naive reasoners $\mathcal{B}_n$. $\mathcal{B}_*$ only concludes $p$ if all $R$ naive reasoners unanimously agree on $p$. We can guarantee this EHPO to be $t$-non-deceptive by assuming a bound on how much the hyper-HPs can affect the HPs.

**Theorem 1.** *Suppose that the set of allowable hyper-HPs $\mathcal{C}$ of $H$ is constrained, such that any two allowable random-search distributions $\mu$ and $\nu$ have Renyi-$\infty$-divergence at most a constant, i.e. $D_\infty(\mu \| \nu) \leq \gamma$. The $(K, R)$-defended random-search EHPO of Definition 7 is guaranteed to be $t$-non-deceptive if we set $R \geq \sqrt{t \exp(\gamma K)/K} = O(\sqrt{t})$.*

We prove Theorem 1 in the Appendix. This result shows that our defense is *actually* a defense, and moreover it defends with a log size $K * R$—and compute requirement for good-faith EHPO—that scales sublinearly in $t$. A good-faith actor can, in sublinear-in-$t$ time, produce a log (of length $K * R$) that will allow our $t$-non-deceptive reasoner to reach conclusions. This means that we defend against adversaries with much larger compute budgets than are expected from good-faith actors.

**Validating our defense empirically and selecting hyper-HPs.** Any defense ultimately depends on the hyper-HPs it uses. Thus, we should have a reasonable belief that choosing differently would not have led an opposite conclusion. We therefore run a two-phased search [11, 34, 59], repeating our VGG16-CIFAR10 experiment from Section 2. First, we run a coarse-grained, dynamic protocol to find reasonable hyper-HPs for Adam's $\epsilon$; second, we use those hyper-HPs to run our defended random search. We start with a distribution to search over $\epsilon$, and note that the performance is best on the high end. We change the hyper-HPs, shifting the distribution until Adam's performance starts to degrade, and use the resulting hyper-HPs ($\epsilon \in [10^{10}, 10^{12}]$) to run our defense (Appendix).

We now run a modified version of our defended EHPO in Definition 7, described in Algorithm 1, with $K * R = 600$ (200 logs for each optimizer). Using a budget of $M = 10000$ iterations, we subsample $\kappa = 11$ logs and pass them to an ensemble of $\kappa$ naive reasoners $\mathcal{B}_n$. We use $\kappa$ logs, relaxing the requirement of using all $K * R$ logs in Definition 7, for efficiency. Each iteration $m$ concludes the majority conclusion of the $\kappa$-sized $\mathcal{B}_n$ ensemble. This is why we set $\kappa$ to an odd number—to avoid ties. $\mathcal{B}_*$ draws conclusions based on the results of the $M$-majority conclusions. That is, we further relax the requirements of Definition 7: Instead of requiring unanimity, $\mathcal{B}_*$ only requires agreement on the truth-value of $p$ for a fractional subset of $M$. We set this fraction using parameter $\delta \in [0, 1]$, where $\delta$ controls how skeptical our defended reasoner $\mathcal{B}_*$ is (lower $\delta$ corresponding to more skepticism). $\mathcal{B}_*$ concludes $p$ when at least $(1 - \delta)$ of our $M$ subsampled runs concluded $p$. When this threshold is not met, $\mathcal{B}_*$ remains skeptical and concludes nothing. We summarize our final results in Table 1, and

---

[8]This is not generally allowable. $\mathcal{F}_*$ can do this because random-search logs contain interchangeable trials.

provide complete results in the Appendix. Given how similar the optimizers all perform on this task (similar to Figure 1), being more skeptical increases the likelihood that we do not conclude anything.

**Algorithm 1** Defense with Random Search

---

**Require:** Set of $K * R$ random-search logs $\{\mathcal{L}_i\}_{i=1}^{KR}$, defense subsampling budget $M$, criterion constant $\delta$, subsample size $\kappa$.
1: **for** $m = 1, \cdots, M$ **do**
2:    Subsample $\kappa$ logs: $\{\mathcal{L}_i\}_{i=1}^{\kappa} \sim \{\mathcal{L}_i\}_{i=1}^{KR}$.
3:    Obtain conclusions $\{\mathcal{P}_i\}_{i=1}^{\kappa}$ from $\{\mathcal{L}_i\}_{i=1}^{\kappa}$.
4:    Obtain output conclusion for $m$:
        $\mathcal{P}^{(m)} \leftarrow \text{Majority}(\{\mathcal{P}_i\}_{i=1}^{\kappa})$
5: **end for**
6: **if** $\exists p$ s.t. $\geq (1 - \delta)M$ of $\{\mathcal{P}^{(m)}\}_{i=1}^{M}$ conclude $p$ **then**
7:    Conclude $p$.
8: **else**
9:    Conclude nothing.
10: **end if**

Table 1: Results from repeating our Section 2 experiment, using Algorithm 1 instead of grid search. $p$ = "Non-adaptive optimizers (SGD and HB) perform better than the adaptive optimizer Adam".

|  | $p$ | $\neg p$ | $1 - \delta$ | Conclude |
|---|---|---|---|---|
| SGD vs. Adam | 0.213 | 0.788 | 0.75 | $\neg p$ |
|  |  |  | 0.8 | Nothing |
|  |  |  | 0.9 | Nothing |
| HB vs. Adam | 0.168 | 0.832 | 0.75 | $\neg p$ |
|  |  |  | 0.8 | $\neg p$ |
|  |  |  | 0.9 | Nothing |

## 6   Conclusion and Practical Takeaways

Much recent empirical work illustrates that it is easy to draw inconsistent conclusions from HPO [7, 11, 16, 49, 50, 53, 60, 65]. We call this problem *hyperparameter deception* and, to derive a defense, **argue that the process of drawing conclusions using HPO should itself be an object of study**. Taking inspiration from Descartes' demon, we formalize a logic for studying an epistemic HPO procedure. The demon can run any number of reproducible HPO passes to try to get us to believe a particular notion about algorithm performance. Our formalization enables us to not believe deceptive notions: It naturally suggests how to guarantee that an EHPO is defended against deception. We offer recommendations to avoid hyperparameter deception in practice (we expand on this in the Appendix):

- **Researchers should construct their own notion of skepticism $\mathcal{B}_*$, appropriate to their specific task.** There is no one-size-fits-all defense solution. Our results are ***broad insights*** about defended EHPO: A defended EHPO is ***always possible***, but finding an efficient one will depend on the task.

- **Researchers should make explicit how they choose hyper-HPs.** What is reasonable is ultimately a function of what the ML community accepts. Being explicit, rather than eliding hyper-HP choices, is essential for helping decide what is reasonable. As a heuristic, we recommend setting hyper-HPs such that they include HPs for which the optimizers' performance starts to degrade, as we do above.

- **Avoiding hyperparameter deception is just as important as reproducibility**. We have shown that reproducibility [7, 29, 34, 57, 64] is only part of the story for ensuring reliability. While necessary for guarding against brittle findings, it is not sufficient. We can replicate results—even statistically significant ones—that suggest conclusions that are altogether wrong.

More generally, our work is a call to researchers to reason more rigorously about their beliefs concerning algorithm performance. In relation to EHPO, this is akin to challenging researchers to reify their notion of $\mathcal{B}$—to justify their belief in their conclusions from the HPO. Such epistemic rigor concerning drawing conclusions from empirical studies has a long history in more mature branches of science and computing, including evolutionary biology [28], statistics [25, 26], programming languages [54], and computer systems [23] (Appendix). We believe that applying similar rigor will contribute significantly to the ongoing effort of making ML more robust and reliable.

## Acknowledgements

A. Feder Cooper is supported by the Artificial Intelligence Policy and Practice initiative at Cornell University, the Digital Life Initiative at Cornell Tech, and the John D. and Catherine T. MacArthur Foundation. Jessica Zosa Forde is supported by ONR PERISCOPE MURI N00014-17-1- 2699. We would additionally like to thank the following individuals for feedback on earlier ideas, drafts, and

iterations of this work: Prof. Rediet Abebe, Harry Auster, Prof. Solon Barocas, Jerry Chee, Prof. Jonathan Frankle, Dr. Jack Goetz, Prof. Hoda Heidari, Kweku Kwegyir-Aggrey, Prof. Karen Levy, Prof. Helen Nissenbaum, and Prof. Gili Vidan. We also thank Meghan Witherow for her whimsical interpretation of Descartes' evil demon, which we include in the Appendix.

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
