# A Glossary

We provide a glossary of terms, definitions, and symbols that we introduce throughout the paper, streamlined in one place.

## A.1 Definitions Reference

*hyper-hyperparameters* (hyper-HPs) are HPO-procedure-input values, such as the spacing between different points in the grid for grid search and the distributions to sample from in random search.

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

## A.2 Symbols and Acronyms Reference

| Term | Explanation | Example |
|------|-------------|---------|
| HPO | Acronym for hyperparameter optimization | |
| $\mathcal{J}, \mathcal{K}$ | Used as examples of arbitrary optimizers | |
| $P$ | Arbitrary atomic proposition | |
| $p, q, \phi$ | Used as arbitrary or (when specified) specific logical formulas | $p$ = "Non-adaptive optimizers have higher test accuracy than adaptive optimizers." |
| HP(s) | Acronym for hyperparameter(s) | |
| $\ell \in \mathcal{L},$ | Log (Definition 1); log set | Figure 5; Log for running HPO using SGD |
| $T$ | The total time it took to run HPO to produce a log $\ell$ | |
| $\mathcal{I}$ | A set of integers | Typically the 64-bit integers |
| $r$ | Random seed; $r \in \mathcal{I}$ | |
| $G$ | Pseudo-random number generator; $G(r)$; $G : \mathcal{I} \rightarrow \mathcal{I}^\infty$ | |
| PRNG | Acronym for pseudo-random number generator; $G$ | |
| $H$ | HPO procedure (Definition 2) | SGD, VGG-16 grid search experiment |
| $H_*$ | A randomized algorithm used in $H$ | Random search |
| $c \in \mathcal{C}$ | Hyper-HP configuration; of set of allowable such configurations for $H_*$ | powers-of-2 grid spacing; configurations the demon has access to |
| $\lambda \in \Lambda$; $\lambda^*$ | HP config. used to run an HPO pass; of allowable HP configs., determined by $c$; $\lambda^*$ is the output HP config. that performs the best | $\alpha = 1$ in Wilson; allowable $\alpha$ values, e.g. $[.001, .01, 1]$ |
| $\mathcal{A}$; $\mathcal{A}_\lambda$ | Training algorithm; parameterized by HPs $\lambda$ | SGD; SGD with $\alpha = 1$ |
| $\mathcal{M}$; $\mathcal{M}_\lambda$ | Model; parameterized by HPs $\lambda$ | VGG16 |
| $X$ | A dataset | CIFAR-10 |
| $\alpha$ | Learning rate | Figure 2, $\alpha = 1$ |
| $\epsilon$ | Adam-specific HP | Figure 2, we set $\epsilon = 10^{12}$ |
| EHPO | Epistemic HPO (Definition 3) | Our defended random search in Section 5 |
| $\mathcal{H}$ | Set of HPO procedures $H$ | |
| $\mathcal{P}$ | Set of concluded logical formulas; $p \in \mathcal{P}$ | |
| $\mathcal{F}$ | A function that maps a set of HPO logs $\mathcal{L}$ to a set of logical formulas $\mathcal{P}$ | $\mathcal{F}_*$ (skeptical belief function); $\mathcal{F}_n$ (naive belief function) |
| $\square$ | Modal logic operator for "necessary" | $\square p$ reads "It is necessary that $p$ |
| $\Diamond$ | Modal logic operator for "possible" | $\Diamond p$ reads "It is possible that $p$ |
| $\vdash$ | Indicates a theorem of propositional logic | $\vdash Q \rightarrow \square Q$ (necessitation) |
| $\Diamond_t$ | EHPO modal operator (Section 4.2; Definition 5) | |
| $\mathcal{B}$ | Belief modal operator | $\mathcal{B}_*$ (skeptical belief); $\mathcal{B}_n$ (naive belief) |
| $\sigma \in \Sigma$ | A randomized strategy function that specifies EHPO actions; set of all such strategies (Section 4.2, Definition 4) | |
| $\sigma(\mathcal{L})$ | Distribution over concrete actions for log set | |
| $\sigma[\mathcal{L}]$ | The logs output from running $\sigma$ on $\mathcal{L}$ | |
| $\tau_\sigma(\mathcal{L})$ | Total time spent executing $\sigma[\mathcal{L}]$ | |
| $\models$ | Denotes "models" | $\mathcal{L} \models \Diamond_t p$: $\mathcal{L}$ model that $p$ is possible in $t$ |
| $\gamma$ | Renyi-$\infty$-divergence constant upper bound (Theorem 1) | |
| $K, R$ | Numbers of independent random search trials (Section 5) | |
| $\kappa$ | Subsampling size (Algorithm 1) | We set $\kappa = 11$ (Section 5) |
| $M$ | Subsampling budget (Algorithm 1) | We set $M = 10000$ (Section 5) |
| $\delta$ | Skeptical reasoner conclusion threshold (Algorithm 1) | See Table 1 |

# B Section 2 Appendix: Additional Notes on the Preliminaries

The code for running these experiments can be found at `https://github.com/pasta41/deception`.

## B.1 Empirical Deception Illustration using Wilson et al. [72]

### B.1.1 Why we chose Wilson et al. [72]

We elaborate on why we specifically chose Wilson et al. [72] as our running example of hyperparameter deception. There are four main reasons why we thought this was the right example to focus on for an illustration. First, the experiment involves optimizers known across ML (e.g. SGD, Adam), a model frequently used for benchmark tasks (VGG16) and a commonly-used benchmark dataset (CIFAR-10). Unlike other examples of hyperparameter deception, one does not need highly-specialized domain knowledge to understand the issue [16, 49]. Second, the paper is exceptionally well-cited and known in the literature, so many folks in the community are familiar with its results. Third, we were certain that we could demonstrate hyperparameter deception before we ran our experiments; we observe that Adam's update rule basically simulates Heavy Ball when its $\epsilon$ parameter is set high enough. So, we were confident that we could (at the very least) get Adam to perform as well as Heavy Ball via changing hyper-HPs, which would demonstrate hyperparameter deception. We then found further support for this observation in concurrent work [65], which cited earlier work [11] that also observes this. Fourth, the claim in Wilson et al. [72] is fairly broad. They make a claim about adaptive vs. non-adaptive optimizers, more generally. If the claim had been narrower – about small $\epsilon$ values for numerical stability, then perhaps hyperparameter deception would not have occurred. In general, we note that narrower claims could help avoid deception.

## B.2 Expanded empirical results

We elaborate on the results we present in Section 2.

### B.2.1 Experimental setup

We replicate and run a variant of Wilson et al. [72]'s VGG16 experiment on CIFAR-10, using SGD, Heavy Ball, and Adam as the optimizers.

We launch each run on a local machine configured with a 4-core 2.6GHz Inter (R) Xeon(R) CPU, 8GB memory and an NIVIDIA GTX 2080Ti GPU.

Following the exact configuration from Wilson et al. [72], we set the mini-batch size to be 128, the momentum constant to be 0.9 and the weight decay to be 0.0005.

The learning rate is scheduled to follow a linear rule: The learning rate is decayed by a factor of 10 every 25 epochs. The total number of epochs is set to be 250.

For the CIFAR-10 dataset, we apply random horizontal flipping and normalization. Note that Wilson et al. [72] does not apply random cropping on CIFAR-10; thus we omit this step to be consistent with their approach. We adopt the standard cross entropy loss.

For each HPO setting, we run 5 times and average the results and include error bars two standard deviations above and below the mean.

### B.2.2 Associated results and logs

In line with our notion of a *log* (Definition 1), we provide data tables (Figures 5, 6, and 7) that correspond with our results graphed in the Figures 2, 3, 4.

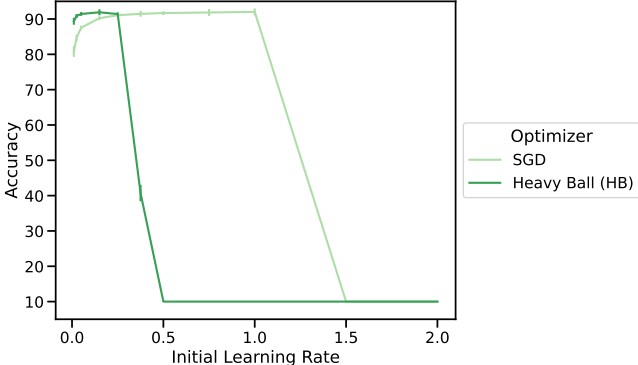

Figure 2: Full test accuracy results of VGG-16 on CIFAR-10 for SGD and Heavy Ball learning rate ($\alpha$) HPO. Error bars indicate two standard deviations above and below the mean. Each HPO setting is measured with five replicates. We achieve similar performance as Wilson et al. [72].

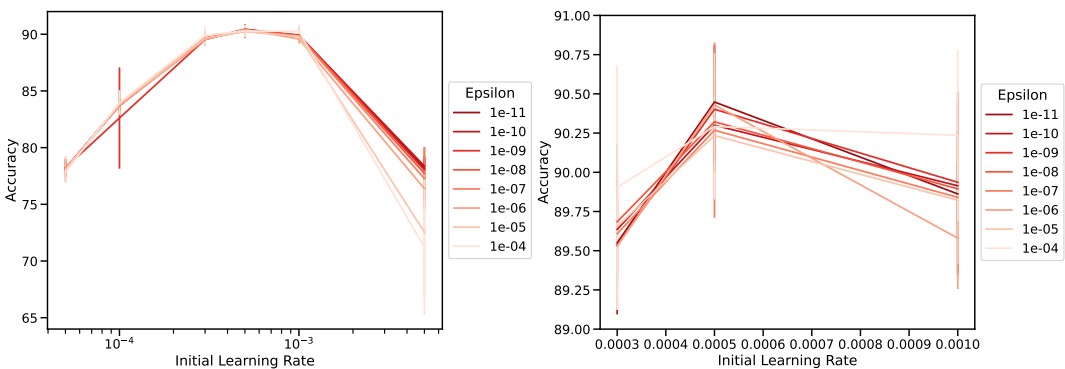

Figure 3: Tuning over learning rate for different small values of $\epsilon$. On the left, we show a wide range of learning rates tested. On the right, we zoom in on the portion of results where the best test accuracy occurs. These results reflect what Wilson et al. [72] showed, but with tuning over $\epsilon$ (small values). Each HP setting is used to train VGG-16 on CIFAR-10 five times, and the error bars represent two standard deviations above and below the mean test accuracy.

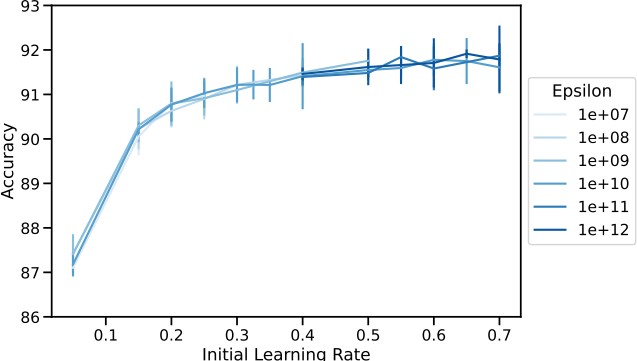

Figure 4: Results for our expanded search over large $\epsilon$ values for Adam. We show test accuracy on CIFAR-10 as a function of different learning rates $\alpha$ for the different large $\epsilon$ values. Error bars show two standard deviations above and below mean test accuracy for five replicates for each HP setting.

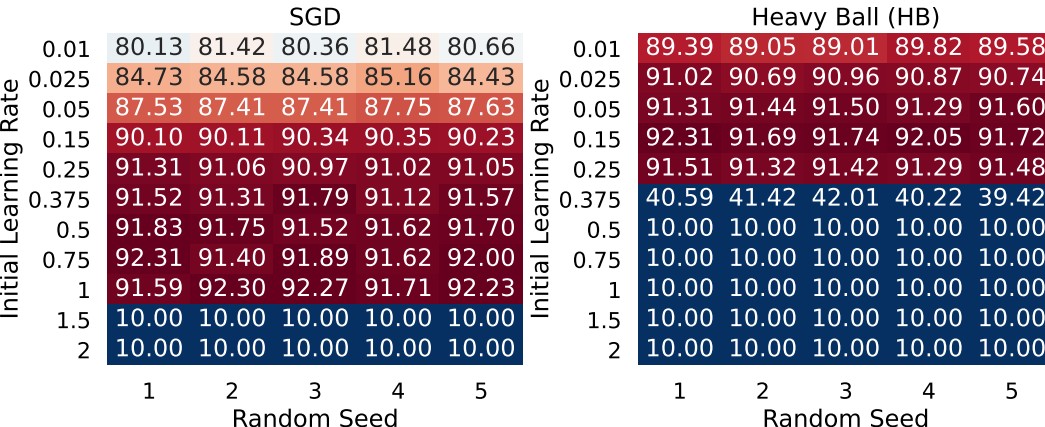

Figure 5: Heatmap logs of test accuracy of VGG-16 on CIFAR-10 for SGD and Heavy Ball for each initial learning rate and random seed. These logs correspond to the results graphed in Figure 2.

**epsilon = 1e-04**

| Initial Learning Rate | 1 | 2 | 3 | 4 | 5 |
|---|---|---|---|---|---|
| 5e-05 | 78.04 | 77.53 | 77.66 | 78.16 | 77.81 |
| 1e-04 | 83.58 | 83.87 | 83.62 | 84.85 | 83.89 |
| 3e-04 | 90.16 | 89.65 | 89.97 | 89.39 | 90.35 |
| 5e-04 | 90.23 | 90.09 | 90.22 | 90.23 | 90.69 |
| 1e-03 | 90.71 | 90.03 | 90.16 | 90.11 | 90.17 |
| 5e-03 | 73.25 | 75.23 | 70.16 | 69.55 | 67.94 |

**epsilon = 1e-05**

| Initial Learning Rate | 1 | 2 | 3 | 4 | 5 |
|---|---|---|---|---|---|
| 5e-05 | 78.99 | 78.03 | 77.64 | 77.76 | 77.91 |
| 1e-04 | 83.80 | 83.85 | 83.65 | 84.15 | 83.80 |
| 3e-04 | 89.55 | 90.08 | 89.45 | 89.73 | 89.46 |
| 5e-04 | 90.41 | 90.09 | 90.22 | 90.28 | 90.17 |
| 1e-03 | 89.60 | 89.68 | 89.85 | 89.86 | 90.12 |
| 5e-03 | 70.79 | 76.68 | 69.19 | 73.55 | 72.53 |

**epsilon = 1e-06**

| Initial Learning Rate | 1 | 2 | 3 | 4 | 5 |
|---|---|---|---|---|---|
| 5e-05 | 78.16 | 78.42 | 77.97 | 78.53 | 77.64 |
| 1e-04 | 83.77 | 83.53 | 83.66 | 83.66 | 83.99 |
| 3e-04 | 89.23 | 89.49 | 89.69 | 89.55 | 89.68 |
| 5e-04 | 90.49 | 90.51 | 90.45 | 90.24 | 90.46 |
| 1e-03 | 89.46 | 89.50 | 89.44 | 89.79 | 89.71 |
| 5e-03 | 76.97 | 77.11 | 75.14 | 75.85 | 76.89 |

**epsilon = 1e-07**

| Initial Learning Rate | 1 | 2 | 3 | 4 | 5 |
|---|---|---|---|---|---|
| 5e-05 | 78.64 | 77.88 | 77.55 | 78.13 | 78.06 |
| 1e-04 | 83.92 | 83.76 | 83.26 | 83.81 | 83.93 |
| 3e-04 | 89.59 | 89.56 | 89.59 | 89.51 | 89.79 |
| 5e-04 | 90.42 | 89.81 | 90.53 | 90.33 | 90.25 |
| 1e-03 | 90.03 | 89.67 | 90.10 | 89.79 | 89.61 |
| 5e-03 | 77.46 | 78.49 | 76.98 | 75.16 | 78.26 |

**epsilon = 1e-08**

| Initial Learning Rate | 1 | 2 | 3 | 4 | 5 |
|---|---|---|---|---|---|
| 5e-05 | 78.27 | 78.10 | 77.89 | 78.55 | 77.94 |
| 1e-04 | 83.51 | 83.75 | 83.89 | 83.80 | 84.03 |
| 3e-04 | 89.85 | 89.46 | 89.78 | 89.50 | 89.83 |
| 5e-04 | 90.24 | 90.33 | 90.34 | 90.37 | 90.33 |
| 1e-03 | 90.01 | 89.91 | 90.27 | 89.55 | 89.73 |
| 5e-03 | 78.21 | 78.19 | 75.73 | 78.37 | 78.14 |

**epsilon = 1e-09**

| Initial Learning Rate | 1 | 2 | 3 | 4 | 5 |
|---|---|---|---|---|---|
| 5e-05 | 78.31 | 78.51 | 77.66 | 78.52 | 78.07 |
| 1e-04 | 78.73 | 84.11 | 83.35 | 83.62 | 83.25 |
| 3e-04 | 89.42 | 89.40 | 89.68 | 89.66 | 89.52 |
| 5e-04 | 90.56 | 90.14 | 90.55 | 90.39 | 90.37 |
| 1e-03 | 90.25 | 89.62 | 89.73 | 89.86 | 90.22 |
| 5e-03 | 78.62 | 78.06 | 77.40 | 77.84 | 78.11 |

**epsilon = 1e-10**

| Initial Learning Rate | 1 | 2 | 3 | 4 | 5 |
|---|---|---|---|---|---|
| 5e-05 | 78.63 | 77.89 | 78.13 | 78.08 | 78.02 |
| 1e-04 | 83.51 | 83.58 | 83.86 | 83.67 | 83.88 |
| 3e-04 | 89.93 | 89.45 | 89.30 | 89.87 | 89.63 |
| 5e-04 | 90.25 | 90.10 | 90.49 | 90.38 | 90.28 |
| 1e-03 | 90.03 | 89.68 | 90.12 | 89.88 | 89.86 |
| 5e-03 | 78.20 | 78.86 | 78.13 | 78.76 | 77.92 |

**epsilon = 1e-11**

| Initial Learning Rate | 1 | 2 | 3 | 4 | 5 |
|---|---|---|---|---|---|
| 5e-05 | 78.18 | 78.09 | 78.08 | 77.99 | 78.15 |
| 1e-04 | 84.15 | 83.73 | 83.88 | 83.67 | 83.65 |
| 3e-04 | 89.62 | 89.64 | 89.66 | 89.19 | 89.64 |
| 5e-04 | 90.40 | 90.27 | 90.66 | 90.61 | 90.30 |
| 1e-03 | 89.96 | 89.90 | 89.93 | 89.75 | 89.77 |
| 5e-03 | 78.73 | 78.18 | 78.39 | 77.72 | 77.91 |

Figure 6: Heatmap logs of test accuracy of VGG-16 on CIFAR-10 for Adam for each initial learning rate and random seed for different small values of Adam's $\epsilon$ HP. These results correspond to those graphed in Figure 3.

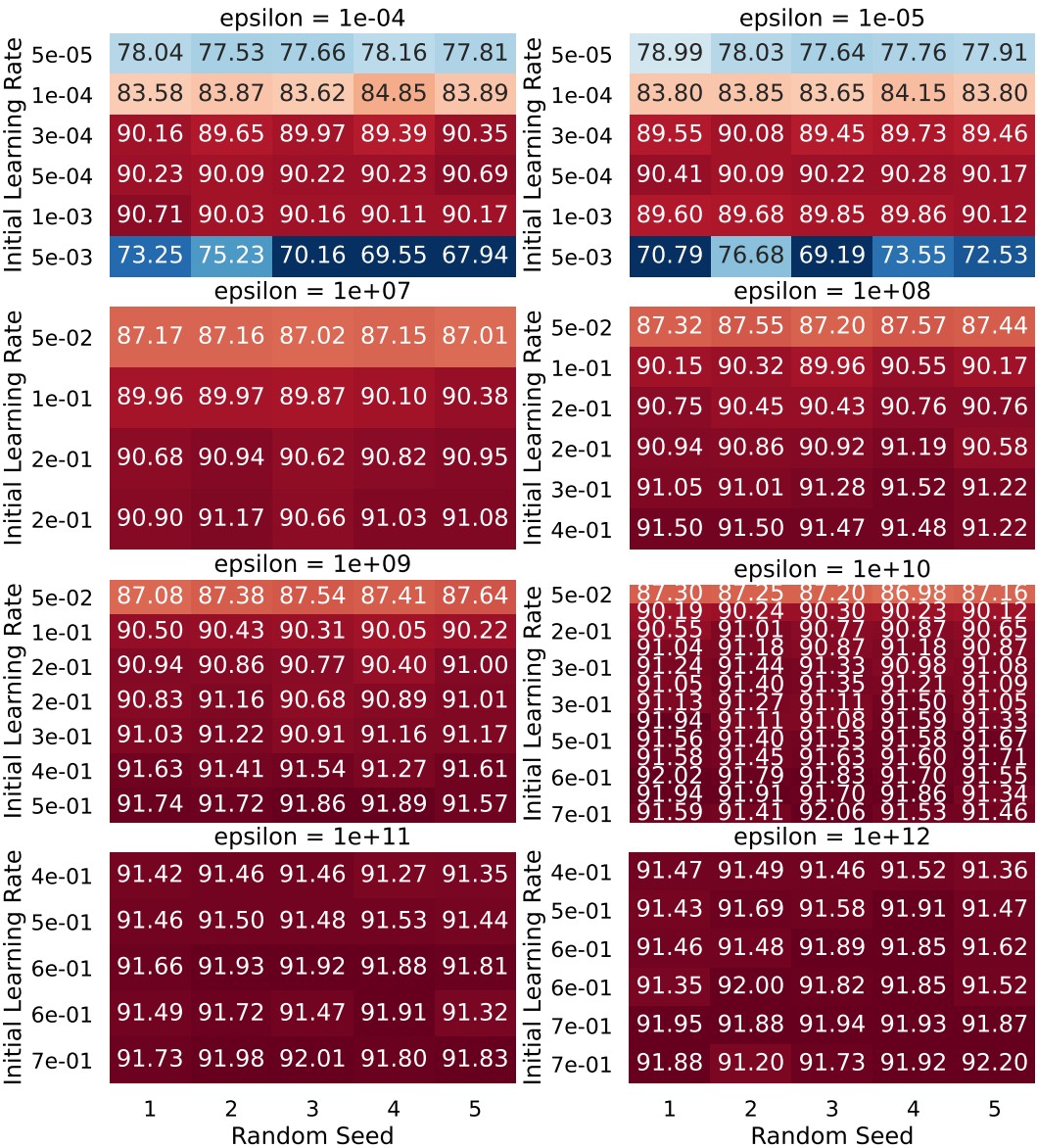

Figure 7: Heatmap logs of test accuracy of VGG-16 on CIFAR-10 for Adam for each initial learning rate and random seed for different values of Adam's $\epsilon$ using our expanded search space. These logs reflect the results graphed in Figure 4.

## B.3 Empirical Deception Illustration using Merity et al. [51]

In addition to the computer vision experiments of Wilson et al. [72], we also show a separate line of experiments from NLP: training an LSTM on Wikitext-2 using Nesterov and Heavy Ball as the optimizers. We illustrate deception (i.e., the possibility of drawing inconsistent conclusions) using two different sets of hyper-HPs to configure HPO grids for tuning the learning rate. We run ten replicates for each optimizer / grid combination (a total of 40 runs). We run these experiments using the same hardware as described in Appendix B.2.1.

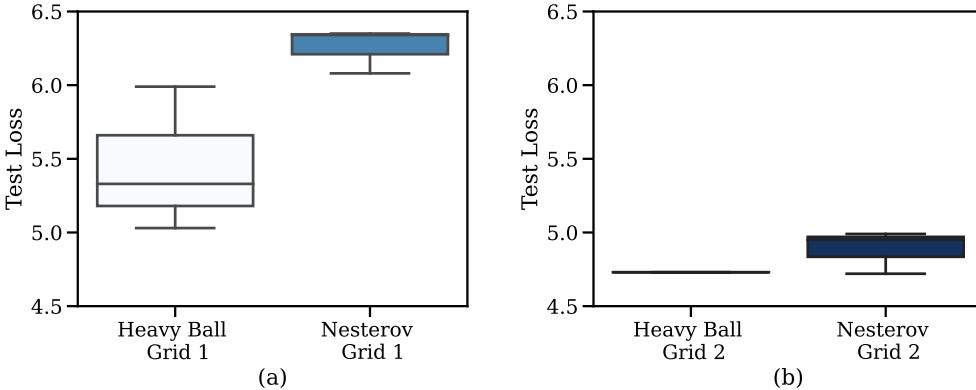

Figure 8: Demonstrating the possibility of drawing inconsistent conclusions from HPO (what we shorthand *hyperparameter deception*) LSTM on Wikitext-2 using Nesterov and Heavy Ball as the optimizers. Each box plot represents a log. In (a), we use the grid $\alpha = 1, 5, 10, 15, 20, 25, 30, 35, 40$, from which we can reasonably conclude that Nesterov outperforms HB. In (b), we use the grid $\alpha = 10, 20, 30, 40$, from which we can reasonably conclude that HB outperforms Nesterov.

# C Section 3 Appendix: Epistemic Hyperparameter Optimization

## C.1 Additional concrete interpretations of EHPO

For concision, in the main text we focus on examples of EHPO procedures that compare the performance of different optimizers. However, it is worth noting that our definition of EHPO (Definition 3) is more expansive than this setting. For example, it is possible to run EHPO to compare different models (perhaps, though not necessarily, keeping the optimizer fixed), to draw conclusions about the relative performance of different models on different learning tasks.

## C.2 Descartes' Evil Demon Thought Experiment

Our formalization was inspired by Descartes' evil genius/demon thought experiment. This experiment more generally relates to his use of systematic doubt in *The Meditations* more broadly. It is this doubt/skepticism (and its relationship to possibility) that we find useful for the framing of an imaginary, worst-case adversary. In particular, we draw on the following quote, from which we came up with the term *hyperparameter deception*:

> *I will suppose...an evil genius, supremely powerful and clever, who has directed his entire effort at deceiving me. I will regard the heavens, the air, the earth, colors, shapes, sounds, and all external things as nothing but the bedeviling hoaxes of my dreams, with which he lays snares for my credulity...even if it is not within my power to know anything true, it certainly is within my power to take care resolutely to withhold my assent to what is false, lest this deceiver, however powerful, however clever he may be, have any effect on me.* – Descartes

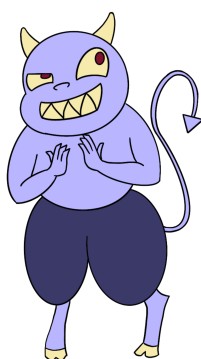

Figure 9: How the authors imagine the EHPO-running demon

For more on the long (and rich) history of the use of imaginary demons and devils as adversaries—notably a different conception of an adversary than the potential real threats posed in computer security research—we refer the reader to Canales [9].

## D Section 4 Appendix: Modal Logic Formalization

### D.1 Further Background on Modal Logic

We first provide the necessary background on modal logic, which will inform the proofs in this appendix (Appendix D.1.1). We then describe our possibility logic—a logic for representing the possible results of the evil demon running EHPO—and prove that it is a valid modal logic (Appendix D.2.1). We then present a primer on modal belief logic (Appendix D.2.4), and suggest a proof for the validity of combining our modal possibility logic with modal belief logic (Appendix D.2.5).

#### D.1.1 Axioms from Kripke Semantics

Kripke semantics in modal logic inherits all of the the axioms from propositional logic, which assigns values $T$ and $F$ to each atom $p$, and adds two operators, one for representing *necessity* ($\Box$) and one for *possibility* ($\Diamond$).

- $\Box p$ reads "It is necessary that $p$".
- $\Diamond p$ reads "It is possible that $p$".

The $\Diamond$ operator is just syntactic sugar, as it can be represented in terms of $\neg$ and $\Box$:

$$\Diamond p \equiv \neg \Box \neg p \tag{1}$$

which can be read as:

"It is possible that $p$" is equivalent to "It is not necessary that not $p$."

The complete set of rules is as follows:

- Every atom $p$ is a sentence.
- If $D$ is a sentence, then
  - $\neg D$ is a sentence.
  - $\Box D$ is a sentence.
  - $\Diamond D$ is a sentence.
- If $D$ and $E$ are sentences, then
  - $D \wedge E$ is a sentence.
  - $D \vee E$ is a sentence.

    – $D \rightarrow E$ is a sentence.

    – $D \leftrightarrow E$ is a sentence

- $\Box(\mathcal{D} \rightarrow \mathcal{E}) \rightarrow (\Box\mathcal{D} \rightarrow \Box\mathcal{E})$ (Distribution)

- $\mathcal{D} \rightarrow \Box\mathcal{D}$ (Necessitation)

### D.1.2 Possible Worlds Semantics

Modal logic introduces a notion of *possible worlds*. Broadly speaking, a possible world represents the state of how the world *is* or potentially *could be* [10, 24]. Informally, $\Box D$ means that $D$ is true at *every* world (Equation 2); $\Diamond D$ means that $D$ is true at *some* world (Equation 3).

Possible worlds give a different semantics from more familiar propositional logic. In the latter, we assign truth values $\{T, F\}$ to propositional variables $p \in \mathcal{P}$, from which we can construct and evaluate sentences $D \in \mathcal{D}$ in a truth table. In the former, we introduce a set of possible worlds, $\mathcal{W}$, for which each $w \in \mathcal{W}$ has own truth value for each $p$. This means that the value of each $p$ can differ across different worlds $w$. Modal logic introduces the idea of valuation function,

$$\mathcal{V} : (\mathcal{W} \times \mathcal{D}) \rightarrow \{T, F\}$$

to assign truth values to logical sentences at different worlds. This in turn allows us to express the formulas, axioms, and inference rules of propositional logic in terms of $\mathcal{V}$. For example,

$$\mathcal{V}(w, \neg D) = T \leftrightarrow \mathcal{V}(w, D) = F$$

There are other rules that each correspond to a traditional truth-table sentence evaluation, but conditioned on the world in which the evaluation occurs. We omit these for brevity and refer the reader to Chellas [10].

We do include the valuation rules for the $\Box$ and $\Diamond$ operators that modal logic introduces (Equations 2 & 3). To do so, we need to introduce one more concept: The accessibility relation, $\mathcal{R}$. $\mathcal{R}$ provides a frame of reference for one particular possible world to access other possible worlds; it is a way from moving from world to world. So, for an informal example, $\mathcal{R}w_1w_2$ means that $w_2$ is possible relative to $w_1$, i.e. we can reach $w_2$ from $w_1$. Such a relation allows for a world to be possible relative potentially to some worlds but not others. More formally,

$$R \subseteq \mathcal{W} \times \mathcal{W}$$

Overall, the important point is that we have a collection of worlds $\mathcal{W}$, an accessibility relation $\mathcal{R}$, and a valuation function $\mathcal{V}$, which together defines a Kripke model, which captures this system:

$$\mathcal{M} = \langle \mathcal{W}, \mathcal{R}, \mathcal{V} \rangle$$

Finally, we can give the valuation function rules for $\Box$ and $\Diamond$:

$$\mathcal{V}(w, \Box D) = T \leftrightarrow \forall w', (\mathcal{R}ww' \rightarrow \mathcal{V}(w', D) = T) \tag{2}$$

$$\mathcal{V}(w, \Diamond D) = T \leftrightarrow \exists w', (\mathcal{R}ww' \wedge \mathcal{V}(w', D) = T) \tag{3}$$

Informally, for $\Box D$ to be true in a world, it must be true in every possible world that is reachable by that world. For $\Diamond D$ to be true in a world, it must be true in some possible world that is reachable by that world.

### D.2 Our Multimodal Logic Formulation

#### D.2.1 A Logic for Reasoning about the Conclusion of EHPO

As in Section 4, we can define the well-formed formulas of our indexed modal logic[11] recursively in Backus-Naur form, where $t$ is any real number and $P$ is any atomic proposition

$$\kappa := P \mid \neg\kappa \mid \kappa \wedge \kappa \mid \Diamond_t \kappa \tag{4}$$

where $\kappa$ is a well-formed formula.

As we note in Section 4, where we first present this form of defining modal-logic, $\Box$ is syntactic sugar, with $\Box p \equiv \neg\Diamond\neg p$ (which remains true for our indexed modal logic). Similarly, "or" has $p \vee q \equiv \neg(\neg p \wedge \neg q)$ and "implies" has $p \to q \equiv \neg p \vee q$, which is why we do not include them for brevity in this recursive definition.

We explicitly define the relevant semantics for $\Diamond_t$ for reasoning about the demon's behavior in running EHPO. For clarity, we replicate that definition of the semantics of expressing the possible outcomes of EHPO conducted in bounded time (Definitions 4 & 5, respectively) below:

**Definition.** *A randomized **strategy** $\sigma$ is a function that specifies which action the demon will take. Given $\mathcal{L}$, its current set of logs, $\sigma(\mathcal{L})$ gives a distribution over concrete actions, where each action is either 1) running a new $H$ with its choice of hyper-HPs $c$ and seed $r$ 2) erasing some logs, or 3) returning. We let $\Sigma$ denote the set of all such strategies.*

We can now define what the demon can reliably bring about, in terms of executing a strategy in bounded time:

**Definition.** *Let $\sigma[\mathcal{L}]$ denote the logs output from executing strategy $\sigma$ on logs $\mathcal{L}$, and let $\tau_\sigma(\mathcal{L})$ denote the total time spent during execution. $\tau_\sigma(\mathcal{L})$ is equivalent to the sum of the times $T$ it took each HPO procedure $H \in \mathcal{H}$ executed in strategy $\sigma$ to run. Note that both $\sigma[\mathcal{L}]$ and $\tau_\sigma(\mathcal{L})$ are random variables, as a function of the randomness of selecting $G$ and the actions sampled from $\sigma(\mathcal{L})$. For any formula $p$ and any $t \in \mathbb{R}_{>0}$, we say $\mathcal{L} \models \Diamond_t p$, i.e. "$\mathcal{L}$ models that it is possible $p$ in time $t$," if*

$$\text{there exists a strategy } \sigma \in \Sigma, \text{ such that } \quad \mathbb{P}(\sigma[\mathcal{L}] \models p) = 1 \ \text{ and } \ \mathbb{E}[\tau_\sigma(\mathcal{L})] \leq t.$$

#### D.2.2 A Possible Worlds Interpretation

Drawing on the possible worlds semantics that modal logic provides (Section D.1.2), we can define specific possible worlds semantics for our logic for expressing the actions of the demon in EHPO from above.

**Definition 8.** *A **possible world** represents the set of logs $\mathcal{L}$ the demon has produced at time $\tau_\sigma(\mathcal{L})$ (i.e., after concluding running EHPO) and the set of formulas $\mathcal{P}$ that are modeled from $\mathcal{L}$ via $\mathcal{F}$.*

Therefore, different possible worlds represent the states that *could have existed* if the evil demon had executed different strategies (Definition 4). In other words, if it had performed EHPO with different learning algorithms, different HPO procedures, different hyper-hyperparameter settings, different amounts of time (less than the total upper bound), different learning tasks, different models, etc... to produce a different set of logs $\mathcal{L}$ and corresponding set of conclusions $\mathcal{P}$.

In this formulation, the demon has knowledge of all possible worlds; it is trying to fool us about the relative performance of algorithms by showing as an intentionally deceptive world. Informally, moving from world to world (via an accessibility relation) corresponds to the demon running more passes of HPO to produce more logs to include in $\mathcal{L}$.

#### D.2.3 Syntax and Semantics for the Logic Modeling the Demon Running the EHPO

We provide proofs and intuitions of the axioms of our EHPO logic in this section, based on a correspondence with un-indexed modal logic.

---

[11]For an example of another indexed modal logic concerning probability, please refer to Heifeitz and Mongin [33].

We remind the reader that the following are the axioms of our indexed modal logic:

$$\vdash (p \rightarrow q) \rightarrow (\Diamond_t p \rightarrow \Diamond_t q) \qquad \textit{(necessitation + distribution)}$$
$$p \rightarrow \Diamond_t p \qquad \textit{(reflexivity)}$$
$$\Diamond_t \Diamond_s p \rightarrow \Diamond_{t+s} p \qquad \textit{(transitivity)}$$
$$\Diamond_s \Box_t p \rightarrow \Box_t p \qquad \textit{(symmetry)},$$
$$\Diamond_t (p \wedge q) \rightarrow (\Diamond_t p \wedge \Diamond_t q) \qquad \textit{(distribution over } \wedge \textit{ )}$$

In short, to summarize these semantics—the demon has knowledge of all possible hyper-hyperparameters, and it can pick whichever ones it wants to run EHPO within a bounded time budget $t$ to realize the outcomes it wants: $\Diamond_t p$ means it can realize $p$.

We inherit distribution and necessitation from un-indexed modal logic; they are axiomatic based on Kripke semantics. We provide greater intuition and proofs below.

**Notes on necessitation for $\Box_t$:**

Necessitation for our indexed necessary operator can be written as follows:

$$\vdash p \rightarrow \Box_i p$$

As we note in Section 4, $\vdash$ just means here that $p$ is a theorem of propositional logic. So, if $p$ is a theorem, then so is $\Box_t p$. By theorem we just mean that $p$ is provable by our axioms (these being the only assumptions we can use); so whenever $p$ fits this definition, we can say $\Box_t p$.

For our semantics, this just means that when $p$ is a theorem, it is necessary that $p$ in time $t$.

**Distribution for $\Box_t$:**

$$\Box_t(p \rightarrow q) \rightarrow (\Box_t p \rightarrow \Box_t q)$$

We provide three ways to verify distribution over implication for $\Box_t$. From this, we will prove distribution over implication for $\Diamond_t$

A. The first follows from an argument about the semantics of possible worlds from the Kripke model of our system (Sections D.1.2 & D.2.2).

    i. It is fair to reason that distribution is self-evident given the definitions of implication ($\rightarrow$, formed from $\neg$ and $\vee$ in our syntax for well-formed formulas for our EPHO logic, given at (4) and necessity ($\Box_t$, formed from $\neg$ and $\Diamond_t$ in our syntax for well-formed formulas for our EHPO logic, given at (4)).

    ii. Similarly, we can further support this via our semantics of possible worlds.
We can understand $\Box_t p$ to mean, informally, that it an adversary does adopt a strategy $\sigma$ that is guaranteed to cause the desired conclusion $p$ to be the case while take at most time $t$ in expectation. Formally, as an "necessary" analog to the semantics of $\Diamond_t$ given in Definition 5:
For any formula $p$, we say $\mathcal{L} \models \Box_t p$ if and only if
$$\forall \sigma \in \Sigma, \ \mathbb{P}(\sigma[\mathcal{L}] \models p) = 1 \ \wedge \ \mathbb{E}[\tau_\sigma(\mathcal{L})] \leq t.$$
Given $p \rightarrow q$ is true in **all accessible worlds** (i.e, the definition of necessary), then we can say that $q$ is true in all accessible worlds whenever $p$ is true in all accessible worlds. As in i. above, this just follows / is axiomatic from the definitions of necessity and implication for Kripke semantics.

B. We can also prove distribution by contradiction.

    i. Suppose that the distribution axiom does not hold. That is, the hypothesis
$$\Box_t(p \rightarrow q)$$
    is true and the conclusion
$$\Box_t p \rightarrow \Box_t q$$
    is false.

ii. By similar reasoning, from above $\Box_t p \to \Box_t q$ being false, we can say that $\Box_t p$ is true and $\Box_t q$ is false.

iii. We can use Modal Axiom M (reflexivity, proven in the next section) to say $\Box_t p \to p$. Since $\Box_t p$ is true, we can use *modus ponens* to determine that $p$ is true.

iv. We can also say

$$\Box_t(p \to q) \to (p \to q) \qquad (\textit{By Modal Axiom M (reflexivity)})$$

v. Since we $\Box_t(p \to q)$ is true from above, we can conclude via *modus ponens* that $p \to q$ must also be true.

vi. We concluded above that $p$ is true, so we can again use *modus ponens* and the fact that $p \to q$ is true to conclude that $q$ is true.

vii. By necessitation, we can then also say $q \to \Box_t q$, and conclude that $\Box_t q$ is true. This is a contradiction, as above we said that $\Box_t$ is false.

viii. Therefore, by contradiction, $\Box_t(p \to q) \to (\Box_t p \to \Box_t q)$ is proved.

C. We can separately take an intuitionistic approach to verify the distribution axiom [1, 5]:

i. Let $b$ be an **actual proof** of $p \to q$ so that we can say $a.b$ is a proof of $\Box_t(p \to q)$.

ii. Let $d$ be an **actual proof** of $p$ so that we can say $c.d$ is a proof of $\Box_t p$.

iii. From i. and ii., $b(d)$ is an **actual proof** of $q$, i.e. $b$ (an actual proof of $p \to q$) is supplied $d$ (an actual proof of $p$), and therefore can conclude $q$ via an actual proof.

iv. From iii., we can say this results in a proof of $\Box_t q$, i.e. $e.[b(d)]$.

v. The above i.-iv. describes a function, $f : a.b \to f_{(a.b)}$. In other words, given **any proof** $a.b$ (i.e., of $\Box_t(p \to q)$) we can return function $f_{(a.b)}$, which turns **any proof** $c.d$ (i.e., of $\Box_t p$) into a proof $e.[b(d)]$ (i.e., of $\Box_t q$).

vi. $f_{(a.b)}$ is thus a proof of $\Box_t p \to \Box_t q$.

vii. From i.-vi., we gone from $a.b$ (a proof of $\Box_t(p \to q)$) to a proof of $\Box_t p \to \Box_t q$, i.e. have intuitionistically shown that $\Box_t(p \to q) \to (\Box_t p \to \Box_t q)$

## Distribution and $\Diamond_t$:

We provide the following axiom in our logic:

$$\vdash (p \to q) \to (\Diamond_t p \to \Diamond_t q) \qquad (\textit{necessitation and distribution})$$

and we now demonstrate it to be valid.

$$
\begin{aligned}
\vdash (p \to q) &\to \Box_t(p \to q) & (\textit{necessitation}) \\
&\to \Box_t(\neg q \to \neg p) & (\textit{modus tollens}) \\
&\to (\Box_t \neg q \to \Box_t \neg p) & (\textit{distribution}) \\
&\to (\neg \Box_t \neg p \to \neg \Box_t \neg q) & (\textit{modus tollens}) \\
&\to (\Diamond_t p \to \Diamond_t q) & (\Diamond_t a \equiv \neg \Box_t \neg a)
\end{aligned}
$$

This concludes our proof, for how the axioms are jointly stated.

Further, we could also say

$$(p \to q) \to \Diamond_t(p \to q) \qquad (\textit{Modal axiom M (reflexivity)})$$

And therefore also derive distribution over implication for possibility:

$$\Diamond_t(p \to q) \to (\Diamond_t p \to \Diamond_t q)$$

## Modal Axiom M: Reflexivity

$$p \rightarrow \Diamond_t p$$

This axiom follows from how we have defined the semantics of our indexed modal logic (Definition 5). It follows from the fact that the demon could choose to do nothing.

We can provide a bit more color to the above as follows:

We can also derive this rule from necessitation, defined above (and from the general intuition / semantics of modal logic that necessity implies possibility). First, we can say that necessity implies possibility. We can see this a) from a possible worlds perspective and b) directly from our axioms. From a possible worlds perspective, this follows from the definition of the operators. Necessity means that there is truth at every accessible possible world, while possibility means there is truth at some accessible possible world, which puts that possible truth in time $t$ as a subset of necessary truth in time $t$. From the axioms, we verify

$$\Box_t p \rightarrow \Diamond_t p \qquad \textit{(Theorem to verify, which also corresponds to Modal Axiom D (serial))}$$
$$\neg\Box_t p \vee \Diamond_t p \qquad\qquad\qquad\qquad\qquad \textit{(Applying } p \rightarrow q \textit{ is equiv. } \neg p \vee q\textit{)}$$
$$\Diamond_t \neg p \vee \Diamond_t p \qquad\qquad\qquad\qquad \textit{(By modal conversion, } \neg\Box_t p \rightarrow \Diamond_t \neg p\textit{)}$$
$$\textit{(Which for our semantics is tautological)}$$

That is, in time $t$ it is possible that $p$ or it is possible that $p$, which allows for us also to not conclude anything (in the case that the demon chooses to do nothing).

We can then say,

$$(\Box_t p \rightarrow p) \rightarrow \Diamond_t p \qquad\qquad \textit{(By necessitation and } \Box_t p \rightarrow \Diamond_t p\textit{)}$$
$$p \rightarrow \Diamond_t p \qquad\qquad\qquad \textit{(By concluding } p \textit{ from necessitation)}$$

Another way to understand this axiom is again in terms of possible worlds. We can say in our system that every world is possible in relation to itself. This corresponds to the accessibility relation $\mathcal{R}ww$. As such, an equivalent way to model reflexivity is in terms of the following:

$$\Box_t p \rightarrow p$$

That is, if $\Box_t p$ holds in world $w$, then $p$ also holds in world $w$, as is the case for $\mathcal{R}ww$. We can see this by proving $\Box_t p \rightarrow p$ by contradiction. Assuming this were false, we would need to construct a world $w$ in which $\Box_t p$ is true and $p$ is false. If $\Box_t p$ is true at world $w$, then by definition $p$ is true at every world that $w$ accesses. For our purposes, this holds, as $\Box_t p$ means that it is necessary for $p$ to be the case in time $t$; any world that we access from this world $w$ (i.e. by say increasing time, running more HPO) would require $p$ to hold. Since $\mathcal{R}ww$ means that $w$ accesses itself, that means that $p$ must also be true at $w$, yielding the contradiction.

**Modal Axiom 4: Transitivity**

$$\Diamond_t \Diamond_s p \rightarrow \Diamond_{t+s} p \tag{5}$$

We can similarly understand transitivity to be valid intuitively from the behavior of the demon and in relation to the semantics of our possible worlds. We do an abbreviated treatment (in relation to what we say for reflexivity above) for brevity.

In terms of the demon, we note that in our semantics $\Diamond_t p$ means that it is possible for the demon to bring about conclusion $p$ via its choices in time $t$. Similarly, we could say the same for $\Diamond_s p$; this means it is possible for the demon to bring about conclusion $p$ in time $s$. If it is possible in time $t$ that

it is possible in time $s$ to bring about $p$, this is equivalent in our semantics to saying that it is possible in time $t + s$ to bring about conclusion $p$.

We can understand this rule (perhaps more clearly) in terms of possible worlds and accessibility relations.

For worlds $w_n$,

$$\forall w_1, \forall w_2, \forall w_3, \mathcal{R}w_1w_2 \wedge \mathcal{R}w_2w_3 \rightarrow \mathcal{R}w_1w_3$$

In other words, this accessibility relation indicates that if $w_1$ accesses $w_2$ and if $w_2$ accesses $w_3$, then $w_1$ accesses $w_3$.

For understanding this in terms of relative possibility, we could frame this as, if $w_3$ is possible relative to $w_2$ and if $w_2$ is possible relative to $w_1$, then $w_3$ is possible relative to $w_1$. For our semantics of the demon, this means that in some time if in some time $b$ we can get to some possible world $w_3$ from when we're in $w_2$ and in time $a$ we can get to some possible world $w_2$ when we're in $w_1$, then in time $a + b$ we can get to $w_3$ from $w_1$

This axiom is akin to us regarding a string of exclusively possible or exclusively necessary modal operators as just one possible or necessary modal operator, respectively; we regard then regard sum of times as the amount of time it takes to bring about $p$ (again, being necessary or possible, respectively).

**Modal Axiom 5: Symmetry**

$$\Diamond_s \Box_t p \rightarrow \Box_t p \tag{6}$$

We can similarly understand that our modal logic is symmetric; this is valid intuitively from the behavior of the demon. We further abbreviate our treatment for brevity. In terms of the demon, we note that in our semantics $\Diamond_s p$ means that it is possible for the demon to bring about conclusion $p$ via its choices in time $s$. We can also say $\Box_t p$ means that it is necessary for the demon to bring about $p$ in time $t$. If it is possible in time $s$ that it is necessary in time $s$ to bring about $p$, this is equivalent in our semantics to saying that it is necessary in time $t$ to bring about conclusion $p$. In other words, we can disregard would could have possibly happened in time $s$ from the demon's behavior and only regard what was necessary in time $t$ for the demon to do in order to bring about $p$.

As another example, consider our reduction of $\Diamond_t \neg \Diamond_t p$ to $\neg \Diamond_t p$ in our proof for deriving a defended reasoner in Section 5. While the intuitive English reading ("It's possible that it's not possible that $p$") does not seem equivalent to this reduction ("It's not possible that $p$), it is in fact valid for our semantics. Think of this in terms of the demon. If $p$ cannot be brought about in time $t$ in expectation (where $t$ is a reasonable upper bound on compute time), then that's the end of it; it doesn't matter which operators come before it (any number of $\Diamond_t$ or $\Box_t$). Adding possibility or necessity before that condition doesn't change that fact that it, for that upper bound $t$, it is not possible to bring about $p$.

This axiom is akin to us just regarding the rightmost modal operator when we have a mix of modal operators applied iteratively; we can disregard what was possible or necessary in the time prior to the rightmost operator, and say that what we can say about a sentence $p$ (whether it is possible or necessary) just relates to how much time the last operator required to bring about $p$.

**Derived axioms**

We can similarly derive other axioms of our indexed modal logic, form the axioms above. Notably,

**$\Box_t$ distributes over $\wedge$**

$\Box_t(p \wedge q) \rightarrow (\Box_t p \wedge \Box_t q)$  ($\Box_t$ *distributes over* $\wedge$)

**Inner proof 1**

$p \wedge q$

$p$

$(p \wedge q) \rightarrow p$

$\Box_t((p \wedge q) \rightarrow p)$  (*Necessitation*)

$$\square_t(p \wedge q) \rightarrow \square_t p \qquad \qquad (\textit{Distribution})$$
$$\square_t p \qquad \qquad (\textit{By assuming the hypothesis})$$

**Inner proof 2**

$$p \wedge q$$
$$q$$
$$(p \wedge q) \rightarrow q$$
$$\square_t((p \wedge q) \rightarrow q) \qquad \qquad (\textit{Necessitation})$$
$$\square_t(p \wedge q) \rightarrow \square_t q \qquad \qquad (\textit{Distribution})$$
$$\square_t q \qquad \qquad (\textit{By assuming the hypothesis})$$
$$\square_t p \wedge \square_t q \qquad (\textit{By inner proof 1, inner proof 2, assuming the hypothesis})$$

**We can show a similar result for $\Diamond_t$ and $\wedge$, omitted for brevity.**

**$\Diamond_t$ distributes over $\vee$**

$$\Diamond_t(p \vee q) \rightarrow (\Diamond_t p \vee \Diamond_t q) \qquad \qquad (\Diamond \textit{ distributes over } \vee)$$
$$\neg\square_t\neg(p \vee q) \rightarrow (\Diamond_t p \vee \Diamond_t q) \qquad \qquad (\Diamond_t a \equiv \neg\square_t\neg a)$$
$$\neg\square_t(\neg p \wedge \neg q) \rightarrow (\Diamond_t p \vee \Diamond_t q) \qquad \qquad (\neg(a \vee b) \equiv (\neg a \wedge \neg b))$$
$$\neg(\square_t\neg p \wedge \square_t\neg q) \rightarrow (\Diamond_t p \vee \Diamond_t q) \qquad \qquad (\square_t \textit{ distributes over } \wedge)$$
$$(\neg\square_t\neg p \vee \neg\square_t\neg q) \rightarrow (\Diamond_t p \vee \Diamond_t q) \qquad \qquad (\neg(a \wedge b) \equiv (\neg a \vee \neg b))$$
$$(\Diamond_t p \vee \Diamond_t q) \rightarrow (\Diamond_t p \vee \Diamond_t q) \qquad \qquad (\Diamond_t a \equiv \neg\square_t\neg a)$$

**We can show a similar result for $\square_t$ and $\vee$, omitted for brevity.**

### D.2.4  Syntax and Semantics for the Logic of our Belief in EHPO Conclusions

The logic of belief is a type of modal logic, called doxastic logic [35], where the modal operator $\mathcal{B}$ is used to express belief[12] Different types of reasoners can be defined using axioms that involve $\mathcal{B}$ [67].

We can formulate the doxastic logic of belief in Backus-Naur form:

For any atomic proposition $P$, we define recursively a well-formed formula $\phi$ as

$$\phi := P \mid \neg\phi \mid \phi \wedge \phi \mid \mathcal{B}\phi \qquad \qquad (7)$$

where $\mathcal{B}$ means "It is believed that $\phi$". We interpret this recursively where $p$ is the base case, meaning that $\phi$ is p if it is an atom, $\neg\phi$ is well-formed if $\phi$ is well-formed. We can also define $\vee$, $\rightarrow$, and $\leftrightarrow$ from $\neg$ and $\wedge$, as in propositional logic.

As stated in the belief logic portion of Section 4, we model a consistent Type 1 reasoner [67], which has access to all of propositional logic, has their beliefs logical closed under *modus ponens*, and does not derive contradictions. In other words, we have the following axioms:

$$\neg(\mathcal{B}p \wedge \mathcal{B}\neg p) \equiv \mathcal{B}p \rightarrow \neg\mathcal{B}\neg p$$

which is the consistency axiom,

$$\vdash p \rightarrow \mathcal{B}p$$

which is akin to Necessitation above in Section D.1.1 and means that we believe all tautologies, and

$$\mathcal{B}(p \rightarrow q) \rightarrow (\mathcal{B}p \rightarrow \mathcal{B}q)$$

which means that belief distributes over implication. This notably does not include

---

[12]Computer scientists do not tend to distinguish between the logic of knowledge (epistemic) and the logic of belief (doxastic) [63].

$$\mathcal{B}p \to p$$

which essentially means that we do not allow for believing $p$ to entail concluding $p$. This corresponds to us actually wanting to run hyperparameter optimization before we conclude anything to be true. We do not just want to conclude something to be true based only on *a priori* information. This is akin to picking folkore parameters and concluding they are optimal without running hyperparameter optimization.

### D.2.5   Combining Logics

It is a well known result that we can combine modal logics to make a multimodal logic [61]. In particular, we refer the reader to results on *fusion* [68].

For a brief intuition, we are able to combine our EHPO logic with belief logic since we are operating over the same set of possible worlds. The results of running EHPO produce a particular possible world, to which we apply our logic of belief in order to reason about the conclusions drawn in that world.

### D.2.6   Our Combined, Multimodal Logic and Expressing Hyperparameter Deception

We develop the following multimodal logic, which we also state in Section 4:

$$\psi := P \mid \neg\psi \mid \psi \wedge \psi \mid \Diamond_t\psi \mid \mathcal{B}\psi$$

### D.2.7   Axioms

We give this multimodal logic semantics to express our $t$-non-deceptiveness axiom, which we repeat below for completeness:

For any formula $p$,
$$\neg\left(\Diamond_t\mathcal{B}p \wedge \Diamond_t\mathcal{B}\neg p\right)$$

We can similarly express the $t$-non-deceptiveness axiom:

For any formula $p$,
$$\Diamond_t\mathcal{B}p \to \neg\Diamond_t\mathcal{B}\neg p$$

We can also express a $t$-deceptiveness-axiom:

For any formula p,
$$\Diamond_t\mathcal{B}p \wedge \Diamond_t\mathcal{B}\neg p$$

To reiterate, *multimodal* just means that we have multiple different modes of reasoning, in this case our $\Diamond_t$ semantics for the demon doing EHPO and our consistent Type 1 reasoner operator $\mathcal{B}$.

Given a reasonable maximum time budget $t$, we say that EHPO is $t$-non-deceptive if it satisfies all of axioms above. Moreover, based on this notion of $t$-non-deceptiveness, we can express what it means to have a defense to being deceived.

### D.2.8   Some notes on strength of belief and belief update

There are potentially interesting connections between our work on defending against hyperparameter deception and belief update [19]. Notably, one could view our notion of skeptical belief as related to work done on "strength of belief" and belief update, or dynamic doxastic logic [43, 63, 71]. Instead of picking an EHPO runtime *a priori* and then running a defended EHPO and at the end evaluating whether or not we believe the conclusions we draw, we could iteratively update and test our belief and terminate if a certain belief threshold is met. In such quantitative theories of belief change, the degree of acceptance of a sentence is represented by a numerical value. Those numerical values can be updated in light of new information (so-called "soft" information updates) [2, 70]. Exploring this is out of scope for our work here, but could be an interesting future research direction for how to reason about empirical results that imply inconsistent conclusions.

# E    Section 5 Appendix: Additional Notes on Defenses

## E.1    Proving a defended reasoners

Suppose that we have been drawing conclusions using some "naive" belief operator $\mathcal{B}_\mathrm{n}$ (based on a conclusion function $\mathcal{F}_\mathrm{n}$) that satisfies the axioms of Section 4.3. We want to use $\mathcal{B}_\mathrm{n}$ to construct a new operator $\mathcal{B}_*$, which is guaranteed to be deception-free. One straightforward way to do this is to define the belief operator $\mathcal{B}_*$ such that for any statement $p$,

$$\mathcal{B}_* p \;\equiv\; \mathcal{B}_\mathrm{n} p \;\wedge\; \neg\Diamond_t \mathcal{B}_\mathrm{n} \neg p.$$

That is, we conclude $p$ only if both our naive reasoner would have concluded $p$, and it is impossible for an adversary to get it to conclude $\neg p$ in time $t$. This enables us to show $t$-non-deceptiveness, following directly from the axioms in a proof by contradiction: Suppose $\mathcal{B}_*$ can be deceived, i.e. $\Diamond_t \mathcal{B}_* p \wedge \Diamond_t \mathcal{B}_* \neg p$ is True:

| | | Rule |
|---|---|---|
| $\Diamond_t \mathcal{B}_* p$ | $\equiv\;\; \Diamond_t \left( \mathcal{B}_\mathrm{n} p \wedge \neg\Diamond_t \mathcal{B}_\mathrm{n}\neg p \right)$ | Applying $\Diamond_t$ to the definition of $\mathcal{B}_* p$   (1) |
| | $\rightarrow\;\; \Diamond_t \left( \neg\Diamond_t \mathcal{B}_\mathrm{n}\neg p \right)$ | Reducing a conjunction to either of its terms: $(a \wedge b) \rightarrow b$ |
| | $\rightarrow\;\; \neg\Diamond_t \mathcal{B}_\mathrm{n}\neg p$ | Symmetry; dropping all but the right-most operator: $\Diamond_t(\Diamond_t a) \rightarrow \Diamond_t a$ |

We provide more detail on these transformations than we do in the main text. The first application is simple; we just put parentheses around our definition of $\mathcal{B}_*$, and apply $\Diamond_t$ to it. The second step is also simple. We apply a change to whats inside the parentheses, i.e. just the definition of $\mathcal{B}_*$. Because this is a conjunction, in order for it to be true, both components have to be true. So, we can reduce the conjunction to just it's second term.

The part that is more unfamiliar is the application of Modal Axiom 5 (Symmetry) to reduce the number of $\Diamond_t$ operators. We provide this example above in Section D, where we explain why Modal Axiom 5 holds for our EHPO logic semantics. We reiterate here for clarity:

While the intuitive English reading ("It's possible that it's not possible that $p$") does not seem equivalent to this reduction ("It's not possible that $p$), it is in fact valid for our semantics. Think of this in terms of the demon. If $p$ cannot be brought about in time $t$ in expectation (where $t$ is a reasonable upper bound on compute time), then that's the end of it; it doesn't matter which operators come before it (any number of $\Diamond_t$ or $\Box_t$). Adding possibility or necessity before that condition doesn't change that fact that it, for that upper bound $t$, it is not possible to bring about $p$.

We then pause to apply our axioms to the right side of the conjunction, $\Diamond_t \mathcal{B}_* \neg p$ :

| | | Rule |
|---|---|---|
| $\Diamond_t \mathcal{B}_* \neg p$ | $\equiv\;\; \Diamond_t \left( \mathcal{B}_\mathrm{n} \neg p \wedge \neg\Diamond_t \mathcal{B}_\mathrm{n} p \right)$ | Applying $\Diamond_t$ to the definition of $\mathcal{B}_* \neg p$   (1) |
| | $\rightarrow\;\; \Diamond_t \mathcal{B}_\mathrm{n}\neg p \wedge \Diamond_t \neg\Diamond_t \mathcal{B}_\mathrm{n} p$ | Distributing $\Diamond_t$ over $\wedge$: $\Diamond_t(a \wedge b) \rightarrow (\Diamond_t a \wedge \Diamond_t b)$ |
| | $\rightarrow\;\; \Diamond_t \mathcal{B}_\mathrm{n}\neg p$ | Reducing a conjunction to either of its terms: $(a \wedge b) \rightarrow a$ |

This transformation is much like the one above. We similarly apply $\Diamond_t$ to the definition of $\mathcal{B}_*$.

We then distribute $\Diamond_t$ over the definition, which holds for our logic since possibility distributes over and. We prove this for our logic in Section D, and provide an intuitive explanation here. If it is possible in time $t$ to bring about a particular formula, then it must also be possible to bring about the sub-conditions that compose that formula in time $t$. If this were not the case, then we would not be able to satisfy bringing about the whole formula in time $t$.

Lastly, as in the first example, we reduce the conjunction to one of its terms (this time taking the first, rather than the second).

We now bring both sides of the conjunction back together: $\Diamond_t \mathcal{B}_* p \wedge \Diamond_t \mathcal{B}_* \neg p \;\equiv\; \neg\Diamond_t \mathcal{B}_\mathrm{n}\neg p \wedge \Diamond_t \mathcal{B}_\mathrm{n}\neg p$. The right-hand side is of the form $\neg a \wedge a$, which must be False. This contradicts our initial assumption that $\mathcal{B}_*$ is $t$-deceptive (i.e., $\Diamond_t \mathcal{B}_* p \wedge \Diamond_t \mathcal{B}_* \neg p$ is True). Therefore, $\mathcal{B}_*$ is $t$-non-deceptive.

## E.2 Theoretically Validating Defenses to Hyperparameter Deception

We prove Theorem 1:

**Theorem 1.** *Suppose that the set of allowable hyper-HPs $\mathcal{C}$ of $H$ is constrained, such that any two allowable random-search distributions $\mu$ and $\nu$ have Renyi-$\infty$-divergence at most a constant, i.e. $D_\infty(\mu\|\nu) \leq \gamma$. The $(K, R)$-defended random-search EHPO of Definition 7 is guaranteed to be $t$-non-deceptive if we set $R \geq \sqrt{t \exp(\gamma K)/K} = O(\sqrt{t})$.*

Suppose we are considering HPO via random search [4], in which the set of allowable hyper-hyperparameters contains tuples $(\mu, M)$, where $\mu$ is a distribution over all possible hyperparameter sets $\Lambda$ and $M$ is the number of different hyperparameter configuration trials to run. This set $S$ is the Cartesian product of the set of allowable distributions $D$ ($\mu \in D$) and $M$.

Suppose that for any two allowable distributions $\mu, \nu \in D$ and any event $A$ (a measurable subset of $\Lambda$), $\mu(A) \leq e^\gamma \cdot \nu(A)$ (i.e., the Renyi $\infty$-divergence between any pair of distributions is at most $\gamma$). This bounds how much the choice of hyper-hyperparameter can affect the hyperparameters in HPO.

We also suppose we start from a naive reasoner (expressed via the operator $\mathcal{B}_n$), which draws conclusions based on a log with $K$ trials. For this scenario, we are only concerned with the reasoner's conclusions from $K$-trial logs. We therefore assume w.l.o.g. that the reasoner draws no conclusions unless presented with exactly one log with exactly $K$ trials.

For some constant $R \in \mathbb{N}$, we construct a new reasoner $\mathcal{B}_*$ that does the following: It draws conclusions only from a single log with exactly $KR$ trials (otherwise it concludes nothing). To evaluate a proposition $p$, it splits the log into $R$ groups of $K$ trials each, evaluates $\mathcal{B}_n$ on $p$ on each of those $R$ groups, and then concludes $p$ only if $\mathcal{B}_n$ also concluded $p$ on all $R$ groups.

Now consider a particular (arbitrary) proposition $p$. Since $\mathcal{B}_*$ draws conclusions based on only a single log, any strategy $\sigma$ for the demon is equivalent to one that maintains at most one log at all times (the "best" log it found so far for its purposes, as it can discard the rest).

Let $Q$ be the supremum, taken over all allowable distributions $\mu$, of the probability that running a group of $K$ random search trials using that distribution will result in a log that would convince the $\mathcal{B}_n$ of $p$. Similarly, let $Q_\neg$ be the supremum, taken over all allowable distributions $\nu$, of the probability that running a group of $K$ trials using that distribution will result in a log that would convince $\mathcal{B}_n$ of $\neg p$.

Observe that $Q$ is the probability of an event in a product distribution of $K$ independent random variables each distributed according to $\mu$, and similarly for $Q_\neg$, and the corresponding events are disjoint. By independent additivity of the Renyi divergence, the Renyi $\infty$-divergence between these corresponding product measures will be $\gamma K$. It follows that

$$1 - Q \geq \exp(-\gamma K)Q_\neg$$

and

$$1 - Q_\neg \geq \exp(-\gamma K)Q$$

From here it's fairly easy to conclude that

$$Q + Q_\neg \leq \frac{2}{1 + \exp(-\gamma K)}.$$

Now, an EHPO procedure using random search with $KR$ trials will convince $\mathcal{B}_*$ of $p$ with probability $Q^R$, since all $R$ independently sampled groups of $K$ trials must "hit" and each hit happens with probability $Q$. Thus, the expected time it will take the fastest strategy to convince us of $p$ is $Q^{-R} \cdot KR$. Similarly, the fastest strategy to convince us of $\neg p$ takes expected time $Q_\neg^{-R} \cdot KR$.

Suppose now, by way of contradiction, that the $t$-non-deceptiveness axiom is violated, and there are strategies that can achieve both of these in time at most $t$. That is,

$$Q^{-R} \cdot KR \leq t \qquad \text{and} \qquad Q_\neg^{-R} \cdot KR \leq t.$$

From here, it's fairly easy to conclude that

$$Q + Q_\neg \geq 2 \left( \frac{KR}{t} \right)^{1/R}.$$

Combining this with our conclusion above gives

$$\left( \frac{KR}{t} \right)^{1/R} \leq \frac{1}{1 + \exp(-\gamma K)}.$$

It's clear that we can cause this to be violated by setting $R$ to be large enough. Observe that

$$\frac{1}{1 + \exp(-\gamma K)} \leq \exp(-\exp(-\gamma K)),$$

so

$$\left( \frac{KR}{t} \right)^{1/R} \leq \exp(-\exp(-\gamma K)).$$

Taking the root of both sides gives

$$\left( \frac{KR}{t} \right)^{\frac{1}{R \exp(-\gamma K)}} \leq \frac{1}{e}.$$

To simplify this expression, let $\beta$ denote

$$\beta = R \exp(-\gamma K).$$

So that

$$\left( \frac{\beta K}{t \exp(-\gamma K)} \right)^{1/\beta} \leq \frac{1}{e}.$$

Finally, we set $R$ such that

$$\beta = \sqrt{\frac{t \exp(-\gamma K)}{K}}.$$

To give

$$\left( \frac{1}{\beta} \right)^{1/\beta} \leq \frac{1}{e}.$$

But this is impossible, as the minimum of $x^x$ occurs above $1/e$. This setting of $R$ gives

$$R = \beta \exp(\gamma K) = \sqrt{\frac{t \exp(\gamma K)}{K}} = O(\sqrt{t}).$$

This shows that, for this task, if we run our constructed EHPO with $R = O(\sqrt{t})$ assigned in this way, it will be guaranteed to be $t$-non-deceptive.

### E.3 Defense Experiments

In this section we provide more information about the implementation of a random-search-based defense to hyperparameter deception in Wilson et al. [72], which we discuss in Section 5.

#### E.3.1 Our Implemented Defense Algorithm

The defense we implement in our experiments is a bit different than what we describe in our theoretical results in Section 5. In particular, in practice it is easier to implement subsampling rather than resampling.

**Protocol of Selecting Hyper-HPs.** As partially illustrated in Figure 1 and elaborated on in Appendix B, Wilson et al. [72]'s choice of hyper-HPs does not capture the space where Adam effectively simulates Heavy Ball. In Wilson et al. [72]i, Adam-specific HPs like numerical variable $\epsilon = 10^{-8}$ [42] are treated as constants, leading to a biased HP-search space.

In contrast, we select the hyper-HPs of $\epsilon$ following a dynamic searching protocol:

Inspired by [11], we start from a wide range $\epsilon \in [10^{-12}, 10^{12}]$ as a wide search space. We iteratively select powers-of-10 grids that are uniformly spaced in the logarithmic scale of the current range. For instance, the selected grids for the prior range would be $\{10^{-12}, 10^{-11}, \cdots, 10^{11}, 10^{12}\}$. We perform a single run on each grid selected, and shrink the range towards grids where the best performance are achieved. The shrinkage follows the policy of either $\times 10$ to the lower boundary or $\times 0.1$ to the upper boundary. For example, for the prior range, we found the best performance is achieved on grid $10^{11}$, so we multiply the lower boundary $10^{-12}$ with 10 and shift the range to $\epsilon \in [10^{-11}, 10^{12}]$. Our protocol terminates with $\epsilon \in [10^{10}, 10^{12}]$ as the final hyper-HPs that we use for our defended random search EHPO.

**Scaling Learning Rate $\eta$.** Note that directly applying the hyper-HP of $\epsilon \in [10^{10}, 10^{12}]$ to Adam would lead to extremely slow convergence, since essentially large $\epsilon$ indicates a small effective learning rate $\eta$. Similar to Choi et al. [11], we explore a shifted hyper-HPs for the $\eta$, scaled proportionally with $\epsilon$. Specifically, note that a large $\epsilon$ would make the update of Adam approach the update rule in the Heavy Ball method; for any randomly selected $\epsilon \in [10^{10}, 10^{12}]$, we perform the random search of $\eta/\epsilon$ instead of $\eta$ itself in the space of $[0.5, 0.7]$, which is the search space of HB's learning rate shown in [72].

#### E.3.2 Experimental setup

We follow the setup from [72], where the details are specified in Section B.2.1.

#### E.3.3 Code

The code for running these experiments can be found at `https://github.com/pasta41/deception`.

#### E.3.4 Associated results and logs

In line with our notion of a log, we provide heatmaps of our logs in Figures 13, 14 and that correspond with our results in Section 5. We note that the performance of Heavy Ball for some random seeds is very bad (e.g., 10% test accuracy). The performance varies widely – also nearing 92% for different random seeds. We affirm that this is the search space that yields the best results for Heavy Ball ( 92%).

The results for Heavy Ball exhibit large variance. This illustrates a strength of our defensed: it actually helps with robustness against potentially making the wrong conclusion about Heavy Ball's performance (more generally), due to not making conclusions off of a single result (and perhaps using a random seed for which performance is particularly bad). We make a different claim about relative algorithm performance than Wilson et al. [72] about Heavy Ball (i.e., we do not claim that it is better than Adam); but we do not reach this conclusion for the wrong reason (i.e., that we got one bad Heavy Ball result for a particular random seed).

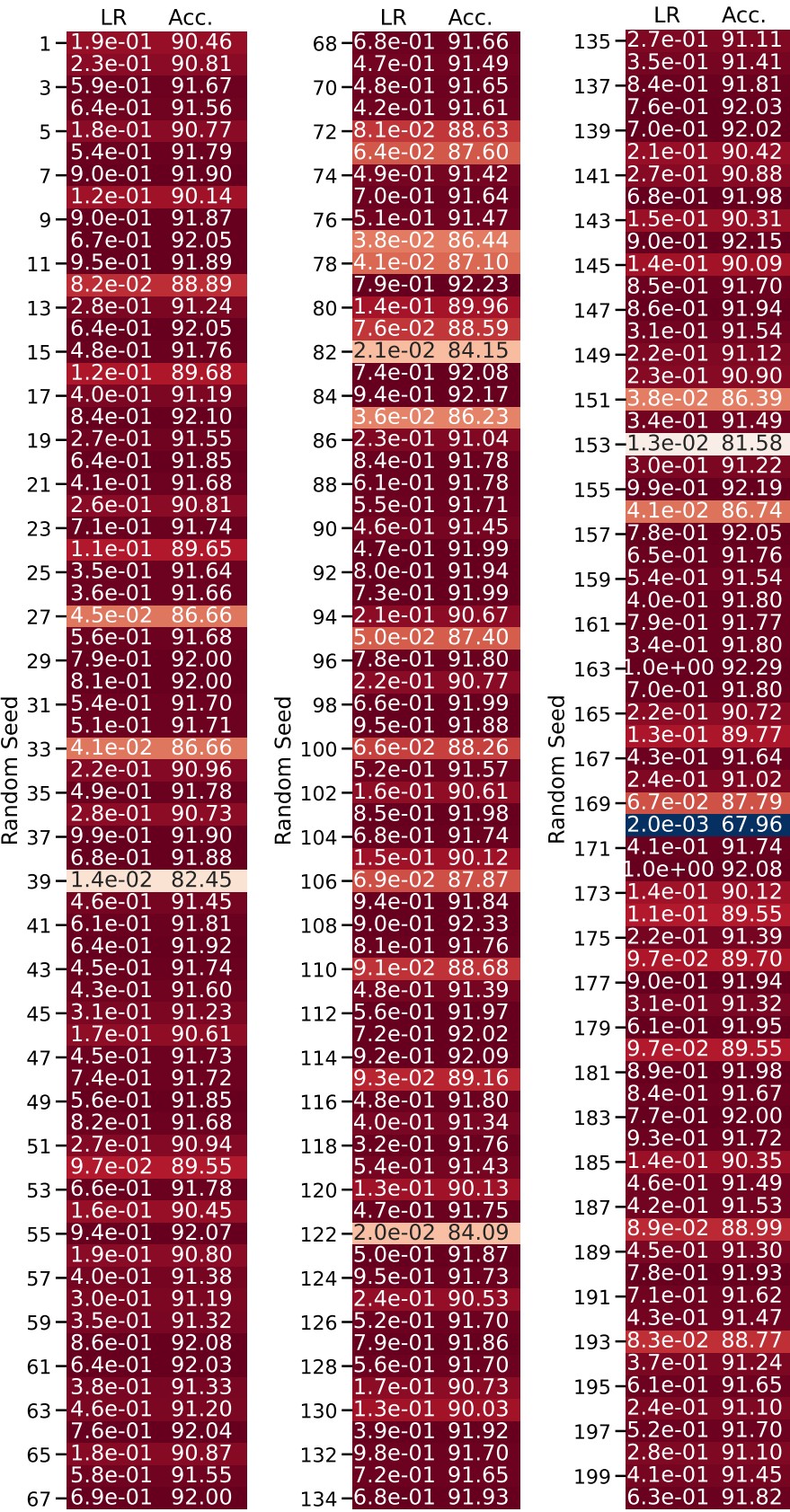

Figure 10: Heatmap logs of SGD defended random search. Redder rows indicate higher test accuracy.

| Random Seed | LR | Mntm. | Acc. | Random Seed | LR | Mntm. | Acc. | Random Seed | LR | Mntm. | Acc. |
|---|---|---|---|---|---|---|---|---|---|---|---|
| 1 | 0.77 | 0.63 | 92.05 | 68 | 0.38 | 0.62 | 92.07 | 135 | 0.04 | 0.93 | 91.74 |
| 2 | 0.68 | 0.72 | 91.72 | 69 | 0.41 | 0.93 | 10.00 | 136 | 0.42 | 0.95 | 10.00 |
| 3 | 0.72 | 0.96 | 10.00 | 70 | 0.90 | 0.85 | 10.00 | 137 | 0.32 | 0.84 | 91.67 |
| 4 | 0.93 | 0.55 | 10.00 | 71 | 0.51 | 0.60 | 91.88 | 138 | 0.11 | 0.64 | 91.34 |
| 5 | 0.92 | 0.54 | 91.93 | 72 | 0.34 | 0.62 | 92.00 | 139 | 0.00 | 0.74 | 81.60 |
| 6 | 0.18 | 0.92 | 91.59 | 73 | 0.88 | 0.55 | 10.00 | 140 | 0.44 | 0.59 | 92.04 |
| 7 | 0.96 | 0.93 | 10.00 | 74 | 0.85 | 0.98 | 10.00 | 141 | 0.62 | 0.62 | 92.05 |
| 8 | 0.24 | 0.85 | 91.70 | 75 | 0.63 | 0.87 | 10.00 | 142 | 0.37 | 0.94 | 10.00 |
| 9 | 0.78 | 0.90 | 10.00 | 76 | 0.40 | 0.88 | 28.27 | 143 | 0.66 | 0.64 | 10.00 |
| 10 | 0.58 | 0.96 | 10.00 | 77 | 0.84 | 0.87 | 10.00 | 144 | 0.42 | 0.74 | 91.82 |
| 11 | 0.44 | 0.99 | 10.00 | 78 | 0.70 | 0.74 | 91.70 | 145 | 0.44 | 0.51 | 91.85 |
| 12 | 0.91 | 0.73 | 10.00 | 79 | 0.60 | 0.52 | 91.93 | 146 | 0.65 | 0.97 | 10.00 |
| 13 | 0.49 | 0.62 | 91.93 | 80 | 0.38 | 0.74 | 91.88 | 147 | 0.12 | 0.80 | 91.93 |
| 14 | 0.52 | 0.84 | 54.31 | 81 | 0.21 | 0.58 | 91.71 | 148 | 0.41 | 0.79 | 92.00 |
| 15 | 0.40 | 0.55 | 91.97 | 82 | 0.66 | 0.81 | 28.87 | 149 | 0.78 | 0.75 | 10.00 |
| 16 | 0.72 | 0.71 | 91.70 | 83 | 0.61 | 0.73 | 91.72 | 150 | 0.85 | 0.78 | 26.81 |
| 17 | 0.48 | 0.71 | 91.86 | 84 | 0.40 | 0.59 | 92.11 | 151 | 0.23 | 0.58 | 91.59 |
| 18 | 0.29 | 0.87 | 91.82 | 85 | 0.59 | 0.74 | 91.69 | 152 | 0.66 | 0.88 | 10.00 |
| 19 | 0.99 | 0.99 | 10.00 | 86 | 0.39 | 0.94 | 10.00 | 153 | 0.06 | 0.53 | 89.91 |
| 20 | 0.58 | 0.93 | 10.00 | 87 | 0.53 | 0.66 | 91.91 | 154 | 0.03 | 0.67 | 89.51 |
| 21 | 0.21 | 0.53 | 91.10 | 88 | 0.54 | 0.85 | 10.00 | 155 | 0.63 | 0.81 | 10.00 |
| 22 | 0.25 | 0.89 | 91.55 | 89 | 0.42 | 0.53 | 92.11 | 156 | 0.33 | 0.94 | 10.00 |
| 23 | 0.40 | 0.76 | 91.85 | 90 | 0.60 | 0.57 | 91.95 | 157 | 0.33 | 0.70 | 92.03 |
| 24 | 0.60 | 0.69 | 91.96 | 91 | 0.99 | 0.53 | 10.00 | 158 | 0.71 | 0.74 | 10.00 |
| 25 | 0.52 | 0.84 | 10.00 | 92 | 0.58 | 0.74 | 92.07 | 159 | 0.81 | 0.54 | 92.09 |
| 26 | 0.47 | 0.74 | 92.24 | 93 | 0.41 | 0.64 | 92.41 | 160 | 0.99 | 0.59 | 91.88 |
| 27 | 0.73 | 0.96 | 10.00 | 94 | 0.29 | 0.72 | 92.18 | 161 | 0.43 | 0.85 | 91.34 |
| 28 | 0.96 | 0.79 | 10.00 | 95 | 0.44 | 0.77 | 91.63 | 162 | 0.37 | 0.73 | 92.09 |
| 29 | 0.72 | 0.54 | 91.80 | 96 | 0.96 | 0.95 | 10.00 | 163 | 0.55 | 0.89 | 10.00 |
| 30 | 0.94 | 0.61 | 10.00 | 97 | 0.49 | 0.55 | 91.78 | 164 | 0.48 | 0.88 | 52.21 |
| 31 | 0.20 | 0.67 | 91.52 | 98 | 0.29 | 0.53 | 92.02 | 165 | 0.50 | 0.56 | 91.82 |
| 32 | 0.91 | 0.84 | 10.00 | 99 | 0.59 | 0.94 | 10.00 | 166 | 0.78 | 0.94 | 10.00 |
| 33 | 0.79 | 0.71 | 10.00 | 100 | 0.96 | 0.64 | 10.00 | 167 | 0.96 | 0.51 | 10.00 |
| 34 | 0.15 | 0.83 | 91.87 | 101 | 0.10 | 0.94 | 91.74 | 168 | 0.26 | 0.57 | 91.78 |
| 35 | 0.25 | 0.78 | 92.28 | 102 | 0.16 | 0.88 | 91.85 | 169 | 0.48 | 0.87 | 80.33 |
| 36 | 0.32 | 0.89 | 91.01 | 103 | 0.65 | 0.55 | 91.83 | 170 | 0.09 | 0.91 | 91.73 |
| 37 | 0.38 | 0.53 | 91.99 | 104 | 0.65 | 0.66 | 92.06 | 171 | 0.80 | 0.68 | 10.00 |
| 38 | 0.11 | 0.53 | 90.97 | 105 | 0.23 | 0.95 | 26.17 | 172 | 0.97 | 0.54 | 92.04 |
| 39 | 0.34 | 0.86 | 91.59 | 106 | 0.63 | 0.96 | 10.00 | 173 | 0.12 | 0.78 | 91.93 |
| 40 | 0.46 | 0.96 | 10.00 | 107 | 0.44 | 0.85 | 90.81 | 174 | 0.71 | 0.62 | 10.00 |
| 41 | 0.45 | 0.82 | 91.45 | 108 | 0.89 | 0.61 | 10.00 | 175 | 0.77 | 0.54 | 91.89 |
| 42 | 0.48 | 0.60 | 92.07 | 109 | 0.30 | 0.90 | 89.66 | 176 | 0.40 | 0.87 | 89.57 |
| 43 | 0.51 | 0.73 | 92.05 | 110 | 0.41 | 0.70 | 92.44 | 177 | 0.20 | 0.56 | 91.43 |
| 44 | 0.27 | 0.60 | 91.88 | 111 | 0.56 | 0.78 | 91.80 | 178 | 0.64 | 0.75 | 91.95 |
| 45 | 0.68 | 0.89 | 10.00 | 112 | 0.15 | 0.88 | 92.04 | 179 | 0.59 | 0.74 | 10.00 |
| 46 | 0.72 | 0.74 | 90.89 | 113 | 0.30 | 0.81 | 91.98 | 180 | 0.12 | 0.52 | 90.99 |
| 47 | 0.63 | 0.96 | 10.00 | 114 | 0.42 | 0.80 | 92.03 | 181 | 0.24 | 0.85 | 91.82 |
| 48 | 0.25 | 0.90 | 91.24 | 115 | 0.33 | 0.67 | 91.81 | 182 | 0.44 | 0.61 | 91.95 |
| 49 | 0.07 | 0.89 | 92.06 | 116 | 0.24 | 0.94 | 75.93 | 183 | 0.73 | 0.94 | 10.00 |
| 50 | 0.65 | 0.61 | 91.84 | 117 | 0.62 | 0.94 | 10.00 | 184 | 0.37 | 0.60 | 91.91 |
| 51 | 0.65 | 0.66 | 10.00 | 118 | 0.93 | 0.59 | 10.00 | 185 | 0.36 | 0.70 | 91.92 |
| 52 | 0.04 | 0.77 | 90.96 | 119 | 0.35 | 0.77 | 92.02 | 186 | 0.86 | 0.68 | 10.00 |
| 53 | 0.49 | 0.95 | 10.00 | 120 | 0.85 | 0.69 | 10.00 | 187 | 0.99 | 0.91 | 10.00 |
| 54 | 0.37 | 0.67 | 92.22 | 121 | 0.68 | 0.97 | 10.00 | 188 | 0.48 | 0.76 | 91.92 |
| 55 | 0.48 | 0.69 | 92.35 | 122 | 0.33 | 0.80 | 91.87 | 189 | 0.87 | 0.60 | 92.06 |
| 56 | 0.06 | 0.67 | 90.71 | 123 | 0.01 | 0.96 | 91.14 | 190 | 0.91 | 0.61 | 10.00 |
| 57 | 0.62 | 0.91 | 10.00 | 124 | 0.46 | 0.70 | 92.18 | 191 | 0.32 | 0.53 | 91.86 |
| 58 | 0.90 | 0.60 | 10.00 | 125 | 0.55 | 0.57 | 92.21 | 192 | 0.70 | 0.63 | 91.93 |
| 59 | 0.75 | 0.52 | 92.11 | 126 | 0.55 | 0.76 | 91.50 | 193 | 0.34 | 0.68 | 91.97 |
| 60 | 0.16 | 0.89 | 91.77 | 127 | 0.27 | 0.57 | 91.52 | 194 | 0.60 | 0.72 | 92.01 |
| 61 | 0.76 | 0.72 | 10.00 | 128 | 0.31 | 0.74 | 92.17 | 195 | 0.17 | 0.70 | 91.67 |
| 62 | 0.58 | 0.84 | 10.00 | 129 | 0.03 | 0.87 | 91.04 | 196 | 0.49 | 0.63 | 92.08 |
| 63 | 0.39 | 0.73 | 92.19 | 130 | 0.80 | 0.74 | 10.00 | 197 | 0.01 | 0.61 | 83.90 |
| 64 | 0.73 | 0.74 | 91.22 | 131 | 0.19 | 0.68 | 91.82 | 198 | 0.30 | 0.83 | 92.02 |
| 65 | 0.40 | 0.92 | 10.00 | 132 | 0.80 | 0.98 | 10.00 | 199 | 0.52 | 0.74 | 91.65 |
| 66 | 0.08 | 0.94 | 91.74 | 133 | 0.20 | 0.92 | 90.90 | 200 | 0.16 | 0.83 | 91.97 |
| 67 | 0.99 | 0.83 | 10.00 | 134 | 0.88 | 0.56 | 91.98 | | | | |

Figure 11: Heatmap logs of Heavy Ball (HB) defended random search. Redder rows indicate higher test accuracy.

Figure 12 heatmap data (Random Seed, LR, Eps., Acc.):

| Seed | LR | Eps. | Acc. | Seed | LR | Eps. | Acc. | Seed | LR | Eps. | Acc. |
|---|---|---|---|---|---|---|---|---|---|---|---|
| 1 | 1.7e+11 | 2.5e+11 | 92.06 | 68 | 1.7e+11 | 2.8e+11 | 91.79 | 135 | 7.5e+10 | 1.3e+11 | 91.58 |
| 2 | 2.0e+11 | 3.8e+11 | 91.37 | 69 | 7.6e+10 | 1.1e+11 | 92.05 | 136 | 3.1e+11 | 5.6e+11 | 91.29 |
| 3 | 1.3e+11 | 2.1e+11 | 91.84 | 70 | 1.9e+11 | 3.1e+11 | 91.94 | 137 | 1.9e+11 | 2.8e+11 | 91.71 |
| 4 | 3.8e+11 | 7.5e+11 | 91.39 | 71 | 2.4e+11 | 3.6e+11 | 91.74 | 138 | 2.5e+11 | 4.2e+11 | 91.64 |
| 5 | 9.8e+10 | 1.8e+11 | 91.93 | 72 | 4.9e+11 | 7.1e+11 | 92.13 | 139 | 3.4e+11 | 5.3e+11 | 92.02 |
| 6 | 4.4e+11 | 6.7e+11 | 91.77 | 73 | 5.1e+11 | 7.3e+11 | 91.73 | 140 | 1.6e+11 | 2.6e+11 | 91.77 |
| 7 | 1.8e+11 | 3.5e+11 | 91.86 | 74 | 6.9e+10 | 1.3e+11 | 91.71 | 141 | 6.2e+11 | 9.6e+11 | 91.54 |
| 8 | 2.7e+11 | 4.2e+11 | 91.53 | 75 | 1.1e+11 | 2.3e+11 | 91.75 | 142 | 1.5e+11 | 2.4e+11 | 91.60 |
| 9 | 2.6e+11 | 3.9e+11 | 91.73 | 76 | 8.8e+10 | 1.4e+11 | 91.38 | 143 | 1.9e+11 | 3.0e+11 | 91.38 |
| 10 | 1.8e+11 | 3.1e+11 | 91.64 | 77 | 4.8e+11 | 8.7e+11 | 91.78 | 144 | 3.8e+11 | 6.3e+11 | 91.72 |
| 11 | 6.3e+11 | 9.2e+11 | 91.78 | 78 | 8.6e+10 | 1.5e+11 | 91.66 | 145 | 4.4e+11 | 6.6e+11 | 91.15 |
| 12 | 3.0e+11 | 5.5e+11 | 91.63 | 79 | 5.5e+11 | 8.8e+11 | 91.64 | 146 | 3.0e+11 | 4.3e+11 | 91.68 |
| 13 | 4.2e+11 | 6.2e+11 | 91.59 | 80 | 1.2e+11 | 2.3e+11 | 91.92 | 147 | 5.5e+11 | 9.9e+11 | 91.21 |
| 14 | 4.5e+11 | 8.4e+11 | 91.79 | 81 | 2.0e+11 | 3.1e+11 | 91.56 | 148 | 1.7e+11 | 2.9e+11 | 91.75 |
| 15 | 4.6e+11 | 7.6e+11 | 91.66 | 82 | 7.5e+10 | 1.1e+11 | 91.71 | 149 | 4.8e+11 | 8.6e+11 | 91.74 |
| 16 | 3.8e+11 | 5.6e+11 | 91.70 | 83 | 9.3e+10 | 1.5e+11 | 91.83 | 150 | 7.2e+10 | 1.3e+11 | 91.47 |
| 17 | 7.1e+10 | 1.1e+11 | 91.89 | 84 | 4.4e+11 | 7.4e+11 | 91.46 | 151 | 8.2e+10 | 1.4e+11 | 91.30 |
| 18 | 3.2e+11 | 5.0e+11 | 91.85 | 85 | 3.8e+11 | 6.1e+11 | 92.02 | 152 | 6.0e+10 | 1.1e+11 | 91.73 |
| 19 | 5.6e+10 | 1.0e+11 | 91.54 | 86 | 1.7e+11 | 3.0e+11 | 91.62 | 153 | 2.4e+11 | 4.7e+11 | 91.51 |
| 20 | 6.0e+10 | 1.2e+11 | 91.40 | 87 | 9.2e+10 | 1.8e+11 | 91.64 | 154 | 4.1e+11 | 7.3e+11 | 91.65 |
| 21 | 3.6e+11 | 5.9e+11 | 91.55 | 88 | 1.8e+11 | 3.3e+11 | 91.83 | 155 | 3.0e+11 | 5.8e+11 | 91.38 |
| 22 | 5.6e+11 | 8.7e+11 | 91.67 | 89 | 5.2e+11 | 9.4e+11 | 91.55 | 156 | 4.1e+11 | 7.3e+11 | 91.57 |
| 23 | 2.0e+11 | 2.9e+11 | 92.00 | 90 | 9.1e+10 | 1.4e+11 | 91.80 | 157 | 1.8e+11 | 3.0e+11 | 91.08 |
| 24 | 1.8e+11 | 3.0e+11 | 91.81 | 91 | 1.8e+11 | 2.8e+11 | 91.38 | 158 | 2.1e+11 | 3.5e+11 | 91.69 |
| 25 | 2.9e+11 | 4.6e+11 | 91.45 | 92 | 3.7e+11 | 6.9e+11 | 91.54 | 159 | 5.8e+11 | 9.7e+11 | 91.67 |
| 26 | 1.9e+11 | 3.5e+11 | 91.30 | 93 | 1.9e+11 | 2.9e+11 | 91.70 | 160 | 1.5e+11 | 2.5e+11 | 91.48 |
| 27 | 3.8e+11 | 7.4e+11 | 91.74 | 94 | 5.0e+11 | 7.7e+11 | 91.58 | 161 | 1.6e+11 | 2.8e+11 | 91.50 |
| 28 | 5.1e+11 | 7.6e+11 | 91.64 | 95 | 2.2e+11 | 3.3e+11 | 91.43 | 162 | 6.0e+11 | 1.0e+12 | 91.81 |
| 29 | 4.7e+11 | 6.8e+11 | 91.39 | 96 | 5.9e+11 | 9.9e+11 | 91.72 | 163 | 8.2e+10 | 1.2e+11 | 91.63 |
| 30 | 5.1e+11 | 7.7e+11 | 91.37 | 97 | 2.8e+11 | 5.3e+11 | 91.80 | 164 | 4.9e+11 | 8.3e+11 | 91.50 |
| 31 | 3.2e+11 | 5.8e+11 | 91.48 | 98 | 6.3e+11 | 9.3e+11 | 91.91 | 165 | 1.3e+11 | 2.1e+11 | 91.49 |
| 32 | 1.9e+11 | 3.0e+11 | 91.72 | 99 | 3.0e+11 | 5.2e+11 | 91.67 | 166 | 4.8e+11 | 7.2e+11 | 91.69 |
| 33 | 6.5e+10 | 1.1e+11 | 91.53 | 100 | 2.0e+11 | 3.2e+11 | 91.92 | 167 | 4.6e+11 | 7.2e+11 | 91.60 |
| 34 | 2.8e+11 | 4.7e+11 | 91.88 | 101 | 2.1e+11 | 3.7e+11 | 91.93 | 168 | 1.3e+11 | 2.0e+11 | 91.71 |
| 35 | 3.6e+11 | 5.7e+11 | 91.63 | 102 | 4.1e+11 | 8.1e+11 | 91.60 | 169 | 3.5e+11 | 7.0e+11 | 91.46 |
| 36 | 4.8e+11 | 8.3e+11 | 91.55 | 103 | 4.5e+11 | 6.7e+11 | 91.93 | 170 | 4.7e+11 | 7.5e+11 | 91.56 |
| 37 | 5.5e+11 | 8.2e+11 | 91.84 | 104 | 3.8e+11 | 7.5e+11 | 91.51 | 171 | 4.4e+11 | 8.0e+11 | 91.78 |
| 38 | 4.1e+11 | 7.8e+11 | 91.56 | 105 | 5.3e+11 | 9.5e+11 | 91.74 | 172 | 3.3e+11 | 5.8e+11 | 91.70 |
| 39 | 2.3e+11 | 4.3e+11 | 91.90 | 106 | 2.6e+11 | 4.1e+11 | 92.05 | 173 | 5.8e+11 | 8.8e+11 | 91.84 |
| 40 | 4.2e+11 | 8.2e+11 | 91.44 | 107 | 3.4e+11 | 5.0e+11 | 91.47 | 174 | 2.8e+11 | 4.9e+11 | 91.73 |
| 41 | 1.5e+11 | 2.7e+11 | 91.61 | 108 | 3.0e+11 | 5.6e+11 | 91.65 | 175 | 1.1e+11 | 1.9e+11 | 91.80 |
| 42 | 6.6e+10 | 1.2e+11 | 91.53 | 109 | 8.6e+10 | 1.5e+11 | 91.67 | 176 | 4.4e+11 | 8.7e+11 | 91.41 |
| 43 | 4.2e+11 | 7.0e+11 | 91.98 | 110 | 4.6e+11 | 6.6e+11 | 91.51 | 177 | 9.4e+10 | 1.5e+11 | 91.71 |
| 44 | 1.8e+11 | 3.1e+11 | 91.58 | 111 | 2.3e+11 | 3.5e+11 | 91.83 | 178 | 1.7e+11 | 3.0e+11 | 91.51 |
| 45 | 3.8e+11 | 5.9e+11 | 91.80 | 112 | 1.9e+11 | 2.9e+11 | 91.61 | 179 | 1.1e+11 | 1.6e+11 | 91.97 |
| 46 | 3.3e+11 | 5.5e+11 | 91.76 | 113 | 4.2e+11 | 7.5e+11 | 91.22 | 180 | 1.4e+11 | 2.1e+11 | 91.52 |
| 47 | 4.5e+11 | 6.5e+11 | 91.56 | 114 | 2.4e+11 | 4.7e+11 | 91.52 | 181 | 9.8e+10 | 1.9e+11 | 91.81 |
| 48 | 6.7e+10 | 1.3e+11 | 91.99 | 115 | 3.2e+11 | 6.4e+11 | 91.61 | 182 | 6.1e+11 | 8.8e+11 | 91.63 |
| 49 | 3.4e+11 | 6.3e+11 | 91.44 | 116 | 2.2e+11 | 3.5e+11 | 92.05 | 183 | 3.8e+11 | 5.7e+11 | 91.58 |
| 50 | 1.5e+11 | 2.1e+11 | 91.42 | 117 | 6.2e+11 | 9.2e+11 | 91.74 | 184 | 5.8e+11 | 9.3e+11 | 91.88 |
| 51 | 2.5e+11 | 4.5e+11 | 91.54 | 118 | 4.3e+11 | 7.9e+11 | 91.33 | 185 | 2.0e+11 | 3.1e+11 | 91.50 |
| 52 | 1.8e+11 | 3.3e+11 | 91.94 | 119 | 5.7e+11 | 8.5e+11 | 91.30 | 186 | 5.6e+11 | 9.5e+11 | 91.98 |
| 53 | 2.3e+11 | 4.2e+11 | 91.62 | 120 | 4.6e+11 | 7.8e+11 | 91.77 | 187 | 4.5e+11 | 7.9e+11 | 91.29 |
| 54 | 4.4e+11 | 8.4e+11 | 91.69 | 121 | 1.3e+11 | 2.4e+11 | 91.33 | 188 | 4.7e+11 | 7.0e+11 | 92.02 |
| 55 | 2.6e+11 | 4.1e+11 | 92.14 | 122 | 2.2e+11 | 4.1e+11 | 91.86 | 189 | 1.0e+11 | 1.8e+11 | 91.70 |
| 56 | 6.6e+11 | 9.7e+11 | 91.63 | 123 | 6.5e+10 | 1.0e+11 | 91.92 | 190 | 1.6e+11 | 2.9e+11 | 91.50 |
| 57 | 1.1e+11 | 2.1e+11 | 91.69 | 124 | 5.0e+11 | 8.8e+11 | 91.74 | 191 | 3.1e+11 | 4.9e+11 | 91.81 |
| 58 | 4.6e+11 | 9.0e+11 | 91.77 | 125 | 5.4e+10 | 1.0e+11 | 91.42 | 192 | 1.5e+11 | 2.6e+11 | 92.02 |
| 59 | 3.0e+11 | 4.8e+11 | 91.97 | 126 | 4.6e+11 | 8.5e+11 | 91.81 | 193 | 1.2e+11 | 2.1e+11 | 91.51 |
| 60 | 1.6e+11 | 2.9e+11 | 91.88 | 127 | 8.8e+10 | 1.3e+11 | 91.42 | 194 | 4.4e+11 | 7.7e+11 | 91.87 |
| 61 | 3.7e+11 | 6.4e+11 | 91.64 | 128 | 3.1e+11 | 5.2e+11 | 91.76 | 195 | 3.6e+11 | 5.8e+11 | 91.48 |
| 62 | 3.1e+11 | 5.4e+11 | 91.81 | 129 | 5.5e+11 | 9.3e+11 | 91.53 | 196 | 2.8e+11 | 4.0e+11 | 91.89 |
| 63 | 4.7e+11 | 8.6e+11 | 91.59 | 130 | 5.7e+11 | 9.8e+11 | 92.00 | 197 | 4.7e+11 | 9.2e+11 | 91.83 |
| 64 | 4.1e+11 | 6.6e+11 | 91.67 | 131 | 4.3e+11 | 8.4e+11 | 91.56 | 198 | 3.0e+11 | 5.5e+11 | 91.73 |
| 65 | 4.5e+11 | 8.2e+11 | 91.47 | 132 | 4.4e+11 | 6.8e+11 | 91.01 | 199 | 2.7e+11 | 4.5e+11 | 91.69 |
| 66 | 2.5e+11 | 3.7e+11 | 91.71 | 133 | 5.9e+11 | 9.4e+11 | 91.52 | 200 | 1.7e+11 | 3.1e+11 | 91.70 |
| 67 | 1.1e+11 | 1.9e+11 | 91.80 | 134 | 4.3e+11 | 7.3e+11 | 91.44 | | | | |

Figure 12: Heatmap logs of Adam defended random search. Redder rows indicate higher test accuracy.

Figure 13 — Heatmap logs of test accuracy (Adam vs. SGD)

| Seed | Adam | SGD | | Seed | Adam | SGD | | Seed | Adam | SGD | | Seed | Adam | SGD |
|---|---|---|---|---|---|---|---|---|---|---|---|---|---|---|
| 1 | 92.06 | 90.46 | | 51 | 91.54 | 90.94 | | 101 | 91.93 | 91.57 | | 151 | 91.30 | 86.39 |
|  | 91.37 | 90.81 | |  | 91.94 | 89.55 | |  | 91.60 | 90.61 | |  | 91.73 | 91.49 |
| 3 | 91.84 | 91.67 | | 53 | 91.62 | 91.78 | | 103 | 91.93 | 91.98 | | 153 | 91.51 | 81.58 |
|  | 91.39 | 91.56 | |  | 91.69 | 90.45 | |  | 91.51 | 91.74 | |  | 91.65 | 91.22 |
| 5 | 91.93 | 90.77 | | 55 | 92.14 | 92.07 | | 105 | 91.74 | 90.12 | | 155 | 91.38 | 92.19 |
|  | 91.77 | 91.79 | |  | 91.63 | 90.80 | |  | 92.05 | 87.87 | |  | 91.57 | 86.74 |
| 7 | 91.86 | 91.90 | | 57 | 91.69 | 91.38 | | 107 | 91.47 | 91.84 | | 157 | 91.08 | 92.05 |
|  | 91.53 | 90.14 | |  | 91.77 | 91.19 | |  | 91.65 | 92.33 | |  | 91.69 | 91.76 |
| 9 | 91.73 | 91.87 | | 59 | 91.97 | 91.32 | | 109 | 91.67 | 91.76 | | 159 | 91.67 | 91.54 |
|  | 91.64 | 92.05 | |  | 91.88 | 92.08 | |  | 91.51 | 88.68 | |  | 91.48 | 91.80 |
| 11 | 91.78 | 91.89 | | 61 | 91.64 | 92.03 | | 111 | 91.83 | 91.39 | | 161 | 91.50 | 91.77 |
|  | 91.63 | 88.89 | |  | 91.81 | 91.33 | |  | 91.61 | 91.97 | |  | 91.81 | 91.80 |
| 13 | 91.59 | 91.24 | | 63 | 91.59 | 91.20 | | 113 | 91.22 | 92.02 | | 163 | 91.63 | 92.29 |
|  | 91.79 | 92.05 | |  | 91.67 | 92.04 | |  | 91.52 | 92.09 | |  | 91.50 | 91.80 |
| 15 | 91.66 | 91.76 | | 65 | 91.47 | 90.87 | | 115 | 91.61 | 89.16 | | 165 | 91.49 | 90.72 |
|  | 91.70 | 89.68 | |  | 91.71 | 91.55 | |  | 92.05 | 91.80 | |  | 91.69 | 89.77 |
| 17 | 91.89 | 91.19 | | 67 | 91.80 | 92.00 | | 117 | 91.74 | 91.34 | | 167 | 91.60 | 91.64 |
|  | 91.85 | 92.10 | |  | 91.79 | 91.66 | |  | 91.33 | 91.76 | |  | 91.71 | 91.02 |
| 19 | 91.54 | 91.55 | | 69 | 92.05 | 91.49 | | 119 | 91.30 | 91.43 | | 169 | 91.46 | 87.79 |
|  | 91.40 | 91.85 | |  | 91.94 | 91.65 | |  | 91.77 | 90.13 | |  | 91.56 | 67.96 |
| 21 | 91.55 | 91.68 | | 71 | 91.74 | 91.61 | | 121 | 91.33 | 91.75 | | 171 | 91.78 | 91.74 |
|  | 91.67 | 90.81 | |  | 92.13 | 88.63 | |  | 91.86 | 84.09 | |  | 91.70 | 92.08 |
| 23 | 92.00 | 91.74 | | 73 | 91.73 | 87.60 | | 123 | 91.92 | 91.87 | | 173 | 91.84 | 90.12 |
|  | 91.81 | 89.65 | |  | 91.71 | 91.42 | |  | 91.74 | 91.73 | |  | 91.73 | 89.55 |
| 25 | 91.45 | 91.64 | | 75 | 91.75 | 91.64 | | 125 | 91.42 | 90.53 | | 175 | 91.80 | 91.39 |
|  | 91.30 | 91.66 | |  | 91.38 | 91.47 | |  | 91.81 | 91.70 | |  | 91.41 | 89.70 |
| 27 | 91.74 | 86.66 | | 77 | 91.78 | 86.44 | | 127 | 91.42 | 91.86 | | 177 | 91.71 | 91.94 |
|  | 91.64 | 91.68 | |  | 91.66 | 87.10 | |  | 91.76 | 91.70 | |  | 91.51 | 91.32 |
| 29 | 91.39 | 92.00 | | 79 | 91.64 | 92.23 | | 129 | 91.53 | 90.73 | | 179 | 91.97 | 91.95 |
|  | 91.37 | 92.00 | |  | 91.92 | 89.96 | |  | 92.00 | 90.03 | |  | 91.52 | 89.55 |
| 31 | 91.48 | 91.70 | | 81 | 91.56 | 88.59 | | 131 | 91.56 | 91.92 | | 181 | 91.81 | 91.98 |
|  | 91.72 | 91.71 | |  | 91.71 | 84.15 | |  | 91.01 | 91.70 | |  | 91.63 | 91.67 |
| 33 | 91.53 | 86.66 | | 83 | 91.83 | 92.08 | | 133 | 91.52 | 91.65 | | 183 | 91.58 | 92.00 |
|  | 91.88 | 90.96 | |  | 91.46 | 92.17 | |  | 91.44 | 91.93 | |  | 91.88 | 91.72 |
| 35 | 91.63 | 91.78 | | 85 | 92.02 | 86.23 | | 135 | 91.58 | 91.11 | | 185 | 91.50 | 90.35 |
|  | 91.55 | 90.73 | |  | 91.62 | 91.04 | |  | 91.29 | 91.41 | |  | 91.98 | 91.49 |
| 37 | 91.84 | 91.90 | | 87 | 91.64 | 91.78 | | 137 | 91.71 | 91.81 | | 187 | 91.29 | 91.53 |
|  | 91.56 | 91.88 | |  | 91.83 | 91.78 | |  | 91.64 | 92.03 | |  | 92.02 | 88.99 |
| 39 | 91.90 | 82.45 | | 89 | 91.55 | 91.71 | | 139 | 92.02 | 92.02 | | 189 | 91.70 | 91.30 |
|  | 91.44 | 91.45 | |  | 91.80 | 91.45 | |  | 91.77 | 90.42 | |  | 91.50 | 91.93 |
| 41 | 91.61 | 91.81 | | 91 | 91.38 | 91.99 | | 141 | 91.54 | 90.88 | | 191 | 91.81 | 91.62 |
|  | 91.53 | 91.92 | |  | 91.54 | 91.94 | |  | 91.60 | 91.98 | |  | 92.02 | 91.47 |
| 43 | 91.98 | 91.74 | | 93 | 91.70 | 91.99 | | 143 | 91.38 | 90.31 | | 193 | 91.51 | 88.77 |
|  | 91.58 | 91.60 | |  | 91.58 | 90.67 | |  | 91.72 | 92.15 | |  | 91.87 | 91.24 |
| 45 | 91.80 | 91.23 | | 95 | 91.43 | 87.40 | | 145 | 91.15 | 90.09 | | 195 | 91.48 | 91.65 |
|  | 91.76 | 90.61 | |  | 91.72 | 91.80 | |  | 91.68 | 91.70 | |  | 91.89 | 91.10 |
| 47 | 91.56 | 91.73 | | 97 | 91.80 | 90.77 | | 147 | 91.21 | 91.94 | | 197 | 91.83 | 91.70 |
|  | 91.99 | 91.72 | |  | 91.91 | 91.99 | |  | 91.75 | 91.54 | |  | 91.73 | 91.10 |
| 49 | 91.44 | 91.85 | | 99 | 91.67 | 91.88 | | 149 | 91.74 | 91.12 | | 199 | 91.69 | 91.45 |
|  | 91.42 | 91.68 | |  | 91.92 | 88.26 | |  | 91.47 | 90.90 | |  | 91.70 | 91.82 |

Figure 13: Heatmap logs of test accuracy of VGG-16 on CIFAR-10 for Adam vs. SGD using our defended random search EHPO. Red indicates higher test accuracy for the given random seed.

Figure 14: Heatmap logs of test accuracy of VGG-16 on CIFAR-10 for Adam vs. Heavy Ball (HB) using our defended random search EHPO. Red indicates higher test accuracy for the given random seed.

| Seed | Adam | HB | Seed | Adam | HB | Seed | Adam | HB | Seed | Adam | HB |
|---|---|---|---|---|---|---|---|---|---|---|---|
| 1 | 92.06 | 92.05 | 51 | 91.54 | 10.00 | 101 | 91.93 | 91.74 | 151 | 91.30 | 91.59 |
| 2 | 91.37 | 91.72 | 52 | 91.94 | 90.96 | 102 | 91.60 | 91.85 | 152 | 91.73 | 10.00 |
| 3 | 91.84 | 10.00 | 53 | 91.62 | 10.00 | 103 | 91.93 | 91.83 | 153 | 91.51 | 89.91 |
| 4 | 91.39 | 10.00 | 54 | 91.69 | 92.22 | 104 | 91.51 | 92.06 | 154 | 91.65 | 89.51 |
| 5 | 91.93 | 91.93 | 55 | 92.14 | 92.35 | 105 | 91.74 | 26.17 | 155 | 91.38 | 10.00 |
| 6 | 91.77 | 91.59 | 56 | 91.63 | 90.71 | 106 | 92.05 | 10.00 | 156 | 91.57 | 10.00 |
| 7 | 91.86 | 10.00 | 57 | 91.69 | 10.00 | 107 | 91.47 | 90.81 | 157 | 91.08 | 92.03 |
| 8 | 91.53 | 91.70 | 58 | 91.77 | 10.00 | 108 | 91.65 | 10.00 | 158 | 91.69 | 10.00 |
| 9 | 91.73 | 10.00 | 59 | 91.97 | 92.11 | 109 | 91.67 | 89.66 | 159 | 91.67 | 92.09 |
| 10 | 91.64 | 10.00 | 60 | 91.88 | 91.77 | 110 | 91.51 | 92.44 | 160 | 91.48 | 91.88 |
| 11 | 91.78 | 10.00 | 61 | 91.64 | 10.00 | 111 | 91.83 | 91.80 | 161 | 91.50 | 91.34 |
| 12 | 91.63 | 10.00 | 62 | 91.81 | 10.00 | 112 | 91.61 | 92.04 | 162 | 91.81 | 92.09 |
| 13 | 91.59 | 91.93 | 63 | 91.59 | 92.19 | 113 | 91.22 | 91.98 | 163 | 91.63 | 10.00 |
| 14 | 91.79 | 54.31 | 64 | 91.67 | 91.22 | 114 | 91.52 | 92.03 | 164 | 91.50 | 52.21 |
| 15 | 91.66 | 91.97 | 65 | 91.47 | 10.00 | 115 | 91.61 | 91.81 | 165 | 91.49 | 91.82 |
| 16 | 91.70 | 91.70 | 66 | 91.71 | 91.74 | 116 | 92.05 | 75.93 | 166 | 91.69 | 10.00 |
| 17 | 91.89 | 91.86 | 67 | 91.80 | 10.00 | 117 | 91.74 | 10.00 | 167 | 91.60 | 10.00 |
| 18 | 91.85 | 91.82 | 68 | 91.79 | 92.07 | 118 | 91.33 | 10.00 | 168 | 91.71 | 91.78 |
| 19 | 91.54 | 10.00 | 69 | 92.05 | 10.00 | 119 | 91.30 | 92.02 | 169 | 91.46 | 80.33 |
| 20 | 91.40 | 10.00 | 70 | 91.94 | 10.00 | 120 | 91.77 | 10.00 | 170 | 91.56 | 91.73 |
| 21 | 91.55 | 91.10 | 71 | 91.74 | 91.88 | 121 | 91.33 | 10.00 | 171 | 91.78 | 10.00 |
| 22 | 91.67 | 91.55 | 72 | 92.13 | 92.00 | 122 | 91.86 | 91.87 | 172 | 91.70 | 92.04 |
| 23 | 92.00 | 91.85 | 73 | 91.73 | 10.00 | 123 | 91.92 | 91.14 | 173 | 91.84 | 91.93 |
| 24 | 91.81 | 91.96 | 74 | 91.71 | 10.00 | 124 | 91.74 | 92.18 | 174 | 91.73 | 10.00 |
| 25 | 91.45 | 10.00 | 75 | 91.75 | 10.00 | 125 | 91.42 | 92.21 | 175 | 91.80 | 91.89 |
| 26 | 91.30 | 92.24 | 76 | 91.38 | 28.27 | 126 | 91.81 | 91.50 | 176 | 91.41 | 89.57 |
| 27 | 91.74 | 10.00 | 77 | 91.78 | 10.00 | 127 | 91.42 | 91.52 | 177 | 91.71 | 91.43 |
| 28 | 91.64 | 10.00 | 78 | 91.66 | 91.70 | 128 | 91.76 | 92.17 | 178 | 91.51 | 91.95 |
| 29 | 91.39 | 91.80 | 79 | 91.64 | 91.93 | 129 | 91.53 | 91.04 | 179 | 91.97 | 10.00 |
| 30 | 91.37 | 10.00 | 80 | 91.92 | 91.88 | 130 | 92.00 | 10.00 | 180 | 91.52 | 90.99 |
| 31 | 91.48 | 91.52 | 81 | 91.56 | 91.71 | 131 | 91.56 | 91.82 | 181 | 91.81 | 91.82 |
| 32 | 91.72 | 10.00 | 82 | 91.71 | 28.87 | 132 | 91.01 | 10.00 | 182 | 91.63 | 91.95 |
| 33 | 91.53 | 10.00 | 83 | 91.83 | 91.72 | 133 | 91.52 | 90.90 | 183 | 91.58 | 10.00 |
| 34 | 91.88 | 91.87 | 84 | 91.46 | 92.11 | 134 | 91.44 | 91.98 | 184 | 91.88 | 91.91 |
| 35 | 91.63 | 92.28 | 85 | 92.02 | 91.69 | 135 | 91.58 | 91.74 | 185 | 91.50 | 91.92 |
| 36 | 91.55 | 91.01 | 86 | 91.62 | 10.00 | 136 | 91.29 | 10.00 | 186 | 91.98 | 10.00 |
| 37 | 91.84 | 91.99 | 87 | 91.64 | 91.91 | 137 | 91.71 | 91.67 | 187 | 91.29 | 10.00 |
| 38 | 91.56 | 90.97 | 88 | 91.83 | 10.00 | 138 | 91.64 | 91.34 | 188 | 92.02 | 91.92 |
| 39 | 91.90 | 91.59 | 89 | 91.55 | 92.11 | 139 | 92.02 | 81.60 | 189 | 91.70 | 92.06 |
| 40 | 91.44 | 10.00 | 90 | 91.80 | 91.95 | 140 | 91.77 | 92.04 | 190 | 91.50 | 10.00 |
| 41 | 91.61 | 91.45 | 91 | 91.38 | 10.00 | 141 | 91.54 | 92.05 | 191 | 91.81 | 91.86 |
| 42 | 91.53 | 92.07 | 92 | 91.54 | 92.07 | 142 | 91.60 | 10.00 | 192 | 92.02 | 91.93 |
| 43 | 91.98 | 92.05 | 93 | 91.70 | 92.41 | 143 | 91.38 | 10.00 | 193 | 91.51 | 91.97 |
| 44 | 91.58 | 91.88 | 94 | 91.58 | 92.18 | 144 | 91.72 | 91.82 | 194 | 91.87 | 92.01 |
| 45 | 91.80 | 10.00 | 95 | 91.43 | 91.63 | 145 | 91.15 | 91.85 | 195 | 91.48 | 91.67 |
| 46 | 91.76 | 90.89 | 96 | 91.72 | 10.00 | 146 | 91.68 | 10.00 | 196 | 91.89 | 92.08 |
| 47 | 91.56 | 10.00 | 97 | 91.80 | 91.78 | 147 | 91.21 | 91.93 | 197 | 91.83 | 83.90 |
| 48 | 91.99 | 91.24 | 98 | 91.91 | 92.02 | 148 | 91.75 | 92.00 | 198 | 91.73 | 92.02 |
| 49 | 91.44 | 92.06 | 99 | 91.67 | 10.00 | 149 | 91.74 | 10.00 | 199 | 91.69 | 91.65 |
| 50 | 91.42 | 91.84 | 100 | 91.92 | 10.00 | 150 | 91.47 | 26.81 | 200 | 91.70 | 91.97 |

# F  Section 6 Appendix: Additional Notes on Conclusion

## F.1  Additional Practical Takeaways

In our conclusion in Section 6, we note the following practical takeaways:

- **Researchers should have their own notion of skepticism, appropriate to their specific task.** There is no one-size-fits-all defense solution. Our results are ***broad insights*** about defended EHPO: A defended EHPO is ***always possible***, but finding an efficient one will depend on the task.

- **Researchers should make explicit how they choose hyper-HPs.** What is reasonable is ultimately a function of what the ML community accepts. Being explicit, rather than eliding hyper-HP choices, is essential for helping decide what is reasonable. As a heuristic, we recommend setting hyper-HPs such that they include HPs for which the optimizers' performance starts to degrade, as we do above.

- **Avoiding hyperparameter deception is just as important as reproducibility**. We have shown that reproducibility [7, 29, 34, 57, 64] is only part of the story for ensuring reliability. While necessary for guarding against brittle findings, it is not sufficient. We can replicate results—even statistically significant ones—that suggest conclusions that are altogether wrong.

We elaborate here that our defended random search EHPO indicates a particular form of skepticism that may (or may not) be appropriate to different ML tasks. That is, we suggest a defended EHPO, but do not claim that that EHPO is optimal or suited for all tasks. Even though it may not be optimal, the guarantees it affords would translate to other tasks (so long as the assumption is maintained that there is an upper bound on how much the hyper-HPs can control the HPs). So, while we do not necessarily encourage practitioners to use our particular defended EHPO, we do not discourage it either. The main take away is that practitioners should develop their own notion of skepticism (appropriate to their particular task) and be explicit about the assumptions they rely on when selecting hyper-HPs. The way one chooses hyper-HPs should be defensible.

When in doubt, as a heuristic we recommend using a search space that includes where an algorithm's performance starts to degrade (to be assured that a maximum, even if a local one, has been found). We refer to our dynamic two-phase protocol (which we describe in detail in Appendix E.3.1) as an example of how to do this. We first do a broad (but coarse) search. We used grid search for that initial sweep. Random search may be a better choice for some tasks. We were familiar with Wilson et al. [72] (and many have written about it), and felt confident that grid search would capture the space well based on the results that others have also reported on this task. We then used this first sweep to determine which hyper-HPs we should use for our second, finer-grained sweep. We apply our more expensive, defended EHPO for this second sweep, using the hyper-HPs we selected from the first sweep. In other words, we spent a bit of time/our compute budget justifying to ourselves that we were picking reasonable hyperparameters – instead of just picking one grid or range for random search to sample, and hoping that our results would be representative of other search spaces.

## F.2  Broader Impact

As we suggest in Section 5, our work can be considered as related to (but orthogonal with) with prior studies on reproducibility as advocating for more robust scientific practices in ML research. In particular, our work complements prior empirical studies that shine a light on reliability issues in ML—issues that relate particularly to traditionally underspecified choices in hyperparameter optimization [11, 48, 65]. In contrast to this prior work, which illustrates the issue with experiments, we provide a theoretical contribution that enables ML practitioners and researchers to defend against unreliable, inconsistent HPO. We provide a theoretically-backed mechanism to promote and facilitate more trustworthy norms and practices in ML research.

More broadly, our work can be understood as a mechanism for dealing with *measurement bias*—the misalignment between what one intends to measure and what they are actually measuring—for overall ML algorithm performance. While alleviating measurement bias is by no means novel to more mature branches of science [28], including other fields of computing [54], until recently it has been under-explored in ML. Beginning in the last couple of years, measurement bias is now coming under increased scrutiny with respect to the origins of empirical gains in ML [22, 53]. In current work, it is often difficult to disentangle whether the concluded measured performance gains are due to properties of the training algorithm or to fortuitous HP selection. Our formalization, rather than

allowing HPO choices to potentially obscure empirical results, provides confidence in the conclusions we can draw about overall algorithm performance.

Our work also highlights how there is a human element, not a just statistical one, to bias in ML pipelines: Practitioners make decisions about HPO that can heavily influence performance (e.g., choice of hyper-hyperparameters). The human element of biasing solution spaces has been discussed in sociotechnical writing [14, 23, 66], in AI [52], in the context of "p-hacking" or "fishing expeditions" for results that fit a desired pattern [25], and was also the focus of Professor Isbell [40]'s NeurIPS 2020 keynote. Formalizing the process for how to draw conclusions from HPO, as we do here, has the potential to alleviate the effects of this type of human bias in ML systems.

Lastly, our insights concerning robustness also extend to growing areas in ML that use learning to guide hyperparemeter selection, such as meta-learning and neural architecture search [8, 17, 38, 39, 75]. While the assisting learning agents in those methods guide choosing hyperparameters for the trained output model, their own hyperparameters tend to be either manually tuned or chosen with more traditional HPO strategies, like grid search [74]. In other words, these processes can exhibit the bias problem discussed above and are therefore potentially subject to hyperparameter deception, which can be mitigated by the work we present here.