# OpenReview forum: "Hyperparameter Optimization Is Deceiving Us, and How to Stop It"
_NeurIPS.cc/2021/Conference — NeurIPS 2021 Poster_

### Official Review · Reviewer_n65K · 2021-07-05

**Rating:** 6
**Confidence:** 4

**Summary:**

This paper argues that hyperparameter optimization can lead researchers to draw opposite conclusions when comparing different algorithms. In particular, the authors show that this happens not only as a result of the choice of hyperparameters, but also as a result of the hyperparameter search procedure (e.g., including the search space choice).

To alleviate this concern, the authors propose a novel framework based on modal logic to determine whether a hyperparameter search procedure is "defended" against such deceptions. The argument and proposal is demonstrated through a running example based on work from Wilson et al., showing that according to a user-defined skepticism threshold one could either not draw any conclusion or draw the opposite conclusion from what is claimed in the paper.

**Limitations And Societal Impact:**

The authors have discussed the societal impact of their work.

**Main Review:**

The paper is very well written and easy to follow. In particular, I found the work well-organised and articulated, with helpful informal explanations following formal statements and an accessible introduction to the framework of modal logic.

By pointing out how hyperparameter optimization can lead to conflicting conclusions, the work highlights a key issue underlying a large number of current research works in machine learning. This makes it a highly relevant contribution, which I expect will foster better practice in the ML community. I particularly appreciated that the author focus on the hyperparameter search choices, such as the search space, rather than purely on hyperparameter values or statistical significance. This points out how hyperparameter optimization and statistical significance alone are not sufficient to draw meaningful conclusions when comparing competing algorithms. I believe this is a step further compared to prior work that indicated the possible deception arising from statistical testing or lack of proper hyperparameter search altogether.

The paper goes one step further to propose the first formalisation of this recurrent issue through the lens of modal logic, which is used to reason about how robust a conclusion is based on whether opposite conclusions could be drawn by alternative hyperparameter tuning procedures. The chosen framework is elegant, but I am slightly concerned as the core conclusion of the paper is quite intuitive and I am unsure to what extent the proposed formal treatment helps.

As room for improvement, I would have liked the authors to expand on their empirical section beyond the case study of Wilson et al. The reader would be further convinced of the utility of the framework if more practical examples were given in the main text or at least summarized. For instance, have the authors applied their framework to some prior work where the drawn conclusion was actually robust/defendable against deception? It would be interesting to get a sense for how often the proposed approach would refuse conclusions drawn from previous works and how often it would accept them, based on reasonable skepticism thresholds.

While the authors focused on a broad range of hyper-hyperparameters, I would like the authors to also elaborate on the role of the search space itself. If the hyperparameter search space is large enough, and so are the computational resources, one could argue that a close-to-optimal hyperparameter configuration can be found with good approximation. In such case, two algorithms could be compared with that close-to-optimal hyperparameter choice so that the drawn conclusion (e.g., algorithm A performs better than algorithm B) would be meaningful. I'd like the authors to comment on this setting. Would it be fair to reject the drawn conclusion that algorithm A outperforms algorithm B based on the fact that this does not hold for many alternative poorly configured search spaces? Would it not be a reliable conclusion itself that, under a close-to-optimal hyperparameter setting, one algorithms works better? This is also relevant in practice as the winning algorithm is also likely to be deployed with the optimal hyperparameter configuration, which makes the possible lack of robustness to search space definitions of secondary importance.

Overall, I am inclined towards acceptance as the paper points to highly significant problem, and proposes an approach to counteract it. I am only weakly inclined as the authors could have provided more empirical evidence and discussion for the usefulness of their formalism on a variety of empirical problems, including those where previous conclusions are confirmed. I am ready to change my score if the authors address my questions and concerns in the rebuttal.

**Time Spent Reviewing:**

3

---

> ### Author Response · Authors · 2021-08-10
> **Initial Author Response**
>
> Thank you for your review, and for recognizing the effort we put into making Sections 3 and 4 “accessible.” This was quite challenging, so we find it greatly rewarding that it was helpful. We respond to specific points below.
>
> * **A: Section 5/ core conclusion**: We also respond to related (though different) comments from reviewer oWpG (please see point B of our response to that reviewer).
>   * **i**:  While we provide a concrete defense in the paper for the running example that we use throughout the paper, which is useful for defending against deception in this context, this is not our core conclusion. We emphasize that we believe our main methodological contribution is different from this concrete defense to Wilson, et. al. In particular, our contribution is to provide a way to reason about hyperparameter optimization, with particular emphasis on evaluating what is “reasonable” for an ML scientist to do (and for a reviewer to double check). We have provided an example, which focuses on Wilson, et. al. as a case study to motivate why we have developed our notion of hyperparameter deception and how to prove defenses against it. Our defense is not intended to be taken as a general proof for all EHPO (See C below for related discussion, in relation to search spaces).
>
>   * **ii**: What is “reasonable” depends on specific experiment / domain context. We have provided a justification for why we think our choice of defense to deception in Wilson, et. al. is “reasonable” by developing a concrete defended reasoner for this task. Therefore, the defense we provide should not be taken as a general method to use to always avoid deception (we note this on lines 336-338, and re-emphasize this point on 395-397). We will try to draw this out clearer in the introduction, as well. Our logic is also a part of our contribution, as it enables us to prove non-deceptiveness for concrete, defended EHPO (of which our defense in Section 5 is just an example). As we note above, this logic should be taken as part of our core contribution (which is proved valid in the Supplementary Material).
>
>   * **iii**: The defense here to avoid deception in Wilson, et. al. is intended to be intuitive. In fact, this is in part why we picked it as our case study example throughout the paper.
> a: There are numerous examples of what we call hyperparameter deception in the literature. We cite some in Section 2 [7, 11, 16, 47, 48, 50, 56, 61]. For one example, [16] discusses this issue for a highly specialized NLP task (for which we are not domain experts). We picked Wilson, et. al. (which we briefly discuss at the very end of Section 2) because it deals with a benchmark task familiar to many in ML, as opposed to a highly specialized task like in [16], which we hoped would make our contribution more broadly accessible. We worried that if we presented our work in relation to something highly specialized that requires a lot of domain knowledge, it would have come across like our framework only works for that specialized case.
> b: Defenses are context-dependent, and will rely on context to ensure they are effective. Others may be more complicated than the one we propose here for Wilson, et. al. (See above, and C for related points). Our formalization is a tool to prove that such suggested defenses can guarantee defending against deception; this is one such proved defense.
>
> * **B: Additional experiments**:
>   * **i**: We performed other experiments to illustrate deception and defenses to it, but ultimately cut due to space constraints in favor of emphasizing what we saw to be our main contribution — our logic for reasoning about what it means to be defended against deception. We will include these results in our revisions.
>
>   * **ii**: As we believe is also alluded to in your feedback, we wanted to show how our framework could be used to reason about multiple problem domains. So, in addition to the computer vision experiments of Wilson, et. al., we also had a separate line of experiments from NLP. In particular, a task on Wikitext-2 using an LSTM as the model with Nesterov and Heavy Ball as the optimizers. This experiment was done in relation to work reported in Merity, S., Xiong, C., Bradbury, J., and Socher, R. Pointer sentinel mixture models. arXiv preprint arXiv:1609.07843, 2016. We have just initiated experiments to show that the defense we came up for Wilson, et. al. is effective in this setting, and will update our response when the results are ready. In the meantime, we provide a table figure to show that we can illustrate hyperparameter deception in this setting (we have a boxplot figure analogous to our current Figure 1, which we have uploaded anonymously to imgur and could embed in markdown, but we are not sure we are allowed to do that, so instead provide this table). We illustrate deception using two grids for the Wikitext-2 task, using an LSTM as the model. We run ten replicates for each optimizer / grid combination (a total of 40 runs).
>
> Nesterov is better when this grid used for HPO
>
> {1,5,10,15,20,25,30,35,40}HB is better this grid on the learning rate is used for HPO :{10,20,30,40}
>
> |          |               |             |       |       |       |        |       |       |       |       |       |       |       |
> |:--------:|:-------------:|:-----------:|:-----:|:-----:|:-----:|:------:|:-----:|:-----:|:-----:|:-----:|:-----:|:-----:|:-----:|
> |          | | Random Seed |       |       |       |        |       |       |       |       |       |   |  |
> |          |  Learning Rate   |      1      | 2   |   3    |   4   |   5   |   6   |   7   |   8   |   9   |  10   |Mean  |  Std  |
> | Nesterov |      1.0      |    5.700    | 5.150 | 5.140 | 5.140 | 5.170  | 5.140 | 5.000 | 5.140 | 7.130 | 4.990 | 5.370 | 0.649 |
> |          |      5.0      |    5.110    | 5.060 | 4.990 | 5.000 | 5.270  | 6.380 | 4.980 | 4.990 | 6.140 | 5.990 | 5.391 | 0.552 |
> |          |     10.0      |    6.140    | 6.350 | 6.320 | 5.300 | 6.120  | 6.510 | 5.900 | 5.340 | 6.590 | 6.360 | 6.093 | 0.453 |
> |          |     15.0      |    4.720    | 4.750 | 4.950 | 4.960 | 4.990  | 4.830 | 5.020 | 4.910 | 6.440 | 4.230 | 4.980 | 0.562 |
> |          |     20.0      |    6.350    | 5.280 | 6.340 | 6.120 | 6.080  | 7.450 | 4.740 | 7.570 | 7.040 | 5.020 | 6.199 | 0.976 |
> |          |     25.0      |    6.350    | 6.370 | 6.360 | 6.280 | 6.340  | 6.490 | 5.180 | 6.560 | 9.700 | 6.930 | 6.656 | 1.158 |
> |          |     30.0      |    6.370    | 6.010 | 6.890 | 6.130 | 6.210  | 5.000 | 5.980 | 5.010 | 5.000 | 6.420 | 5.902 | 0.671 |
> |          |     35.0      |    6.750    | 6.730 | 6.560 | 6.710 | 10.070 | 7.940 | 7.120 | 7.430 | 7.440 | 7.150 | 7.390 | 1.034 |
> |          |     40.0      |    6.730    | 6.560 | 6.780 | 6.600 | 7.220  | 6.990 | 6.390 | 7.450 | 7.390 | 6.660 | 6.877 | 0.367 |
> |    HB    |      1.0      |    4.730    | 5.720 | 5.720 | 5.730 | 6.150  | 6.130 | 6.160 | 6.150 | 7.040 | 4.730 | 5.826 | 0.693 |
> |          |      5.0      |    6.130    | 6.160 | 6.200 | 6.090 | 6.390  | 5.300 | 5.220 | 5.270 | 6.540 | 5.660 | 5.896 | 0.491 |
> |          |     10.0      |    6.390    | 6.380 | 6.320 | 6.350 | 6.340  | 6.510 | 6.720 | 7.100 | 7.050 | 6.360 | 6.552 | 0.300 |
> |          |     15.0      |    4.730    | 5.680 | 4.730 | 5.670 | 4.730  | 4.970 | 4.480 | 6.440 | 4.440 | 4.560 | 5.043 | 0.664 |
> |          |     20.0      |    5.660    | 4.990 | 5.690 | 4.970 | 6.110  | 6.130 | 6.160 | 7.140 | 5.010 | 6.330 | 5.819 | 0.700 |
> |          |     25.0      |    5.230    | 5.240 | 5.250 | 5.230 | 6.340  | 7.190 | 5.000 | 5.260 | 5.770 | 6.120 | 5.663 | 0.694 |
> |          |     30.0      |    5.330    | 5.760 | 5.030 | 5.120 | 5.990  | 6.510 | 5.500 | 5.390 | 5.510 | 6.360 | 5.650 | 0.500 |
> |          |     35.0      |    6.590    | 6.650 | 7.180 | 7.840 | 6.830  | 4.970 | 6.480 | 5.700 | 6.470 | 6.360 | 6.507 | 0.775 |
> |          |     40.0      |    6.630    | 6.640 | 6.590 | 6.550 | 7.220  | 5.440 | 6.160 | 7.570 | 7.040 | 9.310 | 6.915 | 1.023 |
>
>   * **iii**: We can also provide an example of showing that a result was non-deceptive (and confirming this via defended EHPO). However, we believe that to really show the value of our formalization and how it can be used to prove non-deceptive EHPO, it is more useful to show cases of deception (and how to avoid being deceived). We have a concrete example of this that we could develop from the Bayesian inference MCMC literature (that drew a defendable conclusion). We will include this experiment in the revision
>
> * **C: Role of the search space**:
>   * **i**: The comment here seems to be applying the concrete defense we have suggested to avoid deception in Wilson, et. al. to a problem that has behavior quite different from the deception we defend against in Wilson, et. al. We emphasize that the defense we have provided should not be taken as general to avoid all deception. Defenses (and the assumptions they need to make) will depend on context (we note this on lines 336-338, and re-emphasize this point on 395-397). It is possible that the defense we have developed for Wilson, et. al. would not be appropriate for the case you have outlined. However, we do not feel like we can comment with more low-level details without knowing more about the concrete problem context. We refer to our response in point A of this review for more on this / context and defenses.
>
>   * **ii**: We also note that we have included “t” in our formalization so that we are reasoning about what conclusions are possible to bring about in given time t. The note here seems suggestive to us of infinite compute, or more compute than would be available during an attempt to do “reasonable” hyperparameter optimization. Ultimately, the conclusions we draw are going to be in relation to what is deemed reasonable. We talk about this in Section 6, and how choice of hyper-HPs will affect this.

---

> > ### Author Response · Authors · 2021-08-30
> > **Follow up to reviewer n65k**
> >
> > Dear reviewer n65k,
> >
> > Thank you again for the initial detailed review. We hope that we addressed your concerns and adequately answered your questions. To summarize our initial response, we have provided additional details to clarify the contribution / core conclusion, and how Section 5 (the concrete defense we suggest for the illustrative example in the paper) relates to our broader contribution. We have also provided additional empirical results that we had initially cut from the draft due to space constraints. We would greatly appreciate if you could update your review with any further comments/questions.
> >
> > Thanks again,
> >
> > The authors of paper 1327

---

### Official Review · Reviewer_oWpG · 2021-07-15

**Rating:** 7
**Confidence:** 3

**Summary:**

This is a paper in the areas of  *epistemology* and the *philosophy of science*, with strong implications for empiricism in machine learning research, in particular, the way in which we draw conclusions about algorithms that are sensitive to the hyperparameters. The paper advances a formal system, based on modal logic, for reasoning and drawing conclusions about the experimental results of machine learning algorithms, and demonstrate that the framework naturally suggests defenses with guarantees
against deception.

**Limitations And Societal Impact:**

The broader societal impacts are discussed at length in the appendix but perhaps these should be included in the main text.

**Main Review:**

### Originality

The contribution of this paper appears to be original, at least in the context of machine learning research. Something to note: the general premise of this work is not unique to hyperparameter selection in machine learning. The observations and motivations also apply to science in general, e.g. experimental physics, chemistry, biology, and the social sciences such as psychology and economics. Here, the motivation, definitions, and examples are just specialized to machine learning algorithms (e.g. in particular, Definition 2). Therefore, it is questionable how novel and original this work is in the wider context of epistemology and scientific empiricism in general [26, 45], or whether this was just a straightforward specialization of the general framework to the context of ML research.

### Significance

It is debatable how well this paper fits the NeurIPS venue (i.e. a machine learning conference with a strong emphasis on methodological contributions). In many ways, it reads more like an essay or manifesto than it does a technical/scientific paper. Regardless, it does have significant implications for the ML research community. The contribution provides valuable insights and has the potential for impact on the current state of empirical validation in machine learning research, particularly towards the direction of making ML methods more reliable and robust.

Generally speaking, ML algorithms are highly sensitive to the choice of hyperparameters and random seeds. Yet the process itself of drawing conclusions about whether one algorithm outperforms another and gaining reliable knowledge in this regard is largely taken for granted. This has led to a proliferation of published methods that produce underwhelming results when considering a different set of hyperparameters than that tested by the original authors. This particular point is illustrated and argued compellingly on Page 3. Here they provide a poignant example of how one can reach both one conclusion and, at the same time, its logical complement by considering different HPs. This point is intuitively obvious but lacks a rigorous logical framework in which to reason about this phenomenon. This paper seeks to fill this gap. Another important point that is emphasized in this paper: "reproducibility is only part of the story for ensuring reliability. While necessary for guarding against brittle ﬁndings, it is not sufﬁcient. We can replicate results---even statistically signiﬁcant ones---that suggest conclusions that are altogether wrong." This is also an important point that is missing from the discourse in the community.

### Quality and clarity

This paper was of a high level of technical quality and clarity. It was well-written and made for an enjoyable read. I checked the details of the logical entailments and derivations in the main text portions of the supplementary material reasonably carefully.

### General Remarks

- On Definition 2 - this abstraction is fairly general and encompasses most of the typical cases. I understand it cannot be expected to capture everything, but there are a few obvious cases that it misses. An immediate example that comes to mind is that the so-called demon can manipulate not only the hyper-HPs $c \in \mathcal{C}$ and seeds $r \in \mathcal{I}$, but also things like the dataset split. Also, does it fully encompass both inductive and transductive learning settings?

**Time Spent Reviewing:**

6 hours

---

> ### Author Response · Authors · 2021-08-10
> **Initial Author Response**
>
> Thank you for recognizing our attempt to provide insights for more reliable and robust ML. We respond to individual points below.
>
> * **A: Epistemology and scientific empiricism**:
>   * **i**: In addition to the citations that you have referenced (which were included in the main body of the paper, i.e. 26 and 45), we would like to call attention to our more extensive treatment of this issue in the Supplemental Material, specifically Section F.2 paragraphs 2 and 3. We cite how our work talks about topics that have been a matter of discussion for a long time in more mature branches of science, and cite examples of alleviating measurement bias from evolutionary biology [27], programming languages research [51], p-hacking in statistics [25], and sociotechnical writing on computer systems [14, 23, 62]. We placed this discussion in the Appendix due to space constraints.
>
>   * **ii**: It is worth noting that it was the programming languages paper [51] that played a huge role in inspiring this work in the first place. One of the authors wanted to do something analogous for ML, as they found that such treatment was missing from the ML research community (particularly, current work on reproducibility, which is very important but orthogonal to our work presented here). We consider situating our work in relation to this prior work to be a strength of our submission.
>
>   * **iii**: You are correct that it is a specialization of this topic for ML. Our formalization to do this specialization is non-trivial, as we had to construct a valid logic (and prove its semantics to be valid, which we do in detail in the Supplementary material) that captures epistemic uncertainty that comes from hyperparameter optimization (what we call EHPO). To the best of our knowledge, this is the first such work to provide a contribution like this that is semantically meaningful for ML.
>
>   * **iv**: Moreover, we also demonstrate that our logic is useful for framing a concrete defense (i.e., we show how our formalization is useful to avoid being deceived in practice), which shows that our specialization to this area is also useful.
>
> * **B: Methodological contribution**: We also respond to related (though different) comments from reviewer n65k on this subject (please see point A of our response to that reviewer).
>   * **i**: While we provide a concrete defense in the paper for the running example that we use throughout the paper, which is useful for defending against deception in this context, we emphasize that we believe our main methodological contribution is different from this concrete defense. In particular, our contribution is to provide a way to reason about hyperparameter optimization, with particular emphasis on evaluating what is “reasonable” for an ML scientist to do (and for a reviewer to double check).
>
>   * **ii**: What is “reasonable” depends on specific experiment / domain context. Therefore, the defense we provide should not be taken as a general method to use to always avoid deception (we note this on lines 336-338, and re-emphasize this point on 395-397). We will try to draw this out clearer in the introduction, as well. Our logic is also a part of our contribution, as it enables us to prove non-deceptiveness for concrete, defended EHPO (of which our defense in Section 5 is just an example). As we note above, this logic is a technical contribution (which is proved valid in the Supplementary Material).
>
> * **C: Definition 2**: Definition 2 is sufficiently general to include dataset split (as this is a function of the random seed / sampling of the data set). We will make this more explicit in our discussion of Definition 2 or in footnote 2.

---

> > ### Author Response · Authors · 2021-08-30
> > **Follow up to review oWpG**
> >
> > Dear reviewer oWpG,
> >
> > Thank you again for the initial detailed review. We hope that we addressed the points you raised. To summarize our initial response, we have provided additional details (and cited relevant sections in the Supplementary Material) about how our work relates to epistemology and scientific empiricism, attempt to clarify our methodological contribution, and clarify that Definition 2 accounts for dataset split. We would greatly appreciate if you could update your review with any further comments/questions.
> >
> > Thanks again,
> >
> > The authors of paper 1327

---

### Official Review · Reviewer_31vJ · 2021-07-20

**Rating:** 4
**Confidence:** 4

**Summary:**

Motivated by recent work that empirically show how different conclusions can be made depending on the hyperparameter optimization (HPO) protocol, this paper aims to study the act of drawing conclusions from HPO in a rigorous manner. The authors use modal logic to formalize the idea of deception (where a wrong conclusion is made, or a conclusion is made when there is none to make), while taking uncertainty into account. The idea is that deception happens when two contradicting conclusions can be made given a realistic resource budget (time, compute, etc), so a HPO protocol that is safe to deception should be such that this can never happen. Using this idea, the authors propose an example HPO protocol for random search that is safe against deception.

**Limitations And Societal Impact:**

See main review above.

**Main Review:**

Studying the act of making conclusions based on HPO is important because, as the authors mention, it is possible, in the worst case, to game the system and produce results that are more favorable to a particular algorithm (by setting a specific hyperparameter search space or a specific seed), or to draw conclusions from noise. Studying this topic in a practically useful manner is very tricky though, because the selections of hyperparemeters to tune over, the search spaces for the hyperparemters, and also the computational budget to use for tuning, are all arbitrarily chosen.

I think this paper’s approach is interesting and novel, but I’m not sure it adds much to help practically prevent hyperparameter deception, at its current state. The paper provides a good foundation for analysis, and the main result, which is the t-non-deceptiveness axiom, makes intuitive sense. However, the paper provides no guidelines as to how to deal with the more practical issues like setting the search space, and the computational budget. In fact, in the experiments, a specific HPO protocol (two-phsed search) was used leading to some choice of hyperparameter ranges. This choice makes sense, but it still leaves room for deception (for example, if the hyperparameter to test performance mapping has several local optima, and one chooses a suboptimal range that includes the performance degradation point as the authors mentioned, but is still worse than what it could be. This might lead to misleading conclusions). Based on all of this, the paper seems more anecdotal than formal.

Other comments:
- Sections 3 and 4 seem overly long with too much detail and text. Maybe I’m missing something, but in my understanding, the main result of the paper is section 4.4, but this section doesn’t appear until much later in the paper, which makes the paper a bit hard to read.
- Definition 7 and therefore Algorithm 1 resembles bootstrapping. It’s interesting because Choi et al. uses the exact two-phase tuning protocol and reports results using bootstrapping.

Update:
After reading the other reviews and authors response, I am still unsure of the exact contributions of this paper (reasons are stated in the replies to the authors' response in the thread below), therefore, I am keeping my score.

**Time Spent Reviewing:**

6

---

> ### Author Response · Authors · 2021-08-10
> **Initial Author Response**
>
> Thank you for your review, and acknowledging the importance of this problem. We respond to your particular feedback points below.
>
> * **A: Length of Sections 3 and 4**:
>   * **i**: To study this problem formally in such a way that we can capture the uncertainty in it, we provide a formal modal logic framework. We believe that we need to show that the syntax and semantics of this logic is sound / provably valid (and provide extensive proofs in the Supplementary Material, which are necessarily elided in the main body of the paper). 4.4 displays the pay-off of this logic by showing what it is capable of capturing semantically; it ultimately shows why we designed it the way we did. However, we believe that without explaining the building blocks on which the axiom in 4.4 is based, it would not be clear that we have suggested a mathematically sound system for reasoning about hyperparameter deception formally, which we rely on for proving a valid defense in Section 5.
>
>   * **ii**: The framework that we contribute is formal. We exercise it for one particular case to demonstrate its utility in practice to avoid deception, and its success is not anecdotal. In our response n65k.B, we outline additional experiments that we cut due to space constraints, which we will include in revisions. We provide preliminary results that we initially had intended to include in the paper, but cut in favor of highlighting our main contribution (the framework for reasoning about deception formally in such a way that we can prove defenses to be guaranteed against deception).
>
> * **B: Practically preventing deception**:
>   * **i**: Reasonable defenses to deception are context-dependent, as they will require assumptions about the specific learning task (please refer to responses oWpG.B  and n65k.A for more discussion on this subject). We have provided a practical defense of deception for Wilson, et. al., our running example in the paper to categorize and motivate what we study in our submission.
>
>   * **ii**: Since concrete defenses necessarily depend on context, practical defenses will vary based on that context; it is not possible to have a one-size-fits-all solution for all learning tasks. Our contribution is to provide tools for proving defenses to be t-non-deceptive (and a valid logic that shows that these tools are sound). When practitioners are performing hyperparameter optimization for their specific contexts, they will need to reason about whether the procedure they’ve done can easily lead to deception, or if in fact it was a “reasonable” approach to HPO for that problem (we again refer to responses oWpG.B  and n65k.A for more discussion on “reasonable.”)
>
> * **C: Choi, et. al.’s 2-phased tuning protocol**: The two-phased search we implemented in our defense against Wilson seemed like a natural approach to avoid deception in this context. Our work and Choi et. al.’s (2020) work fits within a larger methodology of doing multiple passes of searches to do HPO, as this is generally considered reasonable practice for trying to narrow down the search space. For example, see [32] (2018) and Riquelme, Tucker, and Snoek. Deep Bayesian Bandits Showdown: An Empirical Comparison of Bayesian Deep Networks for Thompson Sampling. 2018. We will add these citations (along with Choi) to clarify that this is a common practice in our defense section (which we have modified and leveraged for our concrete defense to deception in Wilson, et. al.).

---

> > ### Comment · Reviewer_31vJ · 2021-08-25
> > **I'm confused**
> >
> > To the best of my understanding, it seems like the paper doesn’t provide any new guidelines as to how to select the search spaces for the hyperparameters (I’m assuming this is what the authors meant by context-dependent in the responses). In that case, I’m not sure how the running example from Wilson et al., is relevant to the paper; in fact, I’m not sure how Definition 7 is any different from statistical tests (like bootstrapping). I think my confusion comes from the fact that the author’s chosen 2-stage tuning protocol starts with an arbitrary search space. If we didn’t know a priori that Adam could perform better if we increase the range of epsilon to include bigger values, then it seems like we wouldn’t even have known that there would be inconsistent conclusions. In other words, if we ran Algorithm 1 with an initial search space that didn’t include epsilon, then we would have made the opposite conclusion (p) and would have believed it confidently because we are following a defended EHPO according to the paper).
> > If this is the case, then I’m really not sure what exactly this paper brings to the table that will help practically.
> >
> > Please correct me if I completely missed the point.

---

> > > ### Author Response · Authors · 2021-08-25
> > > **Re: Reviewer 31vj response to our author's response**
> > >
> > > Hi there,
> > >
> > > We are going to respond in-line to make sure that our response is as clear as possible.
> > >
> > > > To the best of my understanding, it seems like the paper doesn't provide any new guidelines as to how to select the search spaces for the hyperparameters (I'm assuming this is what the authors meant by context-dependent in the responses).
> > >
> > > It is not possible to provide reliable general recipes for how to select search spaces for all problems. We note in our response and in the paper (and you rightly mention here) that picking hyper-HPs is context-dependent. The point of the paper is **not** to propose ways of selecting hyperparameters or hyper-hyperparameters, **but rather to propose a way to concretely evaluate such an HPO setup in relation to the conclusions we can draw from it.**
> > >
> > > > In that case, I'm not sure how the running example from Wilson et al., is relevant to the paper;
> > >
> > > We picked Wilson  et. al. to show the problem in a specific context --- to have an example that concretely illustrates the reasoning process we are studying in order to ground it / make the exposition clearer. Our contribution is not to fix Wilson. We could have picked another of the examples of hyperparameter deception that we cite in Section 2 to illustrate our contribution. ```"We chose Wilson et al. [68] to illustrate this problem because the experiment does not require highly-specialized ML sub-domain expertise. Moreover, the paper is exceptionally well-cited[.]" (p. 3)``` Wilson et. al.'s choice leaves out hyperparameters (constraining HPO to learning rate), without explicitly justifying their choice (which then would have made that justification available to reviewers, to judge if it was reasonable). **We argue that this justification is a crucial part of testing algorithms empirically. Our theoretical framework provides a way to evaluate such justifications.**
> > >
> > > As we have responded to another reviewer, we have additional concrete / walked-through examples that we can include in our exposition. We chose to cut those due to space constraints in favor of describing our theoretical results. We will add them back in in revisions, so as to make clearer that our paper is not about Wilson et. al. specifically. **Rather, the point is more general and conceptual**: Wilson. et. al. had gaps in their reasoning process concerning how they draw conclusions from their empirical study. (To be clear, we are not saying that Wilson et. al. did bad or shoddy work; this problem had not been previously studied.) We formalize a reasoning process to reduce such gaps;  **we develop a formalization that gives us greater assurance about the conclusions we can draw from empirical investigations involving HPO.**
> > >
> > > > in fact, I'm not sure how Definition 7 is any different from statistical tests (like bootstrapping). I think my confusion comes from the fact that the author's chosen 2-stage tuning protocol starts with an arbitrary search space.
> > >
> > > The search space we chose is not arbitrary, for epsilon we use the broadest possible search space that would be acceptable to the community as a hyperparameter search space for this algorithm. In comparison, tuning just over the learning rate is not and led to hyperparameter deception. (It is of course possible that someone in the community could disagree with our assertion that our choice of search space is reasonable, but the important point is that we **justify this choice** and explain that justification, rather than eliding it. A reviewer, then, would need to concretely take issue with our reasoning process and provide a justification of their own reasoning process, in order to question the validity of the conclusions that we draw.)
> > >
> > > It happens to be that our conclusion is different from Wilson et. al.'s, illustrating that their setup is prone to deception. **Our results would be just as useful if our defended search validated Wilson, et. al.'s conclusion.** The main point is that our defense helps ensure that conclusions are not due to luck or happenstance, which makes the reasoning process concerning our belief **more rigorous**. So, even if we had come to the same conclusion as Wilson et. al., our defense enables us **to be more assured that that conclusion was based on our empirical investigation, rather than luck.**
> > >
> > > > If we didn't know a priori that Adam could perform better if we increase the range of epsilon to include bigger values, then it seems like we wouldn't even have known that there would be inconsistent conclusions. In other words, if we ran Algorithm 1 with an initial search space that didn't include epsilon, then we would have made the opposite conclusion (p) and would have believed it confidently because we are following a defended EHPO according to the paper).
> > >
> > > Because epsilon is a hyperparameter, it is reasonable in this case to choose to tune over it, so as to not leave out a whole portion of the hyperparameter search space. It just happens to be even more reasonable for this problem based on the empirical results that Wilson et. al. got with their search space and the math of the update rule for heavy ball. But we can ignore that second insight and instead cast what we argue to be **reasonably large / justifiable** search space for the **broad claim that Wilson et. al. make about adaptive and non-adaptive algorithm performance.** Separate from the math (but complementing it), it is the output of our phase-1 that indicates that large epsilon leads to better performance. However, it is not clear until we actually run our defense in phase-2 that this performance is better -- that we can justify our conclusions concerning empirical performance rigorously. **This is the point of our defense (and the contribution of our formalization more broadly): to increase our assurance/confidence in our belief concerning the conclusions we can draw from empirical investigations involving HPO.**
> > >
> > > > If this is the case, then I'm really not sure what exactly this paper brings to the table that will help practically.
> > >
> > > The paper is of practical utility because now, when we read a knowledge-claim based on HPO, we can refer to the ideas/ formalization in the paper **to evaluate whether belief in that claim is justified.** This formalization is lacking in the cases of hyperparameter deception that we cite briefly in Section 2 (and discuss in more detail in our responses to other reviewers). If these prior works had had these ideas at the ready, we believe that deception could have been avoided. **This has practical utility, even if it is not a strict recipe or tips and tricks to follow. It is a practical contribution of a different kind.**

---

> > > > ### Comment · Reviewer_31vJ · 2021-09-01
> > > > **I disagree**
> > > >
> > > > I’m still confused as to whether this paper has any meaningful contributions.
> > > >
> > > > I think the core reason behind my confusion is that the paper fails to show any real insights with respect to the Wilson example. The authors claim that Wilson et al., did not justify their choices for hyperparameter tuning, and that epsilon is a natural hyperparameter to tune. However, I believe that all of these insights came through recent findings. Reviewers at the time of Wilson et al., would have deemed their choices as justified, because it was common for many papers to only tune the learning rate. We are only confident with our current results because we don’t know what we don’t know; there might be findings in the future that flips the current set of conclusions.
> > > > This leads to my next point that the proposed defended random search EHPO cannot help detect “deception” at the level of what matters in the ML community. Wilson et al., is an excellent example of such deception, since it led to a widespread belief that Adam has worse generalization-ability than SGD. However, if the proposed algorithm (Algorithm 1) was used in place of Wilson’s tuning protocol at that time, and therefore, without the knowledge that epsilon must be tuned, I believe that the output would be confident in its conclusion.
> > > >
> > > > I believe that defending against “deception” is an extremely difficult task, and cannot be addressed by justifying choices and running Algorithm 1 (which I still believe is just bootstrapping).

---

> > > > > ### Author Response · Authors · 2021-09-01
> > > > > **Re: "I disagree"**
> > > > >
> > > > > We reiterate that Wilson, et. al. is an illustrative example used to situate our contribution, not the contribution itself. We provide numerous other examples in the paper (at the end of Section 2), and went with this one as a running example for reasons that are discussed in the paper. Please refer to our response to reviewer n65k concerning results for a different example in more detail, as that reviewer asked for additional motivating example empirical results. These results further support that our contribution has broader import -- our contribution is not about fixing the deception problem in a specific paper.
> > > > >
> > > > > We believe that our contribution stands, irrespective of the particulars of Wilson. We understand that this reviewer 31vj not think this was the best choice of example. Nevertheless, this example takes up a relatively short amount of space in the paper, given that the majority of the work presents a theoretically-backed discussion for how to reason about hyperparameter deception generally -- not just in relation to the running example we chose to use to illustrate the import of these concepts.

---

### Official Review · Reviewer_6ydM · 2021-07-21

**Rating:** 5
**Confidence:** 3

**Summary:**

This paper concerns an important and broad issue in machine learning regarding hyperparameter optimisation (HPO) and that it might lead to inconsistent outcomes and conclusions about approaches and their performance, in particular the performance that is relative to other approaches. The authors argue that the analysis of an HPO should be an object of study by itself. The paper takes into account some existing related work in regard to HPO. It provides definitions for an (HPO) log, an HPO procedure, an epistemic HPO procedure, to formalise the reasoning process regarding HPOs, among other definitions. It provides an illustration (an example) of how HPOs can be deceptive (that example is also referenced in further sections to help the narrative). As a possible solution, the authors suggest a process (they call it “epistemic hyperparameter optimization” or “EHPO”) and a logical framework (with syntax and semantics provided). That provides a way to analyse uncertainty in EHPO and reason about inconsistent HPO conclusions, to which such uncertainty might lead. The authors define a t-non-deceptive EHPO via a definition that depends on some time (i.e. “t-”) notion (relevant to execution time of experiments). They also show how, theoretically, a defended reasoner can be made, the use of which could result in a guaranteed t-non-deceptive EHPO. Then, they present a defended version of a random search for HPO and present a theorem statement regarding such a search’s t-non-deceptiveness under certain assumptions. Then, the authors explain and show results for an experiment using a modified version of the random search on the illustrative example, and finally they conclude the paper and provide a summary of some practical takeaways.

**Limitations And Societal Impact:**

The paper focuses on the execution time in regard to resource budget. The authors highlight that alternatives (such as energy) might be considered. The issue with the time (including e.g. in relation to the theorem 1), and probably up to some extent with some other resource types, is that e.g. parallel computations or more efficient computational techniques might significantly affect the definitions/considerations covered in the paper.

Some other limitations:

A. The paper talks about “a worst-case setting” (line 156) in regard to “a demon”; that means, if I understand correctly, that a demon, in the worst case, can be capable of the worst possible strategy (that also relates to the definition 5). And, we are still interested whether the demon can reliably execute it under time t. It is also mentioned that there is “the easier case of well-intentioned ML researchers” (line 235).

This could be potentially followed-up on in two related ways (I follow the paper here and use here the metaphorical “demon” narrative that is introduced in the paper, but I think it can be also formulated in a more formal way, and the argument would still stand):

A1. A demon might be fine that it does not always succeed. For example, it might try to deceive us in a limited time t, and it is fine that sometimes it won’t deceive us (or someone else). But sometimes, depending e.g. on particular random samples (across a whole HPO process, not just A_lambda runs), it still can sometimes generate deceptive results. If it succeeds though, we are then deceived, even if that deception was not _guaranteed_. That is, a deception attempt might be opportunistic (or at least with probability less than 1), and it still might be very harmful for science. (It also could be argued that a lot of bad actors in the world are probably opportunistic: they are fine to just try to trick, and they are okay if not everyone is tricked, but they might gain a lot when some victims are tricked.)

A2. That argument can be continued further for “well-intentioned ML researchers”. They might accidentally deceive us (and themselves), even though the chance of that is less than 1.

(A continued.) Because of that, it could be argued that:
i. The definition 5 is too strong because we still could be deceived sometimes even if not always (and that could be generally considered as bad). (There might be even cases when a demon might “almost” always deceive us and/or it might do so “almost” always in a reasonable time _t_, but that is already out of the scope of the definition 5.)
ii. There is probably no general way to reliably check that those conditions (i.e. with the probability and the expectation) from the definition 5 are satisfied, in particular for complex models that are optimised as part of HPOs.

B. The paper mentions “is always possible” on line 335 (and also on line 397 without a reference to “t”) and it also mentions “if we cannot easily evaluate whether [...]” in line 337. While it is appreciated that theoretically a t-defended reasoner has been constructed in the section 5, it might be the case that it is just practically (almost) impossible for quite a few scenarios (and potentially theoretically for some tasks, although not clear) to “evaluate whether [...]” (as in line 337).

C. Lines 363-365 talk about what a “good-faith actor can” do, and how that defends us “against adversaries”. However, it is not clear how hard it is to a good-faith actor to produce a log that would satisfy the requirements of the definition 7.

D. (It is probably obvious from the submission:) The theorem 1 depends on “random-search distributions” under considerations to be in a particular relationship. Because of that, somebody else (an “adversary” including a demon) might always argue that their choice of a distribution is as reasonable as (or even better than) than “our” choice.

E. The choice of delta (see e.g. line 380) is another parameter, and it is not exactly clear what delta to use.

F. It is unclear how far generally (in terms of its eventual reliability and strength) the algorithm 1 is from the defended random search EHPO described on page 8.

*

Societal impact: generally this work is expected to bring (huge) positive social impact. The authors have a broader impact section in the appendix. That is all really appreciated.

It might be important to note that (obviously) generally, most likely, any defended and/or guaranteed approach can be subject to some limitations (especially if approximations are used), and that means that results could not, most likely, be fully guaranteed*. (In particular, that might be the case if an adversary is fine to be opportunistic and they are fine to fail many times before they (e.g. accidentally) succeed.) (* - unless e.g. the full hyper-parameter space is enumerated.)


**Main Review:**

This is a very important submission. Its significance is huge. Thanks extremely much to the authors for their work.

The paper is written generally well.

The prior work is described and scoped well. Based on that, the idea and approach seem to be novel.

Note that one, potentially relevant, point does not seem to be covered in the related work review: the hyperparameter optimisation belongs to a general class of optimisation problems, and it probably would be relevant to cover any relevant literature regarding optimisation in general (or mention that there is none relevant).

Clarity:

A. In the Definition 2, it is not exactly clear whether a PRNG (Pseudo-random number generator) G is used only for running A_lambda-s or also for picking lambda-s (e.g. as part of a random search), or for both? The educated guess from the Definition 2 is that it is used only for the former (i.e. for running A_lambda-s) but that is not fully clear. (If that is the case, is it that the randomness of H* itself (e.g. for choosing different lambda-s) left “ambiguous”?) It would be helpful to have it all clarified. (A clarification for this would be also important to help interpret lines 242-244, where it is discussed what a demon could or could not control regarding a PRNG, and line 262 (to understand and verify whether all randomness is covered there or not)).

B. The line 166 talks about “We want to be sure that we will not be deceived by any logs the demon *could* produce” - is it generally realistic in practice?

C1. (A discussion point and a minor clarity point.) It is explained in lines 180-189 that statistics is not the choice for this work. It is appreciated that applications of statistics are generally made under certain assumptions and its use might give false confidence. However, it is not clear what other tool, rather than statistics, can be used to analyse situations with uncertainty (to quantify it), even though it is not easy to quantify that uncertainty, as the authors highlight. Moreover, the authors do seem to use probability theory/statistics themselves in the paper (e.g. in the definition 5, which is one of the key definitions in the submission, and in theorem 1), _in conjunction_ with modal logic (and that is appreciated).

C2. It is not clear what is meant by “out-of-band” in that context in line 187. (As in, e.g. if a normal distribution is used, can it be “out-of-band”?)

C3. (More a discussion point) Regarding that “Hyper-HP selection is not a random process; [...]” (lines 183-185): it could be argued that it is, from some point of view, if we consider that a choice of hyper-HPs by humans or machines follows some distribution, although it is appreciated that it is hard, if possible at all, to practically define such a distribution.

D. Clarification on the meaning of the definition 5 would help: partially because of another clarity point A described above, it is not fully clear what exactly should be guaranteed (e.g., should a whole deception process (from the moment when a demon does not know anything about the problem) be guaranteed; should a “replication” of that process be guaranteed only up to the extent of sampling from omega(L) and randomly selecting G (if any)?; or something else).

E. It would help to clarify what does “defends with a log size K R” mean in line 362.

F. For the empirical validation at the end of section 5, a modified version of an HPO was used. It is not clear why it was necessary to modify it. (Was it e.g. because of computational limitations?)

G. The word “guarantee” is used on line 393: that might feel too strong given assumptions and limitations.

Correctness:

A. I am not sure about lines 257-258 and lines 295-296: if they are read formally, given the definition 5, to the best of my understanding (and there are have been some things that are not fully clear e.g. see clarity points A and D above) and assuming that is the definition that should be applied to interpret them, it is presumably not proven that e.g. “Wilson’s results show [it is possible p in time t]”: I assume it was not checked in a strict way according to the full definition 5 (and I am not sure if it could be proofed formally for this example).

Given the clarity points, the correctness point and the limitations described in another section of the review, although I think this work is very important, sadly I don’t feel comfortable at this point to recommend its acceptance, my apologies. There is a chance though that: (a) I did misunderstand some points (in particular, I have had some questions described in this review regarding the definition 5 and related aspects, and that might be due to my misunderstanding); (b) some points are obvious and they just were not mentioned in the submission; (c) or/and some points are valid and they can be addressed by the camera-ready version submission if the paper is accepted. Given the authors’ rebuttal, to which I am looking forward, there is a chance that the score might be adjusted.

All parts of the review are provided to the best of my understanding and knowledge. Mistakes/misunderstandings in the review are possible. I kindly ask to point me to them, if any, if possible, please.

*

**For the update, please, see my message dated the 2nd of September 2021 (GMT).**

**Time Spent Reviewing:**

I have not tracked time. I generally conduct a review in a few sub-iterations, to be able to reflect (between them) on a submission/material under review for some time.

---

> ### Author Response · Authors · 2021-08-10
> **Initial Author Response (part 1)**
>
> Thank you for recognizing the importance/ significance of our submission, recognizing our novel approach, and your extensive feedback. We respond to your individual points below (thank you for enumerating them so that we can respond clearly to each one).
> Point about optimization more generally: To the best of our knowledge from doing a lit review, a similar study that is applicable to optimization more generally has not been done. We choose to scope our discussion to machine learning (see also our response oWpG.A for some related notes on epistemic work in other sciences).
>
> # Clarity
>
> * **Clarity A — Definition 2**: Yes, if the HPO procedure is (per your example) random search, it will use the specified PRNG. The point of us including this in our formalization is to be clear that in HPO, there is only one source of (pseudo-) randomness. Moreover, the demon (or any HPO runner) cannot hack the PRNG. That is, we are not considering adversaries that could directly control how data is ordered / submitted to the algorithm under evaluation (i.e., cheat in this way).  This is also what makes our logical construction regarding deception non-trivial: we are able to defend against strong adversaries that can game EHPO / control hyperparameter optimization, but this does not include cheating by hacking the random number generator.
>
> * **Clarity B — “could produce”**: We are discussing a thought experiment (lines 153-154, Supplementary Material C) that helps us describe the worst-case scenario that we want to defend against deception. This is intentionally strong / a maximally powerful adversary; if we can defend against an adversary that is this powerful, then we will also be defended against weaker or accidental deception.
>
> * **Clarity C1 - modal logic vs. statistics**: As you capture in this point, we use modal logic to capture the uncertainty of the problem we are formalizing. Modal logic is useful for helping us reason about higher-level properties and specifications that have uncertainty that is difficult to quantify. To quantify that level of uncertainty, we can use indexed modal logic (which we have done here). Probability theory is one such example of this (for a good primer, see https://plato.stanford.edu/entries/logic-probability/#ModaProbLogi ). We will clarify this more in the beginning of Section 4.  We are using this formalization / representation of probability for our indexed modal logic in Definition 5, which follows from the validity of modal logics (see previously linked primer). It is sound to use probability and modal logic in this way. We also discuss more details in the Supplementary Material concerning the soundness of our logic; we prove that is valid (Section D), which also supports the validity of Definition 5. For two other examples of the connections between modal logic and probability theory (there are many examples of different semantics; ours in Definition 5 is another such example), see https://onlinelibrary.wiley.com/doi/abs/10.1111/j.1755-2567.1967.tb00618.x and https://www.researchgate.net/publication/221551396_The_Modal_Logic_of_Probability
>
> * **Clarity C2 - “out-of-band”**: Thank you for pointing this out. It’s confusing and we will fix it. We mean to distinguish what we are describing from what is the target of reproducibility. That is, when we test a particular set of hyperparameters, we get some spread of results. Those results may be reproducible, but they can still be deceptive. That is, by changing the hyper-HPs that determine the hyperparameters, we can get a spread of results that is (in the worst case) completely non-overlapping with the results that we previously got (in terms of performance). This is what we are trying to show with Figure 1 re: Adam.
>
> * **Clarity C3 — “not a random process”**:  We will weaken this to say that “it is not reasonable to model as a random process,” as it is not something that we can practically model.
>
> * **Clarity D — Definition 5**: Please also refer to above (Clarity A; Clarity C1). There is no randomness in HPO (Definition 2) other than the pseudorandomness of the PRNG. The randomness in Definition 5 comes from this and the randomness in the demon’s actions (which is why we model a strategy in Definition 4 as a “randomized function”). We have deliberately modeled the system this way to make clear where randomness is coming from. As a result, we are able in Definition 5 to provide a guarantee: all the statements made on probability and expectation are made in terms of only the randomness in this formulation.
>
> * **Clarity E — KR log size**: We will clarify this. We mean that we need a log = K * R random search trials in order to be guaranteed defended (where K and R are both integers, K is the number of trials and R is the number of naive reasons — see lines 340, Definition 7, 354). Right now this explanation is split (see prior lines cite) so we will clean it up.
>
> * **Clarity F — Computational efficiency**: We prove in Section 5 that a defense is always possible, but that doesn’t mean it is practical. For the extended example of Wilson et. al. in the paper, we have to define concrete B_* and B_naive reasoners for our defense. We define them this way to make our implemented defense computationally tractable. We will clarify these details in the text.
>
> * **Clarity G — guarantees**: Based on the soundness of our formalization, it is guaranteed. Please see above (Clarity D, which also references Clarity A1 and Clarity C1).
>
> # Correctness
>
> * **Correctness A**: Under EHPO (which we map to Wilson et. al.’s work), they show that it is possible in time t to bring about p (♦tp). We know this because they produced results that led to concluding p. We also showed it is possible to bring about “not p” in our experiments (♦t¬p). “t” is how we denote  the maximum reasonable amount of time (Definition 5). So, “t” for Wilson is whatever maximum time they used; “t” for us is the time we used. For lines 257-258, Wilson’s work proves it is possible to bring about p in time t (♦tp); we show that not p can be brought about in time t. t is a maximal time budget, so you can think of it as the biggest values between Wilson and what we did (or the sum of both of those times, even) if it is easier to think about it concretely that way (for this example). Therefore, this is the deception we define formally in 4.4 (and informally in lines 295-296), because it is possible in time t to get us to believe p (what Wilson exhibited) and possible in time t to get us to believe not p (what we showed). (The negation of this is t-non-deceptiveness). This is also why our notion of deceptiveness (which we formally express as t-non-deceptiveness) has “t” in its name; it is necessarily dependent on this notion of reasonable upper time bound “t”.
>    * **i**: In short, In other words, ♦tp and ♦t¬p for Wilson and for our modified experiment, respectively, are proofs by exhibition. It’s “possible to bring about p and not p in some maximal t” because we demonstrate that it is so. It is possible to do this for many such p and many such t. We use this example to explain concretely what our logic means.
>    * **ii**: What is more difficult is showing impossibility. We note this in Section 5, lines 333-338, where we talk about proving t-defended reasoners. We prove that such reasoners are always possible, but acknowledge that finding them might not be computationally tractable.

---

> > ### Author Response · Authors · 2021-08-10
> > **Initial Author Response (part 2)**
> >
> > # Limitations
> >
> > * **Limitations - “t” as time**: We assign “t” the semantics of time, but could have assigned it the semantics of “energy” or any other monotonically increasing “cost” — even money. This “t” does not capture the interplay between multiple costs, as it was simpler for us to reason about one notion of cost. So handling all of these together is not valid for our logic; handling one of them is. We picked time because “compute time” is often reported as a limiting factor for how many experiments are produced in a paper (including how much HPO is done).
> >
> > * **Limitations A — The demon is meant to help with intuition**: The demon is how we think of / model the worst-case adversary (see Clarity B). If we can protect against this worst case, then we can protect against easier cases (e.g., accidentally-deceived, well-intentioned ML researchers). The demon is just intended as a useful way to understand this “worst case” intuitively. (Supplementary Material Section D goes more into the specific metaphor for worst case deceptive adversary / Descartes’ demon). The demon is not the formalization (and is presented as the intuition at the end of Section 3 / beginning at line 153). The formalization is the syntax and the semantics in Section 4, which the metaphor of the demon helps us explain concretely (i.e., grounds us in intuition), but is mathematically sound independent of this intuition.
> >
> > * **Limitations A1 and A2 — t and probability**: t captures both probability off success and time budget. That is, if with some probability a demon can deceive us in some amount of time, then the demon can reliably deceive us with a larger time budget.In our framework, we model the situation of a demon that can deceive us with some limited time budget t. If the demon fails to produce a deceptive result, it can use the strategy of just re-running (and then stopping running when it gets the result it wants).  In other words, being deceived sometimes (but not always) is equivalent to being deceived always in a larger amount of time; the demon can just keep trying. Our semantics capture this case (see lines 245-253, prior to Definition 5, which sets up this intuition, which is formally captured in Defintion 5).The same explanation holds for A2 (also see line 271). Based on this response, and our clarifications about Definition 5 above (notably in Clarity D, which also references Clarity C1 and A), we hope this clarifies that Definition 5 is not too strong.
> >
> > * **Limitations B — practicality**: Your point here captures an important outcome of our analysis. Our formalization shows that it is always possible to have a t-defended reasoner, but that does not mean it is always practical to construct one. This means that the takeaway is that we need to introduce learning-task specific analysis (which we recommend / underscore throughout Section 5, and in our takeaways in Section 6). We do not claim that our defense is general / applicable to all hyperparameter deception. It is a defense that we have proven for this particular problem context (Wilson, et. al.) We also discuss this and related points in responses oWpG.B and n65K.A.
> >
> > * **Limitations C — Satisfying Definition 7**: Definition 7 can be satisfied by producing a log that has K * R trials in it (See Clarity E). This is the only criteria to satisfy, and we will make this clearer.
> >
> > * **Limitations D — Theorem 1**: This is an important point, which we also raise in the paper. Any understanding of what could lead to deception has to include some notion of belief / what a person (e.g. researcher, reviewer) would consider allowable. We refer to Sections 5 and 6.
> >
> > * **Limitations E - delta**: Delta is a measure of skepticism that we want to use when drawing conclusions. As with above, what is chosen to be used comes down to a definition of what is reasonable for a particular domain (and ultimately the judgment of researchers, reviewers). It is important to be clear about how conclusions were drawn (which is one of our key takeaways, and the motivation for our logical framework), which may include some clear explanation of skepticism used to draw conclusions. This is partly why we showed results for several values of delta in our defense results. Ultimately, we do not believe it is reasonable to conclude that adaptive methods are worse than non-adaptive ones, and we think we have shown evidence to support this. It is of course possible that someone else disagrees with this analysis (if they have a different notion of skepticism). But we have provided our notion clearly and defended it with evidence.
> >
> > * **Limitations F — our defense implementation**: We recall that a defense is context-dependent and that we implement this defense for this problem (Wilson et. al). We do not make claims about how Algorithm 1 behaves more generally.
> >
> > * **Limitations, opportunistic adversary**: For addressing this point, please see our clarification of definition 5 (Clarity D, which also references Clarity A and Clarity C1; Correctness A). We also emphasize that it is always necessary to put assumptions on the problem we are studying. We have noted the assumptions we make, for example in Definition 2, our semantics of time in our logic, as well as the assumptions we put on hyper-HPs in our demonstrated defense.

---

> > > ### Author Response · Authors · 2021-08-30
> > > **Follow up to Reviewer 6ydm**
> > >
> > > Dear reviewer 6ydm,
> > >
> > > Thank you again for the initial detailed review. We hope that we addressed your concerns and adequately answered your questions. To summarize our initial response, we have provided additional details on points concerning clarity and discussed the correctness of Definition 5
> > > We would greatly appreciate if you could update your review with any further comments/questions.
> > >
> > > Thanks again,
> > >
> > > The authors of paper 1327

---

> ### Comment · Reviewer_6ydM · 2021-09-02
> **Update**
>
> Dear Authors, Dear Other Reviewers, Dear Area Chairs, Dear All,
>
> The other reviews and the authors' responses are very helpful. Thanks a lot.
>
> The authors have addressed some of the concerns and they have suggested some edits to be made to the submission. I also particularly thank the authors for their clarification regarding point "Correctness A" (as per the discussion regarding my review): that helps and clarifies things.
>
> My current opinion is that although the paper presents significant work for an important topic, acceptance might not be recommended without another comprehensive review of a modified version, potentially with additional work that could make the paper stronger (e.g. with more empirical results; or/and with more details or/and discussion in Section 5 regarding the presented empirical results, or/and with a different way of structuring that section regarding Algorithm 1 and the related results).
>
> I am sorry, I have kept my score at "5: Marginally below the acceptance threshold".
>
> Yours faithfully & sincerely,
> One of the reviewers

---

> > ### Author Response · Authors · 2021-09-02
> > **Re: 6ydm**
> >
> > Hi Reviewer 6ydm,
> >
> > We are a bit puzzled by this response to our rebuttal. It is unclear to us if you think our rebuttal is insufficient to your concerns, or sufficient and you have new concerns. We have provided concrete rewrites in our rebuttal to your concerns (which is the best that one can do in the NeurIPS review process, as there is no revise-and-resubmit option in the discussion period). We have provided additional empirical results, particularly in our response to reviewer n65k. We do not know what you mean concretely about "a different way of structuring that section regarding Algorithm 1 and related results."
> >
> > To help us make sense of your decision, we would ask that you please respond in kind to each of our points of clarification to your initial review (i.e., does our response allay concerns or not). We also ask that you consider what you would have hoped to see in a rebuttal (given what types of rebuttals are possible in NeurIPS, i.e. no revise-and-resubmit option) to your response--a rebuttal that could have led you to change your mind about the quality of the paper. Based on your response, it is not clear to us what we could have done differently (without a revise-and-resubmit procedure).
> >
> > In that vein, if we have responded satisfactorily to your concerns, we'd ask that you consider revising your score in relation to the process that NeurIPS does make available to us--that we have responded to your concerns in good faith with intent to revise our manuscript accordingly and that, given the constraints of the review process, our paper might be worth publishing, especially in light of the contribution (which you note, you think is a significant one and worth publishing).
> >
> > We acknowledge that the review and discussion process is a challenging one. However, we feel that it would be unfortunate to have a paper that a reviewer deems interesting and good work be rejected, after submitting a rebuttal that allayed that reviewer's concerns (upper bounded by what is possible in NeurIPs response format).
> >
> > Thank you,
> >
> > The authors

---

> > > ### Comment · Reviewer_6ydM · 2021-09-03
> > > **Response**
> > >
> > > Dear Authors,
> > >
> > > Thank you for your message.
> > >
> > > I appreciate the NeurIPS review process is somewhat limited since there is no revise-and-resubmit option.
> > >
> > > When making my decision, I took into account the other reviews and the authors’ responses (including responses to my review). Authors, thank you very much for your responses, they were very helpful.
> > >
> > > My decision was influenced by the concerns that had been raised earlier in the reviews (i.e. not new concerns), in particular regarding the empirical results. I think that either/both more details/discussion should be provided in Section 5 regarding the presented empirical results (e.g. about the delta; and about Algorithm 1 and regarding its role in the narrative given that it is somewhat different from Definition 7) or/and more empirical results (e.g. using Definition 7 directly if possible) might help.
> > >
> > > I appreciate that you mention that the presented defence (regarding the work by Wilson et al.) in Section 5 is context-dependent: because of that, maybe one of possible ways to restructure the paper might be to put Algorithm 1 and related parts into an appendix section and have another example (or more discussion) instead. (That is an example (i.e. a possible way) regarding what I meant by “a different way of structuring that section regarding Algorithm 1 and the related results”. However, there are many different ways to structure it, and it is for the authors to decide on a particular way, of course.)
> > >
> > > That (the empirical results) by itself is a reason why I don’t feel comfortable increasing my score, I am sorry.
> > >
> > > I appreciate that a different experiment has been described in some of the authors’ responses. However, it is hard to fully review it if it is not in a version of the submission.
> > >
> > > Again, I do appreciate the limitations of the review process. Sometimes it is possible to address issues during the rebuttal (even without a revise-and-resubmit option) and a score might change; sometimes it is not possible. I am really sorry, but in this case my opinion is that the score should not increase after the rebuttal (and I have come to this conclusion only at the end of the process after I considered all responses).
> > >
> > > Regarding your response and the points in it in general: that has been very helpful. As mentioned, I think that helped and clarified things. In some places you mentioned e.g. “We will clarify this more [...]”: since I don’t know what exact changes will be made, it is hard to comment further on that.
> > >
> > > Some clarifications for definitions might help. For example, it might potentially help to create a table (or some other way to show that) where all possible sources of and parameters for randomness (as per definitions) are listed, and it is shown how they interplay and matter. Also, for definition 5, it might help to explicitly show what _random_ variables are considered for P(...) and E(...) on the line after line 263.
> > >
> > > Moreover, some theoretical or simple empirical examples might be helpful to illustrate different aspects of definitions (e.g. to show a simple example with sigma(L) from definition 4; or to show how generally the process and framework are different from a simple statement “we should not arrive at a conclusion if in time t we can achieve both X and not X”; or what happens if we have a very huge budget T >> t, during which someone can try to find/construct an adversary that can trick us with some probability in the future in time t).
> > >
> > > > Clarity G — guarantees: Based on the soundness of our formalization, it is guaranteed. Please see above (Clarity D, which also references Clarity A1 and Clarity C1).
> > >
> > > I see how it can be guaranteed in theory under some assumptions, but in practice I don’t see how it can be _guaranteed_, I am sorry.
> > >
> > > ***
> > >
> > > If I miss/misunderstand anything, please, let me know.
> > >
> > > I hope it all helps. If more feedback would be helpful, please, let me know.
> > >
> > > ***
> > >
> > > Authors, thank you very much for your work on this very important submission, whose significance is huge I believe, and for the review process work. I am sorry, but after careful consideration, I came to the conclusion that I am not comfortable increasing my score for the submission (taking into account the review process overall including the rebuttal).
> > >
> > > Best regards,
> > > Yours faithfully,
> > > One of the reviewers

---

### Author Response · Authors · 2021-08-10
**Initial Author Response -- To All Authors**

Thank you all for your extensive and incisive feedback. We really appreciate you taking the time to read our paper, and were happy to see that it was generally seen as an enjoyable read with an important contribution. We respond individually to each reviewer below (and reference points across reviews where we think it might be helpful / there is overlap between reviewer comments). Thanks again, and we look forward to the discussion phase.

---

### Decision · Program_Chairs · 2021-09-27

**Decision:**

Accept (Poster)

**Comment:**

This paper studies an important issue in machine learning research, namely the possibility of drawing wrong conclusions due to specific hyperparameter choices. While it is common practice to optimize hyperparameters, via random search or more advanced global optimization algorithms, a growing body of work suggests that the search space can be the determinant factor. The authors argue for a logical framework to asses whether deception occurs (that is, contradictory conclusions can be drawn). The authors make theoretical contributions acknowledged by all reviewers. They then apply the framework to deception in the context of random search. However, while the topic is important and the approach novel (or unusual for NeurIPS community), it is unclear what its practical implications would be and how researchers could make use of the framework in more complex settings and how it would help them draw more robust conclusions.